# Rethinking Lipschitz Neural Networks and Certified Robustness: A Boolean Function Perspective

**Bohang Zhang**[1,3,4]   **Du Jiang**[1]   **Di He**[1,*]   **Liwei Wang**[1,2,*]
[1]National Key Laboratory of General Artificial Intelligence,
School of Intelligence Science and Technology, Peking University
[2]Center for Data Science, Peking University    [3]Peng Cheng Laboratory    [4]Pazhou Laboratory (Huangpu)
`{zhangbohang,dujiang,dihe}@pku.edu.cn,   wanglw@cis.pku.edu.cn`

## Abstract

Designing neural networks with bounded Lipschitz constant is a promising way to obtain certifiably robust classifiers against adversarial examples. However, the relevant progress for the important $\ell_\infty$ perturbation setting is rather limited, and a principled understanding of how to design expressive $\ell_\infty$ Lipschitz networks is still lacking. In this paper, we bridge the gap by studying certified $\ell_\infty$ robustness from a novel perspective of representing Boolean functions. We derive two fundamental impossibility results that hold for any standard Lipschitz network: one for robust classification on finite datasets, and the other for Lipschitz function approximation. These results identify that networks built upon norm-bounded affine layers and Lipschitz activations intrinsically lose expressive power even in the two-dimensional case, and shed light on how recently proposed Lipschitz networks (e.g., GroupSort and $\ell_\infty$-distance nets) bypass these impossibilities by leveraging *order statistic functions*. Finally, based on these insights, we develop a unified Lipschitz network that generalizes prior works, and design a practical version that can be efficiently trained (making certified robust training free). Extensive experiments show that our approach is scalable, efficient, and consistently yields better certified robustness across multiple datasets and perturbation radii than prior Lipschitz networks.

## 1 Introduction

Modern neural networks, despite their great success in various applications [22, 13], typically suffer from a severe drawback of lacking adversarial robustness. In classification tasks, given an image $x$ correctly classified by a neural network, there often exists a small adversarial perturbation $\delta$, making the perturbed image $x + \delta$ look indistinguishable to humans but fool the network to predict a wrong class with high confidence [57, 4].

It is well-known that the adversarial robustness of a neural network is closely related to its Lipschitz continuity [8, 61] (see Section 3.1). Accordingly, training neural networks with bounded Lipschitz constant has been considered a promising way to address the problem. A variety of works studied Lipschitz architectures for the ordinary Euclidean norm [61, 60, 36, 54], and recent works even established state-of-the-art (deterministic) certified $\ell_2$ robustness [55, 40]. However, when it comes to the more critical (realistic) $\ell_\infty$ perturbation setting, the progress seems to be rather limited. In fact, standard Lipschitz ReLU networks have been shown to perform poorly in terms of $\ell_\infty$ robustness [61, 24, 2]. While other more advanced Lipschitz networks have been proposed [2, 10], the achieved results are still not satisfactory even on simple datasets like MNIST. Until recently, Zhang et al. [73, 74] designed a quite *unusual* Lipschitz network based on a heuristic choice of the $\ell_\infty$-*distance function*, which surprisingly established state-of-the-art certified $\ell_\infty$ robustness on multiple datasets over prior works. Yet, it remains unclear why previous Lipschitz networks typically failed, and what is the essential reason behind the success of this particular $\ell_\infty$-distance network structure.

---

*Correspondence to Liwei Wang `<wanglw@cis.pku.edu.cn>` and Di He `<dihe@pku.edu.cn>`

36th Conference on Neural Information Processing Systems (NeurIPS 2022).

**Theoretical contributions**. In this work, we systematically investigate how to design expressive Lipschitz neural networks (w.r.t. $\ell_\infty$-norm) through the novel lens of representing discrete Boolean functions, which provides a deep understanding on the aforementioned problems. Specifically, we first figure out a fundamental limitation of standard Lipschitz networks in representing a class of logical operations called *symmetric Boolean functions* (SBF), which comprises the basic logical AND/OR as special cases. We prove that for any non-constant SBF of $d$ variables, there exists a finite dataset of size $\mathcal{O}(d)$ such that the certified $\ell_\infty$ robust radius must vanish as $\mathcal{O}(1/d)$ for any classifier induced by a standard Lipschitz network. Remarkably, since logical AND/OR operations correspond to perhaps the most basic classifiers, our result indicates an intrinsic difficulty of such networks in fitting high-dimensional real-world datasets with guaranteed certified $\ell_\infty$ robustness.

Our analysis can be readily extended into the Lipschitz function approximation setting. We point out the relationship between *monotonic* SBF and the *order statistics* (which are 1-Lipschitz functions), and then prove that any $d$-dimensional order statistic (including the max/min function) on a compact domain cannot be approximated by standard Lipschitz networks with error $\mathcal{O}(1 - 1/d)$, regardless of the network size. This impossibility result is significant in that: (i) it applies to all Lipschitz activations (thus extending prior works [2, 24]), (ii) it resolves an open problem raised recently in [43], and (iii) a *quantitative* lower bound of approximation error is established.

Equipped by the above impossibility results, we proceed to examine two advanced Lipschitz architectures: the GroupSort network [2] and the recently proposed $\ell_\infty$-distance net [73, 74]. We find that besides the linear operation, both networks incorporate other Lipschitz aggregation operations into the neuron design, especially the order statistic functions, thus shedding light on how they work. However, for the MaxMin network [2] — a computationally efficient version of the GroupSort network implemented in practice, representing Boolean functions and order statistics is possible only when the network is very *deep*. In particular, we prove that representing certain $d$-dimensional Boolean functions requires a depth of $\Omega(d)$, implying that shallow MaxMin networks are not Lipschitz-universal function approximators. In contrast, we show a *two-layer* $\ell_\infty$-distance net suffices to represent any order statistic function on a compact domain or even all Boolean functions. This strongly justifies the empirical success of $\ell_\infty$-distance net over GroupSort (MaxMin) networks.

**Practical contributions**. Our theoretical insights can also guide in designing better Lipschitz network architectures. Inspired by the importance of order statistics, we propose a general form of Lipschitz network, called SortNet, that extends both GroupSort and $\ell_\infty$ distance networks and incorporates them into a unified framework. Yet, the full-sort operation is computationally expensive and leads to optimization difficulties (as with the GroupSort network). We further propose a specialized SortNet that can be efficiently trained, by assigning each weight vector $\boldsymbol{w}$ using *geometric series*, i.e. $w_i$ proportional to $\rho^i$ for some $0 \le \rho < 1$. This leads to a restricted version of SortNet but still covers $\ell_\infty$-distance net as a special case. For this particular SortNet, we skillfully derive a stochastic estimation that gives an unbiased approximation of the neuron output without performing sorting operations explicitly. This eventually yields an efficient training strategy with similar cost as training standard networks, thus making certified robust training free. Extensive experiments demonstrate that the proposed SortNet is scalable, efficient, and consistently achieves better certified robustness than prior Lipschitz networks across multiple datasets and perturbation radii. In particular, our approach even scales on a variant of ImageNet, and surpasses the best-known result [69] with a 22-fold decrease in training time thanks to our "free" certified training approach.

The contribution and organization of this paper can be summarized as follows:

- We develop a systematic study for the expressive power of Lipschitz neural networks using the tools of Boolean function theory. We prove the impossibility results of standard Lipschitz networks in two settings: a) certified $\ell_\infty$ robustness on discrete datasets (Section 3.2); b) Lipschitz function approximation (Section 3.3).

- We provide insights into how recently proposed networks can bypass the impossibility results. In particular, we show that a *two-layer* $\ell_\infty$-distance net can precisely represent any Boolean functions, while *shallow* GroupSort networks cannot (Section 3.4).

- We propose SortNet, a Lipschitz network that generalizes GroupSort and $\ell_\infty$-distance net. For a special type of SortNet, we derive a stochastic training approach that bypasses the difficulties in calculating sorting operations explicitly and makes certified training free (Section 4).

- Extensive experiments demonstrate that SortNet exhibits better certified robustness on several benchmark datasets over baseline methods with high training efficiency (Section 5).

## 2 Related Work

Extensive studies have been devoted to developing neural networks with certified robustness guarantees. Existing approaches can be mainly divided into the following three categories.

**Certified defenses for standard networks**. A variety of works focus on establishing certified robustness for standard neural networks. However, exactly calculating the certified radius of a standard ReLU network is known to be NP-hard [28]. Researchers thus developed a class of relaxation-based approaches that provide a tight lower bound estimate of the certified robustness efficiently. These approaches typically use convex relaxation to calculate a bound of the neuron outputs under input perturbations layer by layer [65, 66, 64, 53, 42, 19, 62, 76]. See also [3, 15, 16, 47, 48, 68, 11, 35, 63, 23, 51] for more advanced approaches. However, most of these works suffer from high computational costs and are hard to scale up to large datasets. Currently, the only scalable convex relaxation approach is based on *interval bound propagation* (IBP) [42, 21, 75, 69, 52], but the produced bound is known to be loose [50], and a recent study showed that IBP cannot achieve enough certified robustness on simple datasets for any standard ReLU network [41].

**Certified defenses using Lipschitz networks**. On the other hand, Lipschitz networks inherently imply certified robustness, resulting in a *much simpler* certification process based on the output margin (see Proposition 3.1). Yet, most prior works can only handle the $\ell_2$-norm Lipschitz situation by leveraging specific mathematical properties such as the spectral norm [8, 71, 20, 61, 17, 46, 2, 36, 40] or orthogonality of weight matrices [39, 60, 54, 55]. For the $\ell_\infty$-norm, standard Lipschitz networks were shown to give only a vanishingly small certified radius [61]. Huster et al. [24] found that standard Lipschitz ReLU networks cannot represent certain simple functions such as the absolute value, which inspired the first expressive Lipschitz architecture called the GroupSort network [2]. Since then, GroupSort has been extensively investigated [10, 58], but its performance is still much worse than the above relaxation-based approaches even on MNIST. Recently, Zhang et al. [73, 74] first proposed a practical 1-Lipschitz architecture w.r.t. $\ell_\infty$-norm based on a special neuron called the $\ell_\infty$-distance neuron, which can scale to TinyImageNet with state-of-the-art certified robustness over relaxation-based approaches. However, despite its practical success, it is rather puzzling how such a simple architecture can work while prior approaches all failed. Answering this question may require an in-depth re-examination of Lipschitz networks (w.r.t. $\ell_\infty$-norm), which is the focus of this paper.

**Certified defenses via randomized smoothing**. As a rather different and parallel research line, randomized smoothing typically provides *probabilistic* certified $\ell_2$ robustness guarantees. Due to the wide applicability, randomized smoothing has been scaled up to ImageNet and achieves state-of-the-art certified accuracy for $\ell_2$ perturbations [34, 38, 9, 49, 72, 27]. However, certifying robustness with high probability requires sampling a large number of noisy inputs (e.g., $10^5$) for a single image, leading to a high computational cost at inference. Moreover, theoretical results pointed out that it cannot achieve non-trivial certified $\ell_\infty$ robustness if the perturbation radius is larger than $\Omega(d^{-1/2})$ where $d$ is the input dimension [70, 5, 30, 67].

## 3 The Expressive Power of Lipschitz Neural Networks

### 3.1 Preliminaries

**Notations**. We use boldface letters to denote vectors (e.g., $\boldsymbol{x}$) or vector functions (e.g., $\boldsymbol{f}$), and use $x_i$ (or $f_i$) to denote its $i$-th element. For a unary function $\sigma$, $\sigma(\boldsymbol{x})$ applies $\sigma(\cdot)$ element-wise on vector $\boldsymbol{x}$. The $\ell_p$-norm ($p \geq 1$) and $\ell_\infty$-norm of a vector $\boldsymbol{x}$ are defined as $\|\boldsymbol{x}\|_p = (\sum_i |x_i|^p)^{1/p}$ and $\|\boldsymbol{x}\|_\infty = \max_i |x_i|$, respectively. The matrix $\infty$-norm is defined as $\|\mathbf{W}\|_\infty = \max_i \|\mathbf{W}_{i,:}\|_1$ where $\mathbf{W}_{i,:}$ is the $i$-th row of the matrix $\mathbf{W}$. The $k$-th largest element of a vector $\boldsymbol{x}$ is denoted as $x_{(k)}$. We use $[n]$ to denote the set $\{1, \cdots, n\}$, and use $\boldsymbol{e}_i$ to denote the unit vector with the $i$-th element being one. We adopt the big O notations by using $\mathcal{O}(\cdot)$, $\Omega(\cdot)$, and $\Theta(\cdot)$ to hide universal constants.

**Lipschitzness**. A mapping $\boldsymbol{f} : \mathbb{R}^n \to \mathbb{R}^m$ is said to be $L$-Lipschitz continuous w.r.t. norm $\|\cdot\|$ if for any pair of inputs $\boldsymbol{x}_1, \boldsymbol{x}_2 \in \mathbb{R}^n$,

$$\|\boldsymbol{f}(\boldsymbol{x}_1) - \boldsymbol{f}(\boldsymbol{x}_2)\| \leq L\|\boldsymbol{x}_1 - \boldsymbol{x}_2\|. \tag{1}$$

If the mapping $\boldsymbol{f}$ represented by a neural network has a small Lipschitz constant $L$, then (1) implies that the change of network output can be strictly controlled under input perturbations, resulting in *certified* robustness guarantees as shown in the following proposition.

**Proposition 3.1.** (Certified robustness of Lipschitz networks) *For a neural network $\boldsymbol{f}$ with Lipschitz constant $L$ under $\ell_p$-norm $\|\cdot\|_p$, define the resulting classifier $g$ as $g(\boldsymbol{x}) := \arg\max_k f_k(\boldsymbol{x})$ for an input $\boldsymbol{x}$. Then $g$ is provably robust under perturbations $\|\boldsymbol{\delta}\|_p < \frac{c}{L}\operatorname{margin}(\boldsymbol{f}(\boldsymbol{x}))$, i.e.*

$$g(\boldsymbol{x} + \boldsymbol{\delta}) = g(\boldsymbol{x}) \quad \text{for all } \boldsymbol{\delta} \text{ with } \|\boldsymbol{\delta}\|_p < c/L \cdot \operatorname{margin}(\boldsymbol{f}(\boldsymbol{x})). \tag{2}$$

*Here $c = \sqrt[p]{2}/2$ is a constant depending only on the norm $\|\cdot\|_p$, which is $1/2$ for the $\ell_\infty$-norm, and $\operatorname{margin}(\boldsymbol{f}(\boldsymbol{x}))$ is the margin between the largest and second largest output logits.*

The proof of Proposition 3.1 is simple and can be found in Appendix B.1 or [39, Appendix P]. It can be seen that the robust radius is inversely proportional to the Lipschitz constant $L$.

**Standard Lipschitz networks**. Throughout this paper, we refer to standard neural networks as neural networks formed by affine layers (e.g., fully-connected or convolutional layers) and element-wise activation functions. Based on the Lipschitz property of composite functions, most prior works enforce the 1-Lipschitzness of a multi-layer neural network by constraining each layer to be a 1-Lipschitz mapping. For the $\ell_\infty$-norm, it is further equivalent to constraining the weight matrices to have bounded $\infty$-norm, plus using Lipschitz activation functions [2], which can be formalized as

$$\boldsymbol{x}^{(l)} = \sigma^{(l)}(\mathbf{W}^{(l)}\boldsymbol{x}^{(l-1)} + \boldsymbol{b}^{(l)}) \quad \text{s.t. } \|\mathbf{W}^{(l)}\|_\infty \leq 1 \text{ and } \sigma^{(l)} \text{ being 1-Lipschitz, } l \in [M]. \tag{3}$$

Here $M$ is the number of layers and usually $\sigma^{(M)}(x) = x$ is the identity function. The network takes $\boldsymbol{x}^{(0)} := \boldsymbol{x}$ as the input and outputs $\boldsymbol{x}^{(M)}$. For $K$-class classification problems, $\boldsymbol{x}^{(M)} \in \mathbb{R}^K$ and the network predicts the class $g(\boldsymbol{x}) := \arg\max_{k\in[K]} x_k^{(M)}$. We refer to the resulting network as a standard Lipschitz network.

## 3.2 Certified robustness on discrete Boolean datasets

In this section, we will construct a class of counterexamples for which certified $\ell_\infty$ robustness can be arbitrarily poor using standard Lipschitz networks. We focus on the *Boolean dataset*, a discrete dataset where both inputs and labels are Boolean-valued and the relationship between inputs and their labels $(\boldsymbol{x}^{(i)}, y^{(i)}) \in \{0,1\}^d \times \{0,1\}$ can be described using a Boolean function $g^{\mathrm{B}} : \{0,1\}^d \to \{0,1\}$. The key motivation lies in the finding that Boolean vectors correspond to the vertices of a $d$-dimensional hypercube, and thus are geometrically related to the $\ell_\infty$-distance metric. In particular, the $\ell_\infty$-distance between any two different data points in a Boolean dataset is always 1, which means that the dataset is *well-separated*. This yields the following proposition, stating that it is always possible to achieve *optimal* certified $\ell_\infty$ robustness on Boolean datasets by using Lipschitz classifiers.

**Proposition 3.2.** *For any Boolean dataset $\mathcal{D} = \{(\boldsymbol{x}^{(i)}, y^{(i)})\}_{i=1}^n$, there exists a classifier $\hat{g} : \mathbb{R}^d \to \{0,1\}$ induced by a 1-Lipschitz mapping $\hat{\boldsymbol{f}} : \mathbb{R}^d \to \mathbb{R}^2$, such that $\hat{g}$ can fit the whole dataset with $\operatorname{margin}(\hat{\boldsymbol{f}}(\boldsymbol{x}^{(i)})) = 1 \; \forall i \in [n]$, thus achieving a certified $\ell_\infty$ radius of $1/2$ by Proposition 3.1.*

The key observation in the proof (Appendix B.2) is that one can construct a so-called *nearest neighbor classifier* that achieves a large margin on the whole dataset and is *1-Lipschitz*. Based on Proposition 3.2, it is natural to ask whether standard Lipschitz networks of the form (3) can perform well on Boolean datasets. Unfortunately, we show it is not the case, even on a class of simple datasets constructed using *symmetric* Boolean functions.

**Definition 3.3.** A Boolean function $g^{\mathrm{B}} : \{0,1\}^d \to \{0,1\}$ is symmetric if it is invariant under input permutation, i.e. $g^{\mathrm{B}}(x_1, \cdots, x_d) = g^{\mathrm{B}}(x_{\pi(1)}, \cdots, x_{\pi(d)})$ for any $\boldsymbol{x} \in \{0,1\}^d$ and $\pi \in S_d$.

**Example 3.4.** Two of the most basic operations in Boolean algebra are the logical AND/OR, both of which belong to the class of symmetric Boolean functions. Other important examples include the exclusive-or (XOR, also called the parity function), NAND, NOR, and the majority function (or generally, the threshold functions). See Appendix A.2 for a detailed description of these examples.

**Theorem 3.5.** *For any non-constant symmetric Boolean function $g^B : \{0,1\}^d \to \{0,1\}$, there exists a Boolean dataset with labels $y^{(i)} = g^B(\boldsymbol{x}^{(i)})$, such that no standard Lipschitz network can achieve a certified $\ell_\infty$ robust radius larger than $1/2d$ on the dataset.*

**Implications**. Theorem 3.5 shows that the certified radius of standard Lipschitz networks must vanish as dimension $d$ grows, which is in stark contrast to the constant radius given by Proposition 3.2. Also, note that the logical AND/OR functions are perhaps the most basic classifiers (which simply make predictions based on the existence of binary input features). It is thus not surprising to see that standard Lipschitz networks perform poorly on real-world datasets (e.g., even the simple MNIST dataset where input pixels are almost Boolean-valued (black/white) [61]).

We give an elegant proof of Theorem 3.5 in Appendix B.3, where we also prove that the bound of $1/2d$ is *tight* in Proposition B.8. Moreover, we discuss the sample complexity by proving that a dataset of size $\mathcal{O}(d)$ already suffices to give $\mathcal{O}(1/d)$ certified radius (Corollary B.10). To the best of our knowledge, Theorem 3.5 is the first impossibility result that targets the certified $\ell_\infty$ robustness of standard Lipschitz networks with an *quantitative* upper bound on the certified radius. In the next section, we will extend our analysis to the function approximation setting and make discussions with literature results [24, 2].

### 3.3 Lipschitz function approximation

Classic approximation theory has shown that standard neural networks are universal function approximators [12, 37], in that they can approximate any continuous function on a compact domain arbitrarily well. For 1-Lipschitz neural networks, an analogous question is whether they can approximate all 1-Lipschitz functions accordingly. Unfortunately, the result in Section 3.2 already implies a negative answer. Indeed, by combining Proposition 3.2 and Theorem 3.5, $\hat{f}$ is clearly a 1-Lipschitz function that cannot be approximated by any standard Lipschitz network.

To gain further insights into the structure of unrepresentable 1-Lipschitz functions, let us consider the *continuousization* of discrete Boolean functions. For the symmetric case, one needs to find a class of 1-Lipschitz continuous functions that are also invariant under permutations. It can be found that a simple class of candidates is the *order statistics*, i.e. the $k$-th largest element of a vector. One can check that the $k$-th order statistic $x_{(k)}$ is indeed 1-Lipschitz and is precisely the continuousization of the $k$-threshold Boolean function defined as $g^{\mathrm{B},k}(\boldsymbol{x}) := \mathbb{I}(\sum_i x_i \geq k)$. In particular, $x_{(1)} = \max_i x_i$ and $x_{(d)} = \min_i x_i$ corresponds to the logical OR/AND functions, respectively. Importantly, note that any Boolean function that is both symmetric and *monotonic* is a $k$-threshold function, and vice versa (Appendix A.2). Therefore, $k$-threshold functions can be regarded as the most elementary Boolean functions, suggesting that the ability to express order statistics is necessary for neural networks.

Unfortunately, using a similar analysis as the previous section, we have the following theorem:

**Theorem 3.6.** *Any standard Lipschitz network $f : \mathbb{R}^d \to \mathbb{R}$ cannot approximate the simple 1-Lipschitz function $\boldsymbol{x} \to x_{(k)}$ for arbitrary $k \in [d]$ on a bounded domain $\mathcal{K} = [0,1]^d$ if $d \geq 2$. Moreover, there exists a point $\widehat{\boldsymbol{x}} \in \mathcal{K}$, such that*

$$|f(\widehat{\boldsymbol{x}}) - \widehat{x}_{(k)}| \geq \frac{1}{2} - \frac{1}{2d}. \tag{4}$$

We give a proof in Appendix B.4. The above theorem indicates that order statistics cannot be *uniformly* approximated using any standard Lipschitz network regardless of the network size. Moreover, note that the trivial constant function $\tilde{f}(\boldsymbol{x}) = 1/2$ already achieves an approximation error of $1/2$ uniformly on $\mathcal{K}$, implying that standard Lipschitz networks can hardly improve upon trivial solutions.

**Remark 3.7.** Theorem 3.6 can be easily generalized to weaker forms of non-uniform approximation, e.g., using the $\ell_p$-norm as distance metrics [45], by proving that there exists a hypercube $\mathcal{B}_\infty(\widehat{\boldsymbol{x}})$ centered at $\widehat{\boldsymbol{x}}$ with length $\Theta(1)$, such that $|f(\tilde{\boldsymbol{x}}) - \tilde{x}_{(k)}| \geq \Theta(1)$ holds for all $\tilde{x} \in \mathcal{B}_\infty(\widehat{\boldsymbol{x}})$ when $d \geq 2$. See Corollary B.13 for details.

**Discussion with prior works** [2, 24, 43]. The work of Anil et al. also gave negative results on the expressive power of standard Lipschitz networks[2] [2, Theorem 1]. They proved a different (weaker) version of Theorem 3.6, showing that if the activation function $\sigma$ is *monotonic*, the network cannot *precisely* represent non-linear 1-Lipschitz functions whose gradient norm is 1 almost everywhere (e.g., the absolute value function proved before by [24]). They did not give a quantitative approximation error. The intuition is that any monotonic non-linear 1-Lipschitz activation (e.g., ReLU) must have regions with slopes less than 1, leading to *gradient attenuation* during backpropagation. The authors thus attributed the reason to the activation function, which is not *gradient-norm preserving* (GNP). However, such an explanation is still not fully satisfactory, as GNP can be simply achieved using a non-monotonic activation (e.g., $\sigma(x) = |x|$). Consequently, one may expect that a standard Lipschitz network built on a suitable (non-monotonic) activation function can have sufficient expressive power. Such an idea is recently explored in [43], where the authors proved that using a general 1-Lipschitz piecewise linear activation with 3 linear regions, the corresponding network achieves the maximum expressive power compared with other Lipschitz activations and can approximate any *one-dimensional* 1-Lipschitz function. They pose the high dimension setting as an open problem.

---

[2] The result is described w.r.t. $\ell_2$-norm, but with some effort, it can be extended to the $\ell_\infty$-norm case.

Unfortunately, Theorem 3.6 addressed the open problem with negative answer, stating that such networks are not expressive even for the two-dimensional setting. It also implies that GNP is not *sufficient* to explain the failure of standard Lipschitz networks. Instead, we draw a rather different conclusion, arguing that the lack of expressiveness is due to the inability of *weight-constrained* affine transformations to perform basic Boolean operations (even in two dimensions). A further justification is given in Section 3.4. Finally, compared with Anil et al. [2], the form of Theorem 3.6 is more fundamental, in the sense that it does not make assumptions on the activation function, and it gives a quantitative error bound on the approximation that is arbitrarily close to the plain fit $f(\boldsymbol{x}) = 1/2$.

### 3.4 Investigating more advanced Lipschitz networks

Seeing the above impossibility results, we then examine two representative works of (non-standard) Lipschitz networks in literature: the GroupSort network [2] and the recently proposed $\ell_\infty$-distance net [73, 74]. Notably, both networks are Lipschitz-universal function approximators and thus fully expressive. The GroupSort network makes minimum changes to standard Lipschitz networks (3), by replacing element-wise activation $\sigma$ with GroupSort layers. GroupSort partitions the input vector into groups, sorts the sub-vector of each group in descending order, and finally concatenates the resulting sub-vectors. Since sorting is computationally expensive, the authors considered a practical version of GroupSort with a group size of 2, called MaxMin, which simply calculates the maximum and minimum pair by pair [6]. $\ell_\infty$-distance net, on the other hand, is fundamentally different from standard Lipschitz networks. Each neuron in an $\ell_\infty$-distance net is designed based on the $\ell_\infty$-distance function $y = \|\boldsymbol{x} - \boldsymbol{w}\|_\infty + b$ (with parameters $\{\boldsymbol{w}, b\}$). Despite the somewhat unusual structure, $\ell_\infty$-distance net has been shown to substantially outperform GroupSort (MaxMin) in terms of certified $\ell_\infty$ robustness according to [73], a puzzling thing to be understood.

We provide a possible explanation for this. We find that both networks incorporate order statistics into the neuron design, either explicitly (GroupSort) or implicitly ($\ell_\infty$-distance net), thus bypassing the impossibility result in Theorem 3.6. Indeed, the sorting operations in GroupSort explicitly calculate order statistics. As for $\ell_\infty$-distance net, we show its basic neuron can implicitly represent the max function on a bounded domain, by assigning the weight $\boldsymbol{w} = -c\mathbf{1}$ and the bias $b = -c$ with a sufficiently large constant $c$:

$$y = \|\boldsymbol{x} - \boldsymbol{w}\|_\infty + b = \max_i |x_i - (-c)| - c = \max_i x_i \quad \text{if } c \geq \max_i -x_i, \tag{5}$$

and thus can represent the logical OR operation. In general, we have the following theorem:

**Theorem 3.8.** *A two-layer $\ell_\infty$-distance net can exactly represent the following functions:* (i) *any discrete Boolean function;* (ii) *any continuous order-statistic function on a compact domain.*

We give a proof in Appendix B.5. Our proof uses the fundamental result in Boolean algebra that any Boolean function can be written in its disjunctive normal form (DNF, see Appendix A.1), which can be further reduced to using only the composition of logical OR operations of *literals* and thus be realized by a two-layer $\ell_\infty$-distance net. To represent order statistics, we formulate them as nested max-min functions, which can also be realized by a two-layer $\ell_\infty$-distance net. Therefore, the construction in our proof provides a novel understanding of the mechanism behind the success of $\ell_\infty$-distance nets, since *each $\ell_\infty$-distance neuron can be regarded as a basic "logical gate" and the whole network can simulate any Boolean circuit.*

For GroupSort networks with a group size $G \geq d$, a similar result holds. However, it is not the case for practically used MaxMin networks ($G = 2$), where we have the following impossibility results:

**Theorem 3.9.** *An $M$-layer MaxMin network $f : \mathbb{R}^d \to \mathbb{R}$ cannot approximate any $k$-th order statistic function on a bounded domain $\mathcal{K} = [0, 1]^d$ if $M \leq \lceil \log_2 d \rceil$ (no matter how wide the network is). Moreover, there exists a point $\widehat{\boldsymbol{x}} \in \mathcal{K}$, such that*

$$|f(\widehat{\boldsymbol{x}}) - \widehat{x}_{(k)}| \geq \frac{1}{2} - \frac{2^{M-2}}{d} \geq \frac{1}{4} \quad \text{if } M \leq \lfloor \log_2 d \rfloor. \tag{6}$$

**Theorem 3.10.** *Let $M_d$ be the minimum depth such that an $M_d$-layer MaxMin network can represent any (discrete) $d$-dimensional Boolean function. Then $M_d = \Omega(d)$.*

The above theorems show that MaxMin networks must be *very deep* in order to represent Boolean functions and order statistics, which is in stark contrast to Theorem 3.8, as a constant depth is sufficient for $\ell_\infty$-distance nets. Based on Theorem 3.10, we directly have the following corollary:

**Corollary 3.11.** *The function class induced by $M_d$-layer MaxMin networks is not a universal approximator to the $d$-dimensional 1-Lipschitz functions if $M_d = o(d)$.*

The proofs of Theorems 3.9 and 3.10 are deferred to Appendix B.6 and B.7, respectively. In particular, the proof of Theorem 3.10 is non-trivial and makes elegant use of Boolean circuit theory, so we present a proof sketch here. The key insight is that for any Boolean function, if it can be expressed by some MaxMin network $f$, then it can be expressed by a special MaxMin network with the same topology as $f$ such that all the weight vectors $\boldsymbol{w}$ are sparse with at most one non-zero element, either $1$ or $-1$ (Corollary B.23). This implies that *weight vectors have no use* in representing Boolean functions and thus MaxMin networks reduce to *2-ary Boolean circuits*, i.e. directed acyclic graphs whose internal nodes are logical gates including NOT and the 2-ary AND/OR. Note that for a 2-ary Boolean circuit that has $M$ layers and outputs a scalar, the number of nodes will not exceed $2^{M+1} - 1$ (achieved by a complete binary tree). However, the classic result in Boolean circuit theory (Shannon 1942) showed that for most Boolean functions of $d$ variables, a lower bound on the minimum size of 2-ary Boolean circuits is $\Omega(2^d/d)$, which thus yields $M = \Omega(d)$ and concludes the proof.

In Appendix, we also discuss the tightness of the above theorems. We prove that a depth of $\mathcal{O}(\log_2 d)$ is sufficient to represent any order statistic function using Boolean circuit theory (Theorem B.19), and a straightforward construction using DNF shows that a depth of $\mathcal{O}(d)$ is sufficient to represent any Boolean functions (Proposition B.20). Thus both theorems are tight.

Unfortunately, training deep MaxMin networks is known to be challenging due to optimization difficulties [10]. Consequently, prior works only use a shallow MaxMin network with no more than 4 layers [2, 10], which severely lacks expressive power. One possible solution is to increase the group size, and several works explored this aspect using toy examples and observed significant benefits empirically [10, 58]. However, a large group size involves computationally expensive sorting operations and makes the network hard to train [2], limiting its value in practice.

## 4   A Unified Framework of Lipschitz Neural Networks

The above theoretical results have justified order statistics as a crucial component in representing a class of Boolean functions, shedding light on how GroupSort and $\ell_\infty$-distance net work. Based on these insights, in this section, we will propose a unified framework of Lipschitz networks that take the respective advantage of prior Lipschitz architectures, and then give a practical (specialized) version that enables efficient training.

Consider a general Lipschitz network constructed using the following three basic types of 1-Lipschitz operations: (i) norm-bounded affine transformations, e.g. $y = \boldsymbol{w}^{\mathrm{T}}\boldsymbol{x}$ ($\|\boldsymbol{w}\|_1 \leq 1$) and $y = x + b$; (ii) 1-Lipschitz unary activation functions $\sigma$; (iii) order statistics. The first two types are extensively used in standard Lipschitz networks, while the last type is motivated by Section 3 and is of crucial importance. We propose the following network which naturally combines the above components:

**Definition 4.1.** (SortNet) Define an $M$-layer fully-connected SortNet $\boldsymbol{f}$ as follows. The network takes $\boldsymbol{x} = \boldsymbol{x}^{(0)}$ as input, and the $k$-th unit in the $l$-th hidden layer $x_k^{(l)}$ is computed by

$$x_k^{(l)} = (\boldsymbol{w}^{(l,k)})^{\mathrm{T}} \mathrm{sort}(\sigma(\boldsymbol{x}^{(l-1)} + \boldsymbol{b}^{(l,k)})), \quad \text{s.t. } \|\boldsymbol{w}^{(l,k)}\|_1 \leq 1, \quad l \in [M], k \in [d_l] \qquad (7)$$

where $d_l$ is the size of the $l$-th layer, and $\mathrm{sort}(\boldsymbol{x}) := (x_{(1)}, \cdots, x_{(d)})^{\mathrm{T}}$ calculates all order statistics of $\boldsymbol{x} \in \mathbb{R}^d$. The network outputs $\boldsymbol{f}(\boldsymbol{x}) = \boldsymbol{x}^{(M)} + \boldsymbol{b}^{\mathrm{out}}$. Here $\{\boldsymbol{w}^{(l,k)}\}$, $\{\boldsymbol{b}^{(l,k)}\}$ and $\boldsymbol{b}^{\mathrm{out}}$ are parameters.

It is easy to see that SortNet is 1-Lipschitz w.r.t. $\ell_\infty$-norm. We now show that SortNet is a general architecture that extends both GroupSort and $\ell_\infty$-distance networks into a unified framework.

**Proposition 4.2.** *Any GroupSort network with an arbitrary group size on a compact input domain can be represented by a SortNet with the same topological structure using activation $\sigma(x) = x$.*

**Proposition 4.3.** *Any $\ell_\infty$-distance net can be represented by a SortNet with the same topological structure by fixing the weights $\boldsymbol{w}^{(l,k)} = \boldsymbol{e}_1$ and using the absolute-value activation $\sigma(x) = |x|$.*

See Appendix C for a proof. As a result, SortNet can exploit the respective advantage of these Lipschitz networks. Compared with GroupSort, SortNet can freely use activation functions such as the absolute value, thus easily addressing the problem claimed in [24, 2]. Moreover, unlike GroupSort, the bias vector $\boldsymbol{b}^{(l,k)}$ in SortNet (7) can be assigned diversely for different neurons in the same layer. In this way, one can control the sorting behavior of each neuron individually by varying the bias value without disturbing the output of other neurons, which is very beneficial (see Appendix C.1 for details). Compared with $\ell_\infty$-distance net, SortNet adds linear transformation and incorporates all order statistics (rather than only the maximum), thus can represent certain functions more effectively.

**A practical version of SortNet**. As with GroupSort networks, we also design a practical (specialized) version of SortNet which enjoys efficient training. But different from the MaxMin network that reduces the group size, we keep the full-dimensional order statistics as they are crucial for the expressive power (Section 3.4). The key observation is that in (7), the only required computation is the linear combination of order statistics (i.e. $\boldsymbol{w}^{\mathrm{T}} \mathrm{sort}(\cdot)$), rather than the entire sorting results (i.e. $\mathrm{sort}(\cdot)$). We find that for certain carefully designed choices of the weight vector $\boldsymbol{w}$, there exist efficient approximation algorithms that can give a good estimation of $\boldsymbol{w}^{\mathrm{T}} \mathrm{sort}(\cdot)$. In particular, we propose an assignment of the weight vector that follows geometric series, i.e. $w_i$ proportional to $\rho^i$, in which case we have the following result:

**Proposition 4.4.** *Let $\boldsymbol{w} \in \mathbb{R}^d$ be a vector satisfying $w_i = (1 - \rho)\rho^{i-1}, i \in [d]$ for some $0 \leq \rho < 1$. Then for any vector $\boldsymbol{x} \in \mathbb{R}^d_+$ with non-negative elements,*

$$\boldsymbol{w}^{\mathrm{T}} \mathrm{sort}(\boldsymbol{x}) = \mathbb{E}_{\boldsymbol{s} \sim \mathrm{Ber}(1-\rho)}[\max_i s_i x_i]. \tag{8}$$

*Here $\boldsymbol{s}$ is a random vector following independent Bernoulli distribution with probability $1 - \rho$ being 1 and $\rho$ being 0.*

*Proof.* Without loss of generality, assume $x_1, \cdots, x_d$ are different from each other. Denote $j_1, \cdots, j_d$ as the sorting indices such that $\mathrm{sort}(\boldsymbol{x}) = (x_{j_1}, \cdots, x_{j_d})$. Then

$$
\begin{aligned}
\mathbb{E}_{\boldsymbol{s} \sim \mathrm{Ber}(\rho)}[\max_i s_i x_i] &= \textstyle\sum_{k \in [d]} \Pr_{\boldsymbol{s} \sim \mathrm{Ber}(\rho)}[\max_i s_i x_i = x_{j_k}]\, x_{j_k} \\
&= \textstyle\sum_{k \in [d]} \Pr_{\boldsymbol{s} \sim \mathrm{Ber}(\rho)}[s_{j_k} = 1 \text{ and } s_{j_i} = 0 \;\forall 1 \leq i < k]\, x_{j_k} \\
&= \textstyle\sum_{k \in [d]} (1 - \rho)\rho^{k-1} \cdot x_{(k)} = \boldsymbol{w}^{\mathrm{T}} \mathrm{sort}(\boldsymbol{x}). \qquad \square
\end{aligned}
$$

It is easy to check that the weight $\boldsymbol{w}$ in the above proposition satisfies $\|\boldsymbol{w}\|_1 \leq 1$, which guarantees the Lipschitzness. The non-negative condition on $\boldsymbol{x}$ in Proposition 4.4 holds when using a suitable activation function in neuron (7), such as the absolute value function.

Proposition 4.4 suggests that one can use $\max_i s_i x_i$ to give an *unbiased* estimation of $\boldsymbol{w}^{\mathrm{T}} \mathrm{sort}(\boldsymbol{x})$. In this way, the expensive sorting operation is avoided and replaced by a max operation, thus significantly reducing the computational cost in training. We give an efficiently GPU implementation for training SortNet in Appendix D. Note that $\boldsymbol{s}$ is a random Bernoulli vector, so the above calculation is similar to applying a mask on the input of each neuron, like dropout [56]. It means that the introduced stochasticity may further prevent overfitting and benefit generalization performance.

**Regarding the value of $\rho$**. When $\rho = 0$, only the maximum value is taken into the computation and the resulting network can recover the $\ell_\infty$-distance net by choosing the activation function $\sigma(x) = |x|$. This means the specialized SortNet still extends $\ell_\infty$-distance net and thus has sufficient expressive power. When $\rho > 0$, all order statistics become utilized. A simple way of selecting $\rho$ is to regard it as a hyper-parameter and set its value by cross-validation, which is adopted in our experiments. One can also consider treating $\rho$ as learnable parameters for each neuron that participate in the optimization process, but this involves calculating the gradient of $\rho$ which may be complicated due to the stochastic sampling procedure (8). We will leave the study as future work.

## 5 Experiments

In this section, we perform extensive empirical evaluations of the proposed SortNet architecture as well as various prior works in the certified $\ell_\infty$ robustness area. To show the scalability of different approaches, we consider a variety of benchmark datasets, including MNIST [33], CIFAR-10 [29], TinyImageNet [32], and ImageNet ($64 \times 64$) [7]. Due to space limitations, a complete training recipe is given in Appendix E. Our code and trained models are released at `https://github.com/zbh2047/SortNet`.

### 5.1 Experimental setting

**SortNet model configuration**. Since SortNet generalizes the $\ell_\infty$-distance net, we simply follow the same model configurations as [73] and consider two types of models. The first one is a simple SortNet consisting of $M$ fully-connected layers with a hidden size of 5120, which is used in MNIST and CIFAR-10. Like [73], we choose $M = 5$ for MNIST and $M = 6$ for CIFAR-10. Since SortNet is Lipschitz, we directly apply the margin-based certification method to calculate the certified accuracy

(Proposition 3.1). To achieve the best results on ImageNet-like datasets, in our second type of model we consider using a composite architecture consisting of a base SortNet backbone and a prediction head (denoted as SortNet+MLP). Following [73], the SortNet backbone has 5 layers with a width of 5120 neurons, which serves as a robust feature extractor. The top prediction head is a lightweight 2-layer perceptron with 512 hidden neurons (or 2048 for ImageNet), which takes the robust features as input to give classification results. We also try a larger SortNet backbone, denoted as SortNet+MLP (2x), that has roughly four times the training cost (see Appendix E.2 for architectural details). We use the same approach as [73] to train and certify these models, i.e. by combining margin-based certification for the SortNet backbone and interval bound propagation for the top MLP [21].

**Baseline methods and metrics**. We compare SortNet with representative literature approaches including relaxation-based certification (for standard networks), margin-based certification (using Lipschitz networks), and mixed-integer linear programming (MILP) [59]. In Appendix G, we also discuss randomized smoothing approaches [9, 49], which provide probabilistic guarantees rather than deterministic ones. For each method in these tables, we report five metrics: (i) training efficiency, measured by the wall-clock time per training epoch; (ii) certification efficiency, measured by the time needed to calculate the certified accuracy on the test dataset; (iii) the clean test accuracy without perturbation (denoted as Clean); (iv) the robust test accuracy under 100-step PGD attack (denoted as PGD); (v) the certified robust test accuracy (denoted as Certified). For a fair comparison, we reproduce most of baseline methods using the official codes and report the wall-clock time under the same NVIDIA-RTX 3090 GPU. These results are presented in Tables 1, 2 and 3. In Appendix H, we also show the training variance of each setting by running 8 sets of experiments independently, and full results (including the median performance) are reported in Table 9 and 10.

## 5.2 Experimental results

**Performance on MNIST.** The results are presented in Table 1. Following the common practice, we consider both a small perturbation radius $\epsilon = 0.1$ and a larger one $\epsilon = 0.3$. It can be seen that the SortNet models can achieve **98.14%** ($\epsilon = 0.1$) and **93.40%** ($\epsilon = 0.3$) certified accuracy, respectively, both of which outperform all previous baseline methods. In contrast, the GroupSort network can only achieve a trivial certified accuracy for $\epsilon = 0.3$. This matches our theory in Section 3.4, indicating that the expressive power of shallow MaxMin networks is insufficient in real-world applications.

**Performance on CIFAR-10.** The results are presented in Table 2. Following the common practice, we consider two perturbation radii: $\epsilon = 2/255$ and $\epsilon = 8/255$. Our models can achieve **56.94%** ($\epsilon = 2/255$) and **40.39%** ($\epsilon = 8/255$) certified accuracy, respectively. Moreover, the training approach proposed in Section 4 is very efficient, e.g., with a training time of **13~14** seconds per epoch. For both radii, our models perform the best among all existing approaches that can be certified in a reasonable time. Compared with relaxation-based methods, the certified accuracy of SortNet models is much higher (typically $+3 \sim +6$ point for both radii), despite our training speed being several times faster. Such results may indicate that certified $\ell_\infty$ robustness can be better achieved by designing suitable Lipschitz models than by devising relaxation procedures for non-Lipschitz models.

**Performance on TinyImageNet and ImageNet.** To demonstrate the scalability of SortNet models, we finally run experiments on two large-scale datasets: Tiny-ImageNet and ImageNet ($64 \times 64$). Notably, the ImageNet dataset has 1000 classes and contains 1.28 million images for training and 50,000 images for testing. Due to both the large size and the huge number of classes, achieving certified $\ell_\infty$ robustness on the ImageNet level has long been a challenging task.

Table 3 presents our results along with existing baselines. Among them, we achieve **18.18%** certified accuracy on TinyImageNet and achieve **9.54%** certified accuracy on ImageNet, both of which establish state-of-the-art results. The gap is most prominent on ImageNet, where our small SortNet+MLP model already outperforms the largest model of [69] while being **22** times faster to train. Even for the largest model (SortNet+MLP 2x), the training is still 7 times faster, resulting in a training overhead of 4 days using two GPUs. We suspect that continuing to increase the model size will yield better results, given the noticeable improvement of the larger model over the smaller one.

**Comparing with $\ell_\infty$-distance net**. As can be seen, SortNet models consistently achieve better certified accuracy than $\ell_\infty$-distance nets for all different datasets and perturbation levels, and the performance gap is quite prominent compared with the original work [73]. Very recently, a follow-up paper [74] significantly improved the performance of $\ell_\infty$-distance net using a carefully designed

Table 1: Comparison of our results with existing methods on MNIST dataset.

| Method | | Train Time (s) | MNIST ($\epsilon = 0.1$) | | | MNIST ($\epsilon = 0.3$) | | |
|---|---|---|---|---|---|---|---|---|
| | | | Clean | PGD | Certified | Clean | PGD | Certified |
| Relaxation | IBP [21] | 17.5 | 98.92 | 97.98 | 97.25 | 97.88 | 93.22 | 91.79 |
| | IBP [52] | 34.7 | 98.84 | – | 97.95 | 97.67 | – | 93.10 |
| | CROWN-IBP [75] | 60.3 | 98.83 | 98.19 | 97.76 | 98.18 | 93.95 | 92.98 |
| Lipschitz | GroupSort (MaxMin) [2] | – | 97.0 | 84.0 | 79.0 | 97.0 | 34.0 | 2.0 |
| | $\ell_\infty$-dist Net [73] | 17.2 | 98.66 | 97.85 | 97.73 | 98.54 | 94.62 | 92.64 |
| | $\ell_\infty$-dist Net+MLP [73] | 17.2 | 98.86 | 97.77 | 97.60 | 98.56 | 95.05 | 93.09 |
| | $\ell_\infty$-dist Net [74] | 17.0 | 98.93 | 98.03 | 97.95 | 98.56 | 94.73 | 93.20 |
| | SortNet | **10.6** | 99.01 | 98.21 | **98.14** | 98.46 | 94.64 | **93.40** |
| MILP | COLT [3] | – | 99.2 | – | 97.1 | 97.3 | – | 85.7 |

Table 2: Comparison of our results with existing methods on CIFAR-10 dataset.

| Method | | Time (s) | | $\epsilon = 2/255$ | | | $\epsilon = 8/255$ | | |
|---|---|---|---|---|---|---|---|---|---|
| | | Train | Certify | Clean | PGD | Certified | Clean | PGD | Certified |
| Relaxation | CAP [66] | 659.0 | 7,570 | 68.28 | – | 53.89 | 28.67 | – | 21.78 |
| | IBP [21] | 19.0 | 2.74 | 61.46 | 50.28 | 44.79 | 50.99 | 31.27 | 29.19 |
| | IBP [52] | 70.4 | 4.02 | 66.84 | – | 52.85 | 48.94 | – | 34.97 |
| | CROWN-IBP [75] | 87.2 | 7.01 | 71.52 | 59.72 | 53.97 | 45.98 | 34.58 | 33.06 |
| | CROWN-IBP [69] | 45.0 | 4.02 | – | – | – | 46.29 | 35.69 | 33.38 |
| Lipschitz | $\ell_\infty$-dist Net [73] | 19.7 | 1.73 | 60.33 | 51.55 | 50.94 | 56.80 | 36.19 | 33.30 |
| | $\ell_\infty$-dist Net+MLP [73] | 19.7 | 1.74 | 65.62 | 51.47 | 51.05 | 50.80 | 36.51 | 35.42 |
| | $\ell_\infty$-dist Net [74] | 18.9 | 1.73 | 60.61 | 54.28 | 54.12 | 54.30 | 41.84 | 40.06 |
| | SortNet | 14.0 | 8.00 | 65.96 | 57.03 | 56.67 | 54.84 | 41.50 | **40.39** |
| | SortNet+MLP | **13.4** | 8.01 | 67.72 | 57.83 | **56.94** | 54.13 | 41.58 | 39.99 |
| MILP | COLT [3] | 252.0 | $\sim 10^5$ | 78.4 | – | 60.5 | 51.7 | – | 27.5 |

Table 3: Comparison of our results with existing methods on TinyImageNet and ImageNet datasets.

| Method | TinyImageNet ($\epsilon = 1/255$) | | | | ImageNet $64 \times 64$ ($\epsilon = 1/255$) | | | |
|---|---|---|---|---|---|---|---|---|
| | Time (s) | Clean | PGD | Certified | Time (s) | Clean | PGD | Certified |
| IBP [21] | 735 | 26.46 | 20.60 | 14.85 | 11,026 | 15.96 | 9.12 | 6.13 |
| IBP [52] | 284 | 25.71 | – | 17.64 | – | – | – | – |
| CROWN-IBP [69] | 1,256 | 27.82 | 20.52 | 15.86 | 16,269 | 16.23 | 10.26 | 8.73 |
| $\ell_\infty$-dist Net+MLP [73] | 55 | 21.82 | – | 16.31 | – | – | – | – |
| $\ell_\infty$-dist Net [74] | 55 | 12.57 | 11.09 | 11.04 | – | – | – | – |
| SortNet+MLP | **39** | 24.17 | 20.57 | 17.92 | **715** | 13.48 | 10.93 | 9.02 |
| SortNet+MLP (2x larger) | 156 | 25.69 | 21.57 | **18.18** | 2,192 | 14.79 | 11.93 | **9.54** |

training strategy, creating a strong baseline on CIFAR-10. However, we find their approach does not suit the ImageNet-like datasets when the number of classes is large (see Appendix E.6). In contrast, SortNet models enjoy great scalability ranging from MNIST to ImageNet and consistently outperform [74]. The improvement is also remarkable for $\epsilon = 2/255$ on CIFAR-10 (+7.11% and +2.82% in clean / certified accuracy).

In Appendix F, we conduct ablation studies on CIFAR-10 by varying the value of $\rho$ and comparing SortNet models ($\rho > 0$) with $\ell_\infty$-distance net ($\rho = 0$), under *the same training strategy* in this paper without additional tricks. We observe a large gain in certified accuracy when switching from $\ell_\infty$-distance net to general SortNet. This empirically indicates that incorporating other order statistics has extra benefits in certified $\ell_\infty$ robustness than using only the maximum (the first order statistic).

# 6 Conclusion

In this paper, we study certified $\ell_\infty$ robustness from the novel perspective of representing Boolean functions. Our analysis points out an inherent problem in the expressive power of standard Lipschitz networks, and provides novel insights on how recently proposed Lipschitz networks resolve the problem. We also answer several previous open problems, such as (i) the expressive power of standard Lipschitz networks with general activations [43] and (ii) the lower bound on the depth of MaxMin networks to become universal approximators [58, 43]. Finally, guided by the theoretical results, we design a new Lipschitz network with better empirical performance than prior works.

## Limitations, Open Problems, and Broader Impact

**Regarding the $\ell_p$-norm**. One major limitation of this work is that we only focus on $\ell_\infty$ robustness. While such results may shed light on general $\ell_p$-norm settings when $p$ is large, it does not apply to the standard $\ell_2$-norm. In particular, in this case MaxMin is *equivalent* to the absolute value activation function in terms of expressive power [2], which contrasts to the $\ell_\infty$-norm case for which MaxMin networks are strcitly more expressive. Moreover, empirical results suggest that these $\ell_2$ Lipschitz networks may have sufficient expressive power [55] (although it remains a fantastic open problem to prove that MaxMin networks with bounded matrix 2-norm are universal approximators).

Based on the above finding, this work reflects an interesting "phase transition" in the expressive power of standard Lipschitz networks when $p$ is switched from 2 to a large number. Coincidentally, a similar limitation is also proved when using randomized smoothing, which suffers from the curse of dimensionality when $p > 2$ [70]. This raises an interesting question of why the effect of $p$ is very similar for both methods and how things change as $p$ increases.

**Beyond standard Lipschitz networks**. Another limitation is that our results apply only for standard Lipschitz networks. When the Lipschitz constant is constrained using carefully designed bounding methods [47, 18, 31, 51] (rather than a simple stacking of 1-Lipschitz layers), the robustness certification will be less efficient, but the resulting networks are likely to bypass the impossibility results in this paper. It is an interesting direction to study whether we can just use *standard networks* with a carefully-designed Lipschitz bounding method that can achieve good certified robustness while still keeping adequate efficiency.

**Other promising directions**. On the application side, it is interesting to study how to design efficient training approaches for the general SortNet models with learnable weights or learnable $\rho$. Another meaningful question is how to encode inductive biases into these Lipschitz networks (e.g., designing convolutional architectures) to better suit image tasks.

**Broader impact**. Interestingly, our theoretical results point out a surprising connection between MaxMin/$\ell_\infty$-distance networks and Boolean circuits. We believe the value of this paper may go beyond the certified robustness community and link to the field of theoretical computer science.

## Acknowledgement

This work is supported by National Science Foundation of China (NSFC62276005), The Major Key Project of PCL (PCL2021A12), Exploratory Research Project of Zhejiang Lab (No. 2022RC0AN02), and Project 2020BD006 supported by PKUBaidu Fund. Bohang Zhang would like to thank Ruichen Li and Yuxin Dong for helpful discussions. We also thank all the anonymous reviewers for the very careful and detailed reviews as well as the valuable suggestions. Their help has further enhanced our work.

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
