# A  Boolean Functions

In this section, we review some basic concepts in Boolean algebra, which will be frequently used in our subsequent analysis. We only present the content that is most relevant to this paper, in particular, the concept of *disjunctive normal form* and *symmetric Boolean functions*. Throughout this paper, we use $g^{\mathrm{B}} : \{0,1\}^d \to \{0,1\}$ to denote a Boolean function.

## A.1  Disjunctive normal form

Three most elementary Boolean functions in Boolean algebra are the logical AND, logical OR and logical NOT, denoted as $\wedge$, $\vee$, and $\neg$, respectively. The value of $\wedge_{i=1}^d x_i = x_1 \wedge \cdots \wedge x_d$ is 1 if and only if $x_i = 1 \; \forall i \in [d]$. Similarly, $\vee_{i=1}^d x_i = x_1 \vee \cdots \vee x_d$ is 0 if and only if $x_i = 0 \; \forall i \in [d]$. The logical NOT is a unary operation which outputs 1 if and only if the input is 0.

All Boolean functions can be expressed using the above three elementary operations as building blocks. Among all possible forms of expressions, the disjunctive normal form (DNF) is a canonical form used in Boolean algebra theory. To define a DNF, we will need the notion of conjunction/disjunction.

**Definition A.1.** Let $\boldsymbol{x} = (x_1, \cdots, x_d) \in \{0,1\}^d$ be a Boolean vector.

- A *literal* is an atomic formula consisting of a single element or its negation, i.e. $x_i$ or $\neg x_i$.

- A *literal conjunction* is a formula consisting of a set of literals linked by the logical AND operation, i.e. $x_{i_1} \wedge \cdots \wedge x_{i_r} \wedge \neg x_{j_1} \wedge \cdots \wedge \neg x_{j_s}$, $1 \le i_1 \le \cdots \le i_r \le d, 1 \le j_1 \le \cdots \le j_s \le d$, where $r, s \in \mathbb{N}$. Typically, the definition further requires that $i_1, \cdots, i_r, j_1, \cdots, j_s$ are different indices, otherwise the literal conjunction is *unsatisfiable* (i.e. always outputs 0). Similarly, a *literal disjunction* is a formula consisting of a set of literals linked by the logical OR operation.

- A logical formula is considered to be in its disjunctive normal form if it is a disjunction of one or more conjunctions of one or more literals.

We list some examples of DNF: $x_1$, $x_1 \wedge x_2$, $x_1 \vee x_2$, $(x_1 \wedge x_2) \vee (x_1 \wedge \neg x_2) \vee (\neg x_1 \wedge \neg x_2) \vee \neg x_4$, etc. A fundamental result in Boolean algebra is shown in the following:

**Fact A.2.** *Any satisfiable Boolean function can be written in its disjunctive normal form.*

## A.2  Symmetric Boolean functions

Symmetric Boolean functions are an important class of Boolean functions defined in Definition 3.3. In this subsection, we will delve into this concept and give more concrete examples. First note that due to the symmetry property, the output of such functions can only depend on the number of ones (or zeros) in the Boolean input. For this reason, they are also known as *Boolean counting functions*.

As simple examples, the logical AND and logical OR are symmetric Boolean functions. Similarly, the NAND/NOR operations (i.e. $\neg(\wedge_i x_i)$ and $\neg(\vee_i x_i)$) are also symmetric. Another important example is the XOR function, denoted as $\oplus$. XOR outputs 1 iff the number of ones in the input is odd, i.e.

$$x_1 \oplus \cdots \oplus x_d = 1 \quad \text{iff } \left(\textstyle\sum_i x_i\right) \bmod 2 = 1.$$

**Threshold functions**. If a symmetric Boolean function $g^{\mathrm{B}}$ further satisfies a property called monotonicity, then we call it a threshold function. Formally, a function $f$ is *monotonically increasing* if for all vectors $\boldsymbol{x}$ and $\boldsymbol{x}'$ such that $x_i \le x_i'$ ($\forall i \in [d]$), one has $f(\boldsymbol{x}) \le f(\boldsymbol{y})$. It is easy to see that symmetric monotonic Boolean function must have the form: $g^{\mathrm{B}}(\boldsymbol{x}) = \mathbb{I}(\sum_i x_i \ge k)$ with integer $k$, which gives the name "threshold function". Depending on $k$, there are a total of $d + 2$ different $d$-dimensional threshold functions (including two constant functions). In particular, for $k = 1$ and $k = d$, we can recover the logical OR/AND functions, respectively. When $k = \lceil d/2 \rceil$, the resulting function is called the *majority function*.

# B  Proof of Theorems in Section 3

This section provides all the missing proofs in Section 3. For the convenience of reading, we will restate each theorem before giving a proof.

## B.1 Proof of Proposition 3.1

**Proposition B.1.** (Certified Robustness of Lipschitz networks) *For a neural network $\boldsymbol{f} : \mathbb{R}^n \to \mathbb{R}^K$ with Lipschitz constant $L$ under $\ell_p$-norm $\|\cdot\|_p$, define the resulting classifier $g$ as $g(\boldsymbol{x}) := \arg\max_{k \in [K]} f_k(\boldsymbol{x})$ for an input $\boldsymbol{x}$. Then $g$ is provably robust under perturbations $\|\boldsymbol{\delta}\|_p < \frac{c}{L} \operatorname{margin}(\boldsymbol{f}(\boldsymbol{x}))$, i.e.*

$$g(\boldsymbol{x} + \boldsymbol{\delta}) = g(\boldsymbol{x}) \quad \text{for all } \|\boldsymbol{\delta}\|_p < \frac{c}{L} \operatorname{margin}(\boldsymbol{f}(\boldsymbol{x})). \tag{9}$$

*Here $c$ is a constant depending only on the norm $\|\cdot\|_p$, which is $1/2$ for $\ell_\infty$-norm, and $\operatorname{margin}(\boldsymbol{f}(\boldsymbol{x}))$ is the margin between the largest and second largest output logits.*

*Proof.* Let $g(\boldsymbol{x}) = y$. Suppose there exists a $\widehat{\boldsymbol{\delta}}$ such that $g(\boldsymbol{x} + \widehat{\boldsymbol{\delta}}) \neq g(\boldsymbol{x})$, and $f_j(\boldsymbol{x} + \widehat{\boldsymbol{\delta}}) \geq f_y(\boldsymbol{x} + \widehat{\boldsymbol{\delta}})$ for some $j \neq y$. We will prove that $\|\widehat{\boldsymbol{\delta}}\|_p \geq \frac{c}{L} \operatorname{margin}(\boldsymbol{f}(\boldsymbol{x}))$ where $c$ only depends on the norm.

Define $\widehat{\boldsymbol{z}} = f(\boldsymbol{x} + \widehat{\boldsymbol{\delta}})$, then $\widehat{z}_y \leq \widehat{z}_j$. We first bound the difference between outputs $\boldsymbol{z}$ and $\boldsymbol{f}(\boldsymbol{x})$ as follows:

$$\|\widehat{\boldsymbol{z}} - \boldsymbol{f}(\boldsymbol{x})\|_p \geq \|(\widehat{z}_y, \widehat{z}_j)^{\mathrm{T}} - ([f_y(\boldsymbol{x}), f_j(\boldsymbol{x}))^{\mathrm{T}}\|_p \tag{10}$$

$$= (|\widehat{z}_y - f_y(\boldsymbol{x})|^p + |\widehat{z}_j - f_j(\boldsymbol{x})|^p)^{1/p}. \tag{11}$$

In (10) we use the fact that zero out elements for a vector can only decrease its $\ell_p$-norm. Now consider the following optimization problem:

$$\min_{\widehat{\boldsymbol{z}}} |\widehat{z}_y - f_y(\boldsymbol{x})|^p + |\widehat{z}_j - f_j(\boldsymbol{x})|^p \quad \text{s.t. } \widehat{z}_y \leq \widehat{z}_j. \tag{12}$$

It is easy to prove that the minimum is attained when $\widehat{z}_y = \widehat{z}_j = (f_y(\boldsymbol{x}) + f_j(\boldsymbol{x}))/2$. Substituting the assignment into (11) yields

$$\|\widehat{\boldsymbol{z}} - \boldsymbol{f}(\boldsymbol{x})\|_p \geq \frac{\sqrt[p]{2}}{2}(f_y(\boldsymbol{x}) - f_j(\boldsymbol{x})) \tag{13}$$

which can be further lower bounded by $\frac{\sqrt[p]{2}}{2} \operatorname{margin}(\boldsymbol{f}(\boldsymbol{x}))$ based on the definition of margin. Finally, due to the Lipschitz property,

$$\|\widehat{\boldsymbol{z}} - \boldsymbol{f}(\boldsymbol{x})\|_p \leq L\|\boldsymbol{\delta}\|_p. \tag{14}$$

Therefore $\|\boldsymbol{\delta}\|_p \geq \frac{\sqrt[p]{2}}{2L} \operatorname{margin}(\boldsymbol{f}(\boldsymbol{x}))$, which concludes the proof. $\square$

**Remark B.2.** All inequalities in the above proof is tight by choosing a $\widehat{\boldsymbol{\delta}}$ with $\|\widehat{\boldsymbol{\delta}}\|_p = \frac{\sqrt[p]{2}}{2L} \operatorname{margin}(\boldsymbol{f}(\boldsymbol{x}))$ and a Lipschitz function $\boldsymbol{f}$ such that $\widehat{z}_y = f_y(\boldsymbol{x}) - \operatorname{margin}(\boldsymbol{f}(\boldsymbol{x}))/2$, $\widehat{z}_i = f_i(\boldsymbol{x}) + \operatorname{margin}(\boldsymbol{f}(\boldsymbol{x}))/2$ and $\widehat{z}_k = f_k(\boldsymbol{x})$ for all $k \neq y$ and $k \neq i$. This implies that the certified guarantee in the above proposition is tight if only the Lipschitz property is known.

## B.2 Proof of Proposition 3.2

**Proposition B.3.** *For any Boolean dataset $\mathcal{D} = \{(\boldsymbol{x}^{(i)}, y^{(i)})\}_{i=1}^n$, there exists a classifier $\hat{g} : \mathbb{R}^d \to \{0, 1\}$ induced by a 1-Lipschitz mapping $\hat{\boldsymbol{f}} : \mathbb{R}^d \to \mathbb{R}^2$ under $\ell_\infty$-norm, such that $\hat{g}$ can fit the whole dataset with $\operatorname{margin}(\hat{\boldsymbol{f}}(\boldsymbol{x}^{(i)})) = 1 \ \forall i \in [n]$, thus achieving an optimal certified $\ell_\infty$ robust radius of $1/2$.*

*Proof.* Without loss of generality, assume for each class (either 0 or 1), there is at least one sample. Define the nearest neighbor classifier $\hat{g} : \mathbb{R}^d \to \{0, 1\}$ as $\hat{g}(x) = \arg\max_{k \in \{0,1\}} \hat{f}_k(\boldsymbol{x})$ where

$$\hat{f}_k(\boldsymbol{x}) = -\min\{\|\boldsymbol{x} - \boldsymbol{x}^{(i)}\|_\infty : (\boldsymbol{x}^{(i)}, k) \in \mathcal{D}\}, \quad k \in \{0, 1\}.$$

Then for any $(\boldsymbol{x}^{(i)}, y^{(i)}) \in \mathcal{D}$, $f_{y^{(i)}}(\boldsymbol{x}) = 0$ and $f_{1-y^{(i)}}(\boldsymbol{x}) = 1$ because $\|\boldsymbol{x}^{(i)} - \boldsymbol{x}^{(j)}\|_\infty = 1$ for all $i \neq j$. Therefore, the classifier $g$ can correctly classify the whole dataset with margin $\operatorname{margin}(\hat{\boldsymbol{f}}(\boldsymbol{x}^{(i)})) = 1 \ (\forall i \in [n])$. The left thing is to prove the Lipschitzness of $\boldsymbol{f}$, which is equivalent to proving that each function $\hat{f}_k(\boldsymbol{x})$ is 1-Lipschitz w.r.t. $\ell_\infty$-norm. This simply follows from the fact that $\hat{f}_k$ is the composition of two 1-Lipschitz functions: the $\ell_\infty$-distance function $\|\boldsymbol{x} - \boldsymbol{x}^{(i)}\|_\infty$ and the minimum function. Since the composition of 1-Lipschitz functions is still 1-Lipschitz, we have concluded the proof. $\square$

## B.3 Proof of Theorem 3.5

We first present a core lemma which will be used to prove Theorem 3.5.

**Lemma B.4.** *Let $f(\boldsymbol{x}) = \sigma(\boldsymbol{w}^{\mathrm{T}}\boldsymbol{x} + b)$ be a d-dimensional function where $\|\boldsymbol{w}\|_1 \leq 1$ and $\sigma$ is a 1-Lipschitz scalar function, and let $\{(\boldsymbol{u}^{(i)}, \boldsymbol{v}^{(i)}) \in \mathbb{R}^d \times \mathbb{R}^d\}_{i=1}^n$ be n pairs of inputs. Then*

$$\sum_{i=1}^n \left| f(\boldsymbol{u}^{(i)}) - f(\boldsymbol{v}^{(i)}) \right| \leq \left\| \sum_{i=1}^n \left| \boldsymbol{u}^{(i)} - \boldsymbol{v}^{(i)} \right| \right\|_\infty.$$

*Proof.* First note that

$$
\begin{aligned}
\left| f(\boldsymbol{u}^{(i)}) - f(\boldsymbol{v}^{(i)}) \right| &= \left| \sigma(\boldsymbol{w}^{\mathrm{T}}\boldsymbol{u}^{(i)} + b) - \sigma(\boldsymbol{w}^{\mathrm{T}}\boldsymbol{v}^{(i)} + b) \right| \\
&\leq \left| \boldsymbol{w}^{\mathrm{T}}(\boldsymbol{u}^{(i)} - \boldsymbol{v}^{(i)}) \right| \quad\quad\quad (15) \\
&\leq |\boldsymbol{w}|^{\mathrm{T}} \left| \boldsymbol{u}^{(i)} - \boldsymbol{v}^{(i)} \right|. \quad\quad\quad (16)
\end{aligned}
$$

where (15) uses the Lipschitz property of $\sigma$. Therefore

$$\sum_{i=1}^n \left| f(\boldsymbol{u}^{(i)}) - f(\boldsymbol{v}^{(i)}) \right| \leq |\boldsymbol{w}|^{\mathrm{T}} \sum_{i=1}^n \left| \boldsymbol{u}^{(i)} - \boldsymbol{v}^{(i)} \right| \leq \|\boldsymbol{w}\|_1 \left\| \sum_{i=1}^n \left| \boldsymbol{u}^{(i)} - \boldsymbol{v}^{(i)} \right| \right\|_\infty, \quad (17)$$

where the last inequiality in (17) follows by using the Hölder's inequality. Finally, observing $\|\boldsymbol{w}\|_1 \leq 1$, we thus conclude the proof. $\square$

**Corollary B.5.** *Let $\boldsymbol{f} : \mathbb{R}^d \to \mathbb{R}^K$ be a standard Lipschitz network defined in (3), and let $\{(\boldsymbol{u}^{(i)}, \boldsymbol{v}^{(i)}) \in \mathbb{R}^d \times \mathbb{R}^d\}_{i=1}^n$ be n pairs of inputs. Then*

$$\left\| \sum_{i=1}^n \left| \boldsymbol{f}(\boldsymbol{u}^{(i)}) - \boldsymbol{f}(\boldsymbol{v}^{(i)}) \right| \right\|_\infty \leq \left\| \sum_{i=1}^n \left| \boldsymbol{u}^{(i)} - \boldsymbol{v}^{(i)} \right| \right\|_\infty.$$

*Proof.* Denote $\boldsymbol{f}^{(l)}(\boldsymbol{z})$ be the output of the $l$-th layer for a standard Lipschitz network $\boldsymbol{f}$ given input $\boldsymbol{z}$. Based on the definition in (3), for the $l$-th layer, $\sigma^{(l)}$ is 1-Lipschitz and $\|\mathbf{W}^{(l)}\|_\infty \leq 1$, which is equivalent to $\|\mathbf{W}_{j,:}^{(l)}\|_1 \leq 1$ for the weight vector of each neuron $j$ in layer $l$. One can thus apply Lemma B.4 for the $j$-th neuron, given arbitrary inputs $\{\boldsymbol{f}^{(l-1)}(\boldsymbol{u}^{(i)})\}_{i=1}^n$ and $\{\boldsymbol{f}^{(l-1)}(\boldsymbol{v}^{(i)})\}_{i=1}^n$:

$$\sum_{i=1}^n \left| f_j^{(l)}(\boldsymbol{u}^{(i)}) - f_j^{(l)}(\boldsymbol{v}^{(i)}) \right| \leq \left\| \sum_{i=1}^n \left| \boldsymbol{f}^{(l-1)}(\boldsymbol{u}^{(i)}) - \boldsymbol{f}^{(l-1)}(\boldsymbol{v}^{(i)}) \right| \right\|_\infty. \quad (18)$$

This is equivalence to the following inequality due to the definition of $\ell_\infty$-norm:

$$\left\| \sum_{i=1}^n \left| \boldsymbol{f}^{(l)}(\boldsymbol{u}^{(i)}) - \boldsymbol{f}^{(l)}(\boldsymbol{v}^{(i)}) \right| \right\|_\infty \leq \left\| \sum_{i=1}^n \left| \boldsymbol{f}^{(l-1)}(\boldsymbol{u}^{(i)}) - \boldsymbol{f}^{(l-1)}(\boldsymbol{v}^{(i)}) \right| \right\|_\infty. \quad (19)$$

which is a recursive formula. Applying (19) recursively from the last layer to the first yields the desired result. $\square$

We are now ready for the proof of Theorem 3.5, which is restated below.

**Theorem B.6.** *For any non-constant symmetric Boolean function $g^B : \{0,1\}^d \to \{0,1\}$, there exists a Boolean dataset with labels $y^{(i)} = g^B(\boldsymbol{x}^{(i)})$, such that no standard Lipschitz network can achieve a certified $\ell_\infty$ robust radius larger than $1/2d$ on the dataset.*

*Proof.* We first introduce some notations. Denote $\mathcal{S}_k = \left\{ \boldsymbol{x} \in \{0,1\}^d : \sum_i x_i = k \right\}$ as the set containing all Boolean vectors with $k$ ones. It follows that $\{\mathcal{S}_k\}_{k=0}^d$ is a partition of the set of $d$-dimensional Boolean vectors $\{0,1\}^d$. For a Boolean function $g$, define the positive set and the negative set as $\mathcal{S}^+ = \left\{ \boldsymbol{x} \in \{0,1\}^d : g^B(\boldsymbol{x}) = 1 \right\}$ and $\mathcal{S}^- = \left\{ \boldsymbol{x} \in \{0,1\}^d : g^B(\boldsymbol{x}) = 0 \right\}$, respectively.

Recall that the output of a symmetric Boolean function depends only on the number of ones in the input (Appendix A.2). Therefore, instead of using the positive/negative set, we can use the following two sets to equivalently characterize a symmetric Boolean function $g^B$:

$$\mathcal{N}^+ = \left\{ k \in \{0, \cdots, k\} : \mathcal{S}_k \cup \mathcal{S}^+ \neq \emptyset \right\}, \quad \mathcal{N}^- = \left\{ k \in \{0, \cdots, k\} : \mathcal{S}_k \cup \mathcal{S}^- \neq \emptyset \right\}.$$

It turns out that $\mathcal{S}^+ = \bigcup_{k \in \mathcal{N}^+} \mathcal{S}_k$, $\mathcal{S}^- = \bigcup_{k \in \mathcal{N}^-} \mathcal{S}_k$, and $\mathcal{N}^+ \cup \mathcal{N}^- = \{0, \cdots, d\}$.

Since $g^B$ is non-constant, both $\mathcal{N}^+$ and $\mathcal{N}^-$ are non-empty. Therefore, there must exist two adjacent integers $p, q \in \{0, \cdots, d\}$, $p - q = 1$, such that either $(p \in \mathcal{N}^+, q \in \mathcal{N}^-)$ or $(p \in \mathcal{N}^-, q \in \mathcal{N}^+)$. Consider the following set of Boolean vector pairs:

$$\mathcal{T} = \{(\boldsymbol{u}, \boldsymbol{v}) : \boldsymbol{u} \in \mathcal{S}_p, \boldsymbol{v} \in \mathcal{S}_q, \|\boldsymbol{u} - \boldsymbol{v}\|_1 = 1\}. \tag{20}$$

The set $\mathcal{T}$ satisfies the following three properties:

- $g^B(\boldsymbol{u}) \neq g^B(\boldsymbol{v})$ for all $(\boldsymbol{u}, \boldsymbol{v}) \in \mathcal{T}$;

- The size of the set can be calculated by

$$|\mathcal{T}| = |\mathcal{S}_p|p = |\mathcal{S}_q|(d - q) = \tfrac{d!}{q!(d-p)!},$$

by using $|\mathcal{S}_k| = \binom{d}{k}$;

- $\left\| \sum_{(\boldsymbol{u}, \boldsymbol{v}) \in \mathcal{T}} |\boldsymbol{u} - \boldsymbol{v}| \right\|_\infty = \tfrac{(d-1)!}{q!(d-p)!} = \tfrac{|\mathcal{T}|}{d}$. This can be seen from the symmetry of the vector $\sum_{(\boldsymbol{u}, \boldsymbol{v}) \in \mathcal{T}} |\boldsymbol{u} - \boldsymbol{v}|$ and the fact that all vectors $|\boldsymbol{u} - \boldsymbol{v}|$ are unit (one-hot).

Based on the last property, given any standard Lischitz network $\boldsymbol{f}$, one can apply Corollary B.5 on $\mathcal{T}$, which yields

$$\left\| \sum_{(\boldsymbol{u}, \boldsymbol{v}) \in \mathcal{T}} |\boldsymbol{f}(\boldsymbol{u}) - \boldsymbol{f}(\boldsymbol{v})| \right\|_\infty \leq \frac{|\mathcal{T}|}{d}. \tag{21}$$

Using the definition of $\ell_\infty$-norm and noting that the output of function $\boldsymbol{f}$ is 2-dimensional, (21) implies that

$$\sum_{k \in \{0,1\}} \sum_{(\boldsymbol{u}, \boldsymbol{v}) \in \mathcal{T}} |f_k(\boldsymbol{u}) - f_k(\boldsymbol{v})| \leq \frac{2|\mathcal{T}|}{d}. \tag{22}$$

By the Pigeon Hole principle, there must exist a pair of points $(\boldsymbol{u}, \boldsymbol{v}) \in \mathcal{T}$, such that

$$\sum_{k \in \{0,1\}} |f_k(\boldsymbol{u}) - f_k(\boldsymbol{v})| \leq \frac{2}{d}. \tag{23}$$

Now it is time to construct the dataset. Define a Boolean dataset $\mathcal{D} = \{(\boldsymbol{z}, g^B(\boldsymbol{z})) : \boldsymbol{z} \in \mathcal{S}_p \cup \mathcal{S}_q\}$. Then there must exist two data $(\boldsymbol{u}, g^B(\boldsymbol{u}))$ and $(\boldsymbol{v}, g^B(\boldsymbol{v}))$ with different labels (i.e. $g^B(\boldsymbol{u}) \neq g^B(\boldsymbol{v})$) such that (23) holds. Let us calculate the output margin for the two inputs $\boldsymbol{u}$ and $\boldsymbol{v}$. It can be obtained that

$$|(f_0(\boldsymbol{u}) - f_1(\boldsymbol{u})) + (f_1(\boldsymbol{v}) - f_0(\boldsymbol{v}))| = |(f_0(\boldsymbol{u}) - f_0(\boldsymbol{v})) + (f_1(\boldsymbol{v}) - f_1(\boldsymbol{u}))|$$

$$\leq \sum_{k \in \{0,1\}} |f_k(\boldsymbol{v}) - f_k(\boldsymbol{u})| \leq \frac{2}{d}. \tag{24}$$

We will prove that it is impossible to classify both $\boldsymbol{u}$ and $\boldsymbol{v}$ with a margin greater than $1/d$. Otherwise, since $\boldsymbol{u}$ and $\boldsymbol{v}$ are in different classes, $f_0(\boldsymbol{u}) - f_1(\boldsymbol{u})$ and $f_1(\boldsymbol{v}) - f_0(\boldsymbol{v})$ will have the same sign, and both $|f_0(\boldsymbol{u}) - f_1(\boldsymbol{u})|$ and $|f_1(\boldsymbol{v}) - f_0(\boldsymbol{v})|$ will be greater than $1/d$, which contradicts (24). Therefore, the output margin must be no greater than $1/d$ for either input $\boldsymbol{u}$ or $\boldsymbol{v}$, implying that the certified $\ell_\infty$ robust radius cannot exceed $1/2d$ (based on Proposition 3.1) and concluding the proof. $\square$

**Remark B.7.** The symmetry assumption of the Boolean function $g^B$ may be relaxed, and the conclusion is still correct as long as there exists a set $\mathcal{T} \subset \{(\boldsymbol{u}, \boldsymbol{v}) : \boldsymbol{u}, \boldsymbol{v} \in \{0, 1\}^d\}$ satisfying that (i) $g^B(\boldsymbol{u}) \neq g^B(\boldsymbol{v}) \; \forall (\boldsymbol{u}, \boldsymbol{v}) \in \mathcal{T}$ and (ii) $\left\| \sum_{(\boldsymbol{u}, \boldsymbol{v}) \in \mathcal{T}} |\boldsymbol{u} - \boldsymbol{v}| \right\|_\infty = \mathcal{O}(|\mathcal{T}|/d)$.

We now prove that the bound of certified radius is in fact *tight*, in the sense that there exists a standard Lipschitz network with a suitable activation function that achieves a certified $\ell_\infty$ radius of $1/2d$.

**Proposition B.8.** *For any $d$-dimensional symmetric Boolean dataset, there exists a standard Lipschitz network that correctly classifies the whole dataset and achieves a certified $\ell_\infty$ robust radius of $1/2d$.*

*Proof.* Consider a $d$-dimensional Boolean Dataset $\mathcal{D} = \{(\boldsymbol{x}^{(i)}, y^{(i)}\}_{i=1}^n\}$ where $y^{(i)} = g^{\mathrm{B}}(\boldsymbol{x}^{(i)})$ for some symmetric Boolean function $g^{\mathrm{B}}$. Since the output of a symmetric Boolean function depends only on the number of ones in the input (Appendix A.2), we can equivalently express $g^{\mathrm{B}}$ by a Boolean vector $(g_0, \cdots, g_d) \in \{0,1\}^{d+1}$ with $g^{\mathrm{B}}(\boldsymbol{x}) = g_k$ if $\sum_i x_i = k$. Consider a special one-layer standard Lipschitz network $\boldsymbol{f} : \mathbb{R}^d \to \mathbb{R}^2$ with activation $\sigma$ defined as follows:

$$\boldsymbol{f}(\boldsymbol{x}) = (f_0(\boldsymbol{x}), f_1(\boldsymbol{x}))^{\mathrm{T}} = \left(\sigma\left(-\frac{1}{d}\sum_i x_i - \frac{1}{d}\right), \sigma\left(\frac{1}{d}\sum_i x_i + \frac{1}{d}\right)\right)^{\mathrm{T}}.$$

$\boldsymbol{f}$ is clearly 1-Lipschitz w.r.t. $\ell_\infty$-norm as long as $\sigma$ is 1-Lipschitz. If we choose a special $\sigma$ defined on the interval $[-1 - 1/d, 1 + 1/d]$ as

$$\sigma(x) = \begin{cases} \frac{(-1)^{g_0+1}}{2}x & \text{if } -1 \le dx \le 1, \\ \frac{\mathrm{sgn}(x)}{2d}(2g_{k-1} - 1 + 2(d|x| - k)(g_k - g_{k-1})) & \text{if } k < d|x| \le k+1, \quad k \in [d], \end{cases}$$

which is a piece-wise linear function, it is easy to see that $\sigma$ is continuous and 1-Lipschitz. Furthermore, we can prove that the classifier induced by $\boldsymbol{f}$ can correctly classify the dataset with a margin of $1/d$ for any data. Consider an input $\boldsymbol{x}$ with $\sum_i x_i = k$. The output of $\boldsymbol{f}(\boldsymbol{x})$ is thus $\left(\sigma(-\frac{k+1}{d}), \sigma(\frac{k+1}{d})\right)^{\mathrm{T}}$. Noting that $\sigma$ is an odd function, we have

$$f_1(\boldsymbol{x}) - f_0(\boldsymbol{x}) = 2\sigma\left(\frac{k+1}{d}\right) = \frac{2g_k - 1}{d} = \frac{1}{d}(2g^{\mathrm{B}}(\boldsymbol{x}) - 1) = \begin{cases} 1/d & \text{if } g^{\mathrm{B}}(\boldsymbol{x}) = 1, \\ -1/d & \text{if } g^{\mathrm{B}}(\boldsymbol{x}) = 0. \end{cases}$$

This proves that the margin is always $1/d$ for each data of $\mathcal{D}$, and we have concluded the proof. □

**Remark B.9.** Let us consider the sample complexity in the proof of Theorem 3.5. The proof constructs the Boolean dataset $\mathcal{D} = \{(\boldsymbol{z}, g^{\mathrm{B}}(\boldsymbol{z})) : \boldsymbol{z} \in \mathcal{S}_p \cup \mathcal{S}_q\}$, which has size $|\mathcal{D}| = \binom{d}{p} + \binom{d}{q}$. In the simple case when $g^{\mathrm{B}}$ is the logical AND/OR function, one has $|\mathcal{D}| = d + 1$, which means that a dataset of size $\mathcal{O}(d)$ is sufficient to yield the impossibility result. However, in the extreme case when $p = \lceil\frac{d}{2}\rceil$, the size of $\mathcal{D}$ will become exponential in $d$, which makes the theorem impractical. One may ask whether for all non-constant symmetric Boolean functions, such an impossibility result holds on some dataset with a size that grows linear in $d$. We will show it is indeed true if the bound of certified radius is relaxed by constant to $1/d$, which is formalized in the following corollary.

**Corollary B.10.** *For any non-constant symmetric Boolean function $g^{\mathrm{B}} : \{0,1\}^d \to \{0,1\}$, there exists a Boolean dataset of size no more than $d + 1$ with labels $y^{(i)} = g^{\mathrm{B}}(\boldsymbol{x}^{(i)})$, such that no standard Lipschitz network can achieve a certified $\ell_\infty$ robust radius larger than $1/d$ on the dataset.*

*Proof.* The proof is almost the same as the one in Theorem 3.5, except for a different construction of the set $\mathcal{T}$ than in (20). We use the same notations as before, such as the sets $\mathcal{S}_k, \mathcal{S}^+, \mathcal{S}^-, \mathcal{N}^+, \mathcal{N}^-$, and the integers $p, q$ with $p = q + 1$. We separately consider two cases:

- $p \ge \lceil d/2 \rceil$. In this case, pick any vector $\boldsymbol{u} \in \mathcal{S}_p$ and define

$$\mathcal{T} = \{(\boldsymbol{u}, \boldsymbol{v}) : \boldsymbol{v} \in \mathcal{S}_q, \|\boldsymbol{u} - \boldsymbol{v}\|_1 = 1\}. \tag{25}$$

- $p < \lceil d/2 \rceil$. In this case, pick any vector $\boldsymbol{v} \in \mathcal{S}_q$ and define

$$\mathcal{T} = \{(\boldsymbol{u}, \boldsymbol{v}) : \boldsymbol{u} \in \mathcal{S}_p, \|\boldsymbol{u} - \boldsymbol{v}\|_1 = 1\}. \tag{26}$$

In both bases, the set $\mathcal{T}$ satisfies the following three properties:

- $g^{\mathrm{B}}(\boldsymbol{u}) \ne g^{\mathrm{B}}(\boldsymbol{v})$ for all $(\boldsymbol{u}, \boldsymbol{v}) \in \mathcal{T}$;
- The size of the set can be bounded by $\lceil d/2 \rceil \le |\mathcal{T}| \le d$;
- $\left\|\sum_{(\boldsymbol{u}, \boldsymbol{v}) \in \mathcal{T}} |\boldsymbol{u} - \boldsymbol{v}|\right\|_\infty = 1 \le \frac{2|\mathcal{T}|}{d}$.

The remaining proof directly parallels the proof of Theorem 3.5. □

## B.4 Proof of Theorem 3.6

**Theorem B.11.** *Any standard Lipschitz network $f : \mathbb{R}^d \to \mathbb{R}$ cannot approximate the simple 1-Lipschitz function $\boldsymbol{x} \to x_{(k)}$ for arbitrary $k \in [d]$ on a bounded domain $\mathcal{K} = [0,1]^d$ if $d \geq 2$. Moreover, there exists a point $\widehat{\boldsymbol{x}} \in \mathcal{K}$, such that*

$$|f(\widehat{\boldsymbol{x}}) - \widehat{x}_{(k)}| \geq \frac{1}{2} - \frac{1}{2d}. \tag{27}$$

*Proof.* Consider the $k$-threshold function $g^{\mathrm{B},k}(\boldsymbol{x}) = \mathbb{I}(\sum_i x_i \geq k)$. Following the proof of Theorem 3.5, there exists a set $\mathcal{T} \subset \{(\boldsymbol{u}, \boldsymbol{v}) : \boldsymbol{u}, \boldsymbol{v} \in \{0,1\}^d\}$ satisfying the following two properties:

- $g^{\mathrm{B}}(\boldsymbol{u}) \neq g^{\mathrm{B}}(\boldsymbol{v})$ for all $(\boldsymbol{u}, \boldsymbol{v}) \in \mathcal{T}$;

- $\left\| \sum_{(\boldsymbol{u}, \boldsymbol{v}) \in \mathcal{T}} |\boldsymbol{u} - \boldsymbol{v}| \right\|_\infty = \frac{|\mathcal{T}|}{d}$.

By applying Corollary B.5 and noting that $f$ outputs a scalar, we obtain

$$\sum_{(\boldsymbol{u}, \boldsymbol{v}) \in \mathcal{T}} |f(\boldsymbol{u}) - f(\boldsymbol{v})| \leq \frac{|\mathcal{T}|}{d}. \tag{28}$$

By the Pigeon Hole principle, there must exist a pair of points $(\boldsymbol{u}, \boldsymbol{v}) \in \mathcal{T}$, such that

$$|f(\boldsymbol{u}) - f(\boldsymbol{v})| \leq \frac{1}{d}. \tag{29}$$

However, since $g^{\mathrm{B},k}(\boldsymbol{u}) \neq g^{\mathrm{B},k}(\boldsymbol{v})$, we have $\left| g^{\mathrm{B},k}(\boldsymbol{u}) - g^{\mathrm{B},k}(\boldsymbol{v}) \right| = 1$. Therefore

$$\left| f(\boldsymbol{v}) - g^{\mathrm{B},k}(\boldsymbol{v}) \right| + \left| f(\boldsymbol{u}) - g^{\mathrm{B},k}(\boldsymbol{u}) \right| \geq \left| f(\boldsymbol{v}) - f(\boldsymbol{u}) + g^{\mathrm{B},k}(\boldsymbol{u}) - g^{\mathrm{B},k}(\boldsymbol{v}) \right| \tag{30}$$

$$\geq \left| g^{\mathrm{B},k}(\boldsymbol{u}) - g^{\mathrm{B},k}(\boldsymbol{v}) \right| - |f(\boldsymbol{v}) - f(\boldsymbol{u})| \tag{31}$$

$$\geq 1 - \frac{1}{d} \tag{32}$$

where (30) and (31) are based on the triangular inequality. Therefore, either $\left| f(\boldsymbol{v}) - g^{\mathrm{B},k}(\boldsymbol{v}) \right|$ or $\left| f(\boldsymbol{u}) - g^{\mathrm{B},k}(\boldsymbol{u}) \right|$ must be no smaller than $\frac{1}{2} - \frac{1}{2d}$. Proof finishes by noting that $g^{\mathrm{B},k}(\boldsymbol{x}) = x_{(k)}$ for any Boolean input $\boldsymbol{x} \in \{0,1\}^d$. $\square$

**Remark B.12.** The bound of approximation error $\frac{1}{2} - \frac{1}{2d}$ is *tight*. Indeed, it is easy to prove that the linear function $f(\boldsymbol{x}) = \frac{1}{2} + \frac{1}{d}(\sum_i x_i - k + \frac{1}{2})$ can satisfy the bound, i.e. $|f(\boldsymbol{x}) - x_{(k)}| \leq \frac{1}{2} - \frac{1}{2d}$ for all $\boldsymbol{x} \in \mathcal{K}$.

**Corollary B.13.** *For any standard Lipschitz network $f : \mathbb{R}^d \to \mathbb{R}$ and any $k \in [d]$ where $d \geq 2$, there exists a hypercube $\mathcal{B}_\infty^r(\widehat{\boldsymbol{x}}) := \{\boldsymbol{x} : \|\boldsymbol{x} - \widehat{\boldsymbol{x}}\|_\infty \leq r\} \subset [0,1]^d$ satisfying $r \geq 1/20$, such that for all $\tilde{\boldsymbol{x}} \in \mathcal{B}_\infty^r(\widehat{\boldsymbol{x}})$,*

$$|f(\tilde{\boldsymbol{x}}) - \tilde{x}_{(k)}| \geq \frac{1}{20}. \tag{33}$$

*Proof.* Let $\boldsymbol{u} \in \{0,1\}^d$ be a point satisfying (4) in Theorem 3.6, i.e. $|f(\boldsymbol{u}) - u_{(k)}| \geq 1/4$. Consider the hypercube $\mathcal{B}_\infty^{1/10}(\boldsymbol{u})$ with a length of $1/5$. For any $\boldsymbol{x} \in \mathcal{B}_\infty^{1/10}(\boldsymbol{u})$, $\|\boldsymbol{x} - \boldsymbol{u}\|_\infty \leq 1/10$. Therefore,

$$|f(\boldsymbol{x}) - x_{(k)}| = |f(\boldsymbol{x}) - f(\boldsymbol{u}) + f(\boldsymbol{u}) - u_{(k)} + u_{(k)} - x_{(k)}| \tag{34}$$

$$\geq |f(\boldsymbol{u}) - u_{(k)}| - |f(\boldsymbol{u}) - f(\boldsymbol{x})| - |u_{(k)} - x_{(k)}| \tag{35}$$

$$\geq \frac{1}{4} - \|\boldsymbol{x} - \boldsymbol{u}\|_\infty - \|\boldsymbol{x} - \boldsymbol{u}\|_\infty \geq \frac{1}{20} \tag{36}$$

where (35) uses the triangular inequality and (36) is based on the fact that $f$ and $x_{(k)}$ are both 1-Lipschitz w.r.t. $\ell_\infty$-norm. Finally, consider the intersection $\mathcal{B}_\infty^{1/10}(\boldsymbol{u}) \cup [0,1]^d$, which must contain a hypercube with a raius no smaller than $1/20$. We have thus finished the proof. $\square$

## B.5    Proof of Theorem 3.8

To prove Theorem 3.8, we need the following lemma:

**Lemma B.14.** *A single $\ell_\infty$-distance neuron with suitable parameters can exactly represent any literal disjunction defined in Definition A.1. Formally, for any Boolean function*

$$g^B(\boldsymbol{x}) = x_{i_1} \vee \cdots \vee x_{i_r} \vee \neg x_{j_1} \vee \cdots \vee \neg x_{j_s},$$

*where $1 \le i_1 \le \cdots \le i_r \le d, 1 \le j_1 \le \cdots \le j_s \le d \ (r + s \ge 1)$ are different indices, there exists an $\ell_\infty$-distance neuron $f(\boldsymbol{x}) = \|\boldsymbol{x} - \boldsymbol{w}\|_\infty + b$, such that $f(\boldsymbol{x}) = g^B(\boldsymbol{x})$ for all $\boldsymbol{x} \in \{0, 1\}^d$.*

*Proof.* Consider an $\ell_\infty$-distance neuron $f$ with parameters $\boldsymbol{w}$ and $b$ defined as

$$w_{i_k} = -1, \ k \in [r], \quad w_{j_k} = 2, \ k \in [s], \quad \text{and } w_k = \tfrac{1}{2} \text{ for other } k,$$

$$b = -1.$$

It is easy to see that

- When $x_{i_k} = 0 \ \forall k \in [r]$ and $x_{j_k} = 1 \ \forall k \in [s]$,

$$
\begin{aligned}
f(\boldsymbol{x}) &= \|\boldsymbol{x} - \boldsymbol{w}\|_\infty + b \\
&= \max\left( \max_{k \in [r]} |x_{i_k} - w_{i_k}|, \max_{k \in [s]} |x_{j_k} - w_{j_k}|, \max_{k \in [d]/\{i_1, \cdots, i_r, j_1, \cdots, j_s\}} |x_k - w_k| \right) - 1 \\
&= \max\left( 1, 1, \max_{k \in [d]/\{i_1, \cdots, i_r, j_1, \cdots, j_s\}} \left| x_k - \frac{1}{2} \right| \right) - 1 \\
&= 1 - 1 = 0 = g^B(\boldsymbol{x}).
\end{aligned}
$$

- When $x_{i_k} = 1$ for some $k \in [r]$, or $x_{j_k} = 0$ for some $k \in [s]$, a similar calculation yields $f(\boldsymbol{x}) = 2 - 1 = 1 = g^B(\boldsymbol{x})$.

This concludes the proof. $\qquad\qquad\square$

**Theorem B.15.** *A two-layer $\ell_\infty$-distance net can exactly represent any discrete Boolean function as well as any continuous order-statistic function on a compact domain.*

*Proof.* Based on Fact A.2, any satisfiable Boolean function can be written as a DNF $g^B(\boldsymbol{x}) = \vee_{i=1}^m g_m^B(\boldsymbol{x})$ where $g_m^B(\boldsymbol{x})$ are literal conjunctions. By the De Morgan's law, any literal conjunction can be equivalently written as the negative of a literal disjunction, i.e.

$$x_{i_1} \wedge \cdots \wedge x_{i_r} \wedge \neg x_{j_1} \wedge \cdots \wedge \neg x_{j_s} = \neg(\neg x_{i_1} \vee \cdots \vee \neg x_{i_r} \vee x_{j_1} \vee \cdots \vee x_{j_s}). \tag{37}$$

Therefore, we can write $g^B(\boldsymbol{x}) = \vee_{i=1}^m \neg \hat{g}_i^B(\boldsymbol{x})$ where $\hat{g}_i^B(\boldsymbol{x}) := \neg g_i^B(\boldsymbol{x})$ are all disjunctive literals. Due to Lemma B.14, each $\hat{g}_i^B(\boldsymbol{x})$ can be represented by an $\ell_\infty$-distance neuron that takes $\boldsymbol{x}$ as input, and $g^B(\boldsymbol{x})$ can be represented by an $\ell_\infty$-distance neuron that takes the vector $(\hat{g}_1^B(\boldsymbol{x}), \cdots, \hat{g}_m^B(\boldsymbol{x}))^{\mathrm{T}}$ as input. This proves that a two-layer $\ell_\infty$-distance net suffices to represent any satisfiable discrete Boolean function. Finally, for the unsatisfiable case where the Boolean function simply outputs zero, we can construct the following two-layer $\ell_\infty$-distance net:

$$f(\boldsymbol{x}) = \left\| (\|\boldsymbol{x}\|_\infty, \|\boldsymbol{x} - \boldsymbol{1}\|_\infty)^{\mathrm{T}} \right\|_\infty - 1 \tag{38}$$

where $\boldsymbol{1}$ is the all-one vector. It follows that $(\|\boldsymbol{x}\|, \|\boldsymbol{x} - \boldsymbol{1}\|)^{\mathrm{T}}$ must be one of the following three cases: $(0, 1)^{\mathrm{T}}$ (when $\boldsymbol{x} = \boldsymbol{0}$), $(1, 0)^{\mathrm{T}}$ (when $\boldsymbol{x} = \boldsymbol{1}$), or $(1, 1)^{\mathrm{T}}$ (when $\boldsymbol{x} \ne \boldsymbol{0}$ and $\boldsymbol{x} \ne \boldsymbol{1}$). Therefore $f(\boldsymbol{x}) = 0$ always hold $\forall \boldsymbol{x} \in \{0, 1\}^d$. This concludes the first part of the proof.

We next turn to the case of representing order statistics. We use the key observation that any order statistic function can be written as a nested max-min function:

$$x_{(k)} = \max_{S \subset [d], |S| = k} \min_{i \in S} x_i. \tag{39}$$

This can be further equivalently written as a nested max-max function:

$$x_{(k)} = \max_{S \subset [d], |S| = k} -f^S(\boldsymbol{x}) \tag{40}$$

where $f^S(\boldsymbol{x}) = \max_{i \in S} -x_i$. This basically concludes the proof, as the max function over a set of elements $\{x_{i_1}, \cdots, x_{i_r}, -x_{j_1}, \cdots, -x_{j_s}\}$ can be exactly represented by an $\ell_\infty$-distance neuron following the construction in the proof of Lemma B.14. Therefore, each $f^S(\boldsymbol{x})$ can be represented by an $\ell_\infty$-distance neuron with input $\boldsymbol{x}$, and $x_{(k)}$ can be represented by an $\ell_\infty$-distance neuron that takes all $f^S(\boldsymbol{x})$ as input. As a result, a two-layer $\ell_\infty$-distance net suffices to represent the order statistics. $\qquad\square$

**Remark B.16.** (On the representation efficiency of $\ell_\infty$-distance net) Based on the above proof, the width of the two-layer $\ell_\infty$-distance net in representing a Boolean function depends on the minimum number of conjunctions in its DNF, which may be exponentially large. Indeed, it is known that no DNF with a polynomial number of conjunctions can approximate the majority function defined in Section A.2 [44]. Similarly, in the continuous setting, the construction in (39, 40) needs a network of size $\binom{d}{\lfloor d/2 \rfloor}$ in order to represent the median function $k = \lfloor d/2 \rfloor$, which is exponential in $d$. However, when using a *multi-layer* $\ell_\infty$-distance net, the representation efficiency can be improved dramatically. For example, using the idea of sorting, it is easy to construct an $\ell_\infty$-distance net with polynomial size that represents the median function.

### B.6 Proof of Theorem 3.9

The proof is almost the same as the one in Theorem 3.6. The major difference is to replace Lemma B.4 by the following lemma:

**Lemma B.17.** *Let* $f(\boldsymbol{x}) = \sigma(\boldsymbol{w}_1^{\mathrm{T}}\boldsymbol{x} + b_1, \boldsymbol{w}_2^{\mathrm{T}}\boldsymbol{x} + b_2)$ *be a d-dimensional function where* $\|\boldsymbol{w}_i\|_1 \leq 1$ $\forall i \in [2]$ *and* $\sigma$ *is a two-dimensional 1-Lipschitz function w.r.t.* $\ell_\infty$*-norm. Let* $\{(\boldsymbol{u}^{(i)}, \boldsymbol{v}^{(i)}) \in \mathbb{R}^d \times \mathbb{R}^d\}_{i=1}^n$ *be n pairs of inputs. Then*

$$\sum_{i=1}^n \left| f(\boldsymbol{u}^{(i)}) - f(\boldsymbol{v}^{(i)}) \right| \leq 2 \left\| \sum_{i=1}^n \left| \boldsymbol{u}^{(i)} - \boldsymbol{v}^{(i)} \right| \right\|_\infty.$$

*Proof.* Using the Lipschitz property of $\sigma$, we have

$$\left| f(\boldsymbol{u}^{(i)}) - f(\boldsymbol{v}^{(i)}) \right| = \left| \sigma(\boldsymbol{w}_1^{\mathrm{T}}\boldsymbol{u}^{(i)} + b_1, \boldsymbol{w}_2^{\mathrm{T}}\boldsymbol{u}^{(i)} + b_2) - \sigma(\boldsymbol{w}_1^{\mathrm{T}}\boldsymbol{v}^{(i)} + b_1, \boldsymbol{w}_2^{\mathrm{T}}\boldsymbol{v}^{(i)} + b_2) \right|$$

$$\leq \max\left( \left| \boldsymbol{w}_1^{\mathrm{T}}(\boldsymbol{u}^{(i)} - \boldsymbol{v}^{(i)}) \right|, \left| \boldsymbol{w}_2^{\mathrm{T}}(\boldsymbol{u}^{(i)} - \boldsymbol{v}^{(i)}) \right| \right) \tag{41}$$

$$\leq \max\left( |\boldsymbol{w}_1|^{\mathrm{T}} \left| \boldsymbol{u}^{(i)} - \boldsymbol{v}^{(i)} \right|, |\boldsymbol{w}_2|^{\mathrm{T}} \left| \boldsymbol{u}^{(i)} - \boldsymbol{v}^{(i)} \right| \right) \tag{42}$$

$$\leq (|\boldsymbol{w}_1| + |\boldsymbol{w}_2|)^{\mathrm{T}} \left| \boldsymbol{u}^{(i)} - \boldsymbol{v}^{(i)} \right| \tag{43}$$

Therefore

$$\sum_{i=1}^n \left| f(\boldsymbol{u}^{(i)}) - f(\boldsymbol{v}^{(i)}) \right| \leq (|\boldsymbol{w}_1| + |\boldsymbol{w}_2|)^{\mathrm{T}} \sum_{i=1}^n \left| \boldsymbol{u}^{(i)} - \boldsymbol{v}^{(i)} \right|$$

$$\leq \| |\boldsymbol{w}_1| + |\boldsymbol{w}_2| \|_1 \left\| \sum_{i=1}^n \left| \boldsymbol{u}^{(i)} - \boldsymbol{v}^{(i)} \right| \right\|_\infty, \tag{44}$$

where the last inequality in (44) follows by using the Hölder's inequality. Finally, observing that $\| |\boldsymbol{w}_1| + |\boldsymbol{w}_2| \|_1 \leq \|\boldsymbol{w}_1\|_1 + \|\boldsymbol{w}_2\|_1 \leq 2$, we have arrived at the conclusion. $\qquad\square$

**Theorem B.18.** *Any M-layer MaxMin network* $f : \mathbb{R}^d \to \mathbb{R}$ *cannot approximate any k-th order statistic function on a bounded domain* $\mathcal{K} = [0,1]^d$ *if* $M \leq \lceil \log_2 d \rceil$*. Moreover, there exists a point* $\widehat{\boldsymbol{x}} \in \mathcal{K}$*, such that*

$$|f(\widehat{\boldsymbol{x}}) - \widehat{x}_{(k)}| \geq \frac{1}{2} - \frac{2^{M-2}}{d} \geq \frac{1}{4} \quad \textit{if } M \leq \lfloor \log_2 d \rfloor. \tag{45}$$

*Proof.* For an $M$-layer MaxMin network, there are $M - 1$ layers with the MaxMin activation. Using the same proof as in Corollary B.5, we have that for any set $\{(\boldsymbol{u}^{(i)}, \boldsymbol{v}^{(i)}) \in \mathbb{R}^d \times \mathbb{R}^d\}_{i=1}^n$,

$$\left\| \sum_{i=1}^n \left| f(\boldsymbol{u}^{(i)}) - f(\boldsymbol{v}^{(i)}) \right| \right\|_\infty \leq 2^{M-1} \left\| \sum_{i=1}^n \left| \boldsymbol{u}^{(i)} - \boldsymbol{v}^{(i)} \right| \right\|_\infty.$$

The remaining proof is the same as the proof of Theorem 3.6, with (28) replaced by

$$\left\|\sum_{(\boldsymbol{u},\boldsymbol{v})\in\mathcal{T}}|f(\boldsymbol{u})-f(\boldsymbol{v})|\right\|_\infty \leq \frac{2^{M-1}|\mathcal{T}|}{d}. \tag{46}$$

$\square$

We can also give an upper bound of the minimum depth required. The upper bound in the following theorem shows that the lower bound of the depth in Theorem 3.9 is tight up to a constant factor.

**Theorem B.19.** *An $M$-layer MaxMin network $f : \mathbb{R}^d \to \mathbb{R}$ with dpeth $M = \mathcal{O}(\log_2 d)$ can exactly represent any $d$-dimensional order statistic function on a bounded domain $\mathcal{K}$.*

*Proof.* Our proof leverages the work of Ajtai et al. [1], who constructed a so-called *sorting network* which has depth $\mathcal{O}(\log_2 d)$. The basic component of a sorting network is called the *comparator*. Each comparator connects two inputs and swaps the values if and only if the value of the first input is greater than the second. The comparators have a hierarchical structure (arranged layer by layer), and the execution is parallel for all comparators in the same layer. To avoid conflicts, no input can be simultaneously connected by two comparators in the same layer. We will show that given a sorting network (denoted as sort), one can construct an equivalent MaxMin network $\boldsymbol{f} : \mathbb{R}^d \to \mathbb{R}^d$ that has the same depth, such that for some fixed permutation $\pi \in S_d$, $\mathrm{sort}(\boldsymbol{x}) = (f_{\pi_1}(\boldsymbol{x}), \cdots, f_{\pi_d}(\boldsymbol{x}))^\mathrm{T}$ holds for all inputs $\boldsymbol{x} \in \mathcal{K}$.

Intuitively, each pair of MaxMin neurons can perform the same operation as a comparator. This is simply because

$$(\max(x_i, x_j), \min(x_i, x_j)) = \mathrm{MaxMin}(\boldsymbol{e}_i^\mathrm{T}\boldsymbol{x}, \boldsymbol{e}_j^\mathrm{T}\boldsymbol{x}).$$

For those inputs that are not connected to any comparator, one can construct an identity function to propagate the input without change, by using a pair of MaxMin neurons with properly chosen bias values so that one neuron is always larger than the other:

$$(x_i, b) = \mathrm{MaxMin}(\boldsymbol{e}_i^\mathrm{T}\boldsymbol{x}, b) \quad \text{where } b \to -\infty.$$

Note that we waste the neuron that always outputs $b$. Eventually, we have constructed a MaxMin network that can output the sorting result (differing by up to a permutation) within $\mathcal{O}(\log_2 d)$ depth. Therefore, by adding one more layer that extracts the $k$-th order statistic, the network then exactly represents the $k$-th order statistic function. $\square$

## B.7 Proof of Theorem 3.10

**Proposition B.20.** *An $M$-layer MaxMin network can exactly represent any (discrete) $d$-dimensional Boolean function if $M > d + \lceil \log_2 d \rceil$.*

*Proof.* First, it is easy to see that a pair of MaxMin neurons can represent any two-dimensional literal conjunctions/disjunctions, i.e. $r_i \wedge r_j$ and $r_i \vee x_j$, where $r_i$ is either $x_i$ or $\neg x_i$. For example, $\max(x_i, x_j) = x_i \vee x_j$, $\min(x_i, x_j) = x_i \wedge x_j$, $\min(1 - x_i, 1 - x_j) = \neg x_i \wedge \neg x_j$, etc. Next, using reduction, any $d$-dimensional literal conjunction/disjunction can be represented by a $(\lceil \log_2 d \rceil + 1)$-layer MaxMin network (here $+1$ means that the last layer does not have MaxMin activation). Finally, recall that any Boolean function can be written in its DNF (Appendix A.1) and the number of disjunctions cannot exceed $2^d$ for any $d$-dimensional Boolean function, we have proved that a MaxMin network with depth $M = d + \lceil \log_2 d \rceil + 1$ can represent any Boolean function. $\square$

In the following, we will prove that the bound of $M = \mathcal{O}(d)$ is *tight* in order to interpolate $d$-dimensional Boolean functions. To do this, we first define several notations.

**Definition B.21.** Let $\mathcal{D} = \{(\boldsymbol{x}^{(i)}, y^{(i)}) \in \mathbb{R}^d \times \{0,1\}\}_{i=1}^n$ be a dataset with real vectors as inputs and Boolean labels, and the inputs have a bounded diameter of 1 w.r.t. $\ell_\infty$-norm, i.e. $\|\boldsymbol{x}^{(j)} - \boldsymbol{x}^{(k)}\|_\infty \leq 1$ $\forall j, k \in [n]$.

- Denote $l_i(\mathcal{D}) = \min_{j \in [n]} x_i^{(j)}$ and $u_i(\mathcal{D}) = \max_{j \in [n]} x_i^{(j)}$. We say the $i$-th input element is *useful* if $u_i(\mathcal{D}) - l_i(\mathcal{D}) = 1$; Correspondingly, the $i$-th input element is *useless* if $u_i(\mathcal{D}) - l_i(\mathcal{D}) < 1$.

- Given an index set $\mathcal{I} \subset [d]$, define the $\mathcal{I}$-*masked* dataset $\mathcal{D}_{\mathcal{I}}^{\mathrm{M}}$ to be the dataset obtained by zeroing out the $i$-th input element of each sample for each $i \in \mathcal{I}$. Specifically, $\mathcal{D}_{\mathcal{I}}^{\mathrm{M}} := \{(\tilde{\boldsymbol{x}}^{(j)}, y^{(j)})\}_{j=1}^{n}$ where

$$\tilde{x}_i^{(j)} = \begin{cases} x_i^{(j)} & i \notin \mathcal{I}, \\ 0 & i \in \mathcal{I}. \end{cases}$$

- Given an index set $\mathcal{I} \subset [d]$, define the $\mathcal{I}$-*thresholded* dataset $\mathcal{D}_{\mathcal{I}}^{\mathrm{T}}$ to be any dataset satisfying that, for each $i \in \mathcal{I}$, change the $i$-th input element of all samples to the extreme values, either the minimum $l_i(\mathcal{D})$ or the maximum $u_i(\mathcal{D})$, but keep the original minimum and maximum elements unchanged. Specifically, $\mathcal{D}_{\mathcal{I}}^{\mathrm{T}} := \{(\tilde{\boldsymbol{x}}^{(j)}, y^{(j)})\}_{j=1}^{n}$ where

$$\begin{cases} \tilde{x}_i^{(j)} = x_i^{(j)} & \text{if } i \notin \mathcal{I}, \\ \tilde{x}_i^{(j)} = x_i^{(j)} & \text{if } i \in \mathcal{I},\ x_i^{(j)} = l_i(\mathcal{D}) \text{ or } x_i^{(i)} = u_i(\mathcal{D}), \\ \tilde{x}_i^{(j)} \in \{l_i(\mathcal{D}), u_i(\mathcal{D})\} & \text{if } i \in \mathcal{I},\ l_i(\mathcal{D}) < x_i^{(j)} < u_i(\mathcal{D}). \end{cases}$$

  Clearly, the $\mathcal{I}$-thresholded dataset is not unique.

- We say a new dataset $\mathcal{D}^{\mathrm{F}}$ is *formatted* from dataset $\mathcal{D}$, if $\mathcal{D}^{\mathrm{F}}$ can be obtained by *masking* all the *useless* elements in $\mathcal{D}$ and *thresholding* all *useful* elements in $\mathcal{D}$.

**Lemma B.22.** *Let $\mathcal{D} = \{(\boldsymbol{x}^{(i)}, y^{(i)}) \in \mathbb{R}^d \times \{0,1\}\}_{i=1}^{n}$ be a finite dataset with a bounded diameter of 1 w.r.t. $\ell_\infty$-norm. If $\mathcal{D}$ can be interpolated by a MaxMin network, then any dataset $\mathcal{D}^F$ formatted from $\mathcal{D}$ can be interpolated by a MaxMin network with the same structure (i.e. depth and width).*

*Proof.* Our proof is based on induction over the network depth. For the base case, consider the network that performs the identify mapping: $f(x) = x$ where $x \in \mathbb{R}$ is a scalar. If $f$ interpolates the dataset $\mathcal{D}$, then $d = 1$ and $\boldsymbol{x}^{(i)} \in \{0,1\}\ \forall i \in [n]$. Therefore, $\mathcal{D}^{\mathrm{F}} = \mathcal{D}$ and $f$ interpolates $\mathcal{D}^{\mathrm{F}}$.

Now assume the conclusion holds for a MaxMin network $f : \mathbb{R}^{d_f} \to \mathbb{R}$. We would like to prove that it also holds for the MaxMin network $f \circ \boldsymbol{h}$ where $\boldsymbol{h} : \mathbb{R}^{d_h} \to \mathbb{R}^{d_f}$ is either the affine layer or the MaxMin activation layer. This will yield the conclusion, as any MaxMin network can be constructed by starting with the identity function and stacking affine layers and MaxMin activations (from the last layer to the first).

**Case 1**: $\boldsymbol{h} : \mathbb{R}^{d_h} \to \mathbb{R}^{d_f}$ is an affine layer. We denote $h_i(\boldsymbol{x}) = \boldsymbol{w}_i^{\mathrm{T}} \boldsymbol{x} + b_i$ to be the $i$-th element of $\boldsymbol{h}$ with weight $\boldsymbol{w}_i$ and bias $b_i$. Assume $f \circ \boldsymbol{h}$ can interpolate dataset $\mathcal{D}$. We will construct a new MaxMin network $\tilde{f} \circ \tilde{\boldsymbol{h}}$ such that $\tilde{f}$ has the same topology as $f$, $\tilde{\boldsymbol{h}} : \mathbb{R}^{d_h} \to \mathbb{R}^{d_f}$ is an affine layer with weights $\tilde{\boldsymbol{w}}_i$ and biases $\tilde{b}_i$ $(i \in [d_f])$, and $\tilde{f} \circ \tilde{\boldsymbol{h}}$ interpolates $\mathcal{D}^{\mathrm{F}}$. Note that $\|\boldsymbol{w}_i\|_1 \le 1$ and $\|\tilde{\boldsymbol{w}}_i\|_1 \le 1$ must hold $(\forall i \in [d_f])$.

Given dataset $\mathcal{D}$, denote $\boldsymbol{z}^{(j)} = \boldsymbol{h}(\boldsymbol{x}^{(j)})\ \forall j \in [n]$. Applying $\boldsymbol{h}$ to $\mathcal{D}$ yields another dataset denoted as $\boldsymbol{h}(\mathcal{D}) := \{(\boldsymbol{z}^{(j)}, y^{(j)})\}_{j=1}^{n}$, which also has a bounded diameter of 1 due to the 1-Lipschitz property of $\boldsymbol{h}$. By the assumption, $f$ interpolates $\boldsymbol{h}(\mathcal{D})$. Therefore, one can use induction to prove that there exists $\tilde{f}$ that interpolates $(\boldsymbol{h}(\mathcal{D}))^{\mathrm{F}}$ where $(\boldsymbol{h}(\mathcal{D}))^{\mathrm{F}}$ is any dataset formatted from $\boldsymbol{h}(\mathcal{D})$.

We now construct the weights and biases of $\tilde{\boldsymbol{h}}$ as follows:

- If the $i$-th element of $\boldsymbol{h}(\mathcal{D})$ is useless (i.e. $u_i(\boldsymbol{h}(\mathcal{D})) - l_i(\boldsymbol{h}(\mathcal{D})) < 1$), then set $\tilde{\boldsymbol{w}}_i = \boldsymbol{0}$ and $\tilde{b}_i = 0$;

- If the $i$-th element of $\boldsymbol{h}(\mathcal{D})$ is useful, then there exists two samples $\boldsymbol{x}^{(j)}$ and $\boldsymbol{x}^{(k)}$ satisfying $|z_i^{(j)} - z_i^{(k)}| = 1$. Therefore, $|\boldsymbol{w}_i^{\mathrm{T}}(\boldsymbol{x}^{(j)} - \boldsymbol{x}^{(k)})| = 1$. Note that $\|\boldsymbol{w}_i\|_1 \le 1$ and $\|\boldsymbol{x}^{(j)} - \boldsymbol{x}^{(k)}\|_\infty \le 1$. Thus by Hölder's inequality, it must be $\|\boldsymbol{x}^{(j)} - \boldsymbol{x}^{(k)}\|_\infty = 1$ and $[\boldsymbol{w}_i]_s = 0$ for all $s$ satisfying $|x_s^{(j)} - x_s^{(k)}| < 1$. Pick any $s$ with $[\boldsymbol{w}_i]_s \ne 0$ (thus $|x_s^{(j)} - x_s^{(k)}| = 1$). Choose $\tilde{\boldsymbol{w}}_i = \mathrm{sgn}([\boldsymbol{w}_i]_s)\boldsymbol{e}_s$ (the unit vector with the $s$-th element being 1 or -1 depending on the sign of $[\boldsymbol{w}_i]_s$) and $\tilde{b}_i = (\boldsymbol{w}_i - \tilde{\boldsymbol{w}}_i)^{\mathrm{T}}\boldsymbol{x}^{(j)} + b_i$.

Similarly, denote $\tilde{\boldsymbol{z}}^{(j)} = \tilde{\boldsymbol{h}}(\boldsymbol{x}^{(j)})\ \forall j \in [n]$ and the the corresponding dataset $\tilde{\boldsymbol{h}}(\mathcal{D}) := \{(\tilde{\boldsymbol{z}}^{(j)}, y^{(j)})\}_{j=1}^{n}$ (which also has a bounded diameter of 1 due to the 1-Lipschitz property of $\tilde{\boldsymbol{h}}$). First observe that if the $i$-th element is useless in $\boldsymbol{h}(\mathcal{D})$, then it is also useless in $\tilde{\boldsymbol{h}}(\mathcal{D})$. We now prove that

the converse also holds: if the $i$-th element is useful in $\boldsymbol{h}(\mathcal{D})$, then it is also useful in $\tilde{\boldsymbol{h}}(\mathcal{D})$. Otherwise, there exists $\boldsymbol{x}^{(j)}$ and $\boldsymbol{x}^{(k)}$ such that $|\boldsymbol{w}_i^{\mathrm{T}}(\boldsymbol{x}^{(j)} - \boldsymbol{x}^{(k)})| = 1$ but $|\tilde{\boldsymbol{w}}_i^{\mathrm{T}}(\boldsymbol{x}^{(j)} - \boldsymbol{x}^{(k)})| < 1$. However, in this case, by construction $\tilde{\boldsymbol{w}}_i = \pm\boldsymbol{e}_s$ is the unit vector for some $s$, implying that $|x_s^{(j)} - x_s^{(k)}| < 1$. Since $[\boldsymbol{w}_i]_s \neq 0$, Hölder's inequality implies that $|\boldsymbol{w}_i^{\mathrm{T}}(\boldsymbol{x}^{(j)} - \boldsymbol{x}^{(k)})| < 1$, a contradiction.

Therefore, the $i$-th element is useful in $\boldsymbol{h}(\mathcal{D})$ if and only if it is useful in $\tilde{\boldsymbol{h}}(\mathcal{D})$. Now assume that the $i$-th element is both useful in $\boldsymbol{h}(\mathcal{D})$ and $\tilde{\boldsymbol{h}}(\mathcal{D})$. We next prove that $l_i(\boldsymbol{h}(\mathcal{D})) = l_i(\tilde{\boldsymbol{h}}(\mathcal{D}))$ and $u_i(\boldsymbol{h}(\mathcal{D})) = u_i(\tilde{\boldsymbol{h}}(\mathcal{D}))$. By the assignment of $\tilde{b}_i$, there exists an $\boldsymbol{x}^{(j)}$ such that $\boldsymbol{w}_i^{\mathrm{T}}\boldsymbol{x}^{(j)} + b_i = \tilde{\boldsymbol{w}}_i^{\mathrm{T}}\boldsymbol{x}^{(j)} + \tilde{b}_i$, namely $z_i^{(j)} = \tilde{z}_i^{(j)}$. Since $z_i^{(j)}$ must at the extreme value, without loss of generality, assume $z_i^{(j)} = l_i(\boldsymbol{h}(\mathcal{D}))$. Based on the above paragraph, for any $\boldsymbol{x}^{(k)}$ such that $|\boldsymbol{w}_i^{\mathrm{T}}(\boldsymbol{x}^{(j)} - \boldsymbol{x}^{(k)})| = 1$, $|\tilde{\boldsymbol{w}}_i^{\mathrm{T}}(\boldsymbol{x}^{(j)} - \boldsymbol{x}^{(k)})| = 1$. Thus for any $z_i^{(k)} = u_i(\boldsymbol{h}(\mathcal{D}))$, $|z_i^{(j)} - z_i^{(k)}| = |\tilde{z}_i^{(j)} - \tilde{z}_i^{(k)}|$. Also, $|z_i^{(k)} - \tilde{z}_i^{(k)}| < 2$ because by construction $\|\tilde{\boldsymbol{w}} - \boldsymbol{w}\|_1 < 2$. This proves that $\tilde{z}_i^{(j)} = l_i(\tilde{\boldsymbol{h}}(\mathcal{D}))$ and $\tilde{z}_i^{(k)} = u_i(\tilde{\boldsymbol{h}}(\mathcal{D}))$, thus $l_i(\boldsymbol{h}(\mathcal{D})) = l_i(\tilde{\boldsymbol{h}}(\mathcal{D}))$ and $u_i(\boldsymbol{h}(\mathcal{D})) = u_i(\tilde{\boldsymbol{h}}(\mathcal{D}))$. Note that we have actually proved a stronger result: given samples $\boldsymbol{z}^{(k)}$ and $\tilde{\boldsymbol{z}}^{(k)}$, $z_i^{(k)} = l_i(\boldsymbol{h}(\mathcal{D})) \iff \tilde{z}_i^{(k)} = l_i(\tilde{\boldsymbol{h}}(\mathcal{D}))$, and $z_i^{(k)} = u_i(\boldsymbol{h}(\mathcal{D})) \iff \tilde{z}_i^{(k)} = u_i(\tilde{\boldsymbol{h}}(\mathcal{D}))$.

Combining the above results, one can conclude that for a dataset $(\boldsymbol{h}(\mathcal{D}))^{\mathrm{F}}$ formatted from $\boldsymbol{h}(\mathcal{D})$, it is also formatted from $\tilde{\boldsymbol{h}}(\mathcal{D})$. $\hfill$ (*)

Now let $\tilde{\boldsymbol{h}}(\mathcal{D}^{\mathrm{F}})$ be the dataset obtained by applying $\tilde{\boldsymbol{h}}$ to the formatted dataset $\mathcal{D}^{\mathrm{F}}$. Note that the weights in $\tilde{\boldsymbol{h}}$ are special in that all $\tilde{\boldsymbol{w}}_i$ are either unit or zero vectors by construction. We will use this property to prove that $\tilde{\boldsymbol{h}}(\mathcal{D}^{\mathrm{F}})$ is just the dataset formatted from $\tilde{\boldsymbol{h}}(\mathcal{D})$. The key observation is that

- If $[\tilde{\boldsymbol{w}}_i]_s = 0$, then masking/thresholding the $s$-th element in $\mathcal{D}$ does not change the network output after applying $\tilde{h}_i$.

- If $[\tilde{\boldsymbol{w}}_i]_s \neq 0$, then the $s$-th element must be useful in $\mathcal{D}$ (bacause the $i$-th element is useful in $\boldsymbol{h}(\mathcal{D})$ and $[\boldsymbol{w}_i]_s \neq 0$). Since $\tilde{\boldsymbol{w}}_i$ is unit ($[\tilde{\boldsymbol{w}}_i]_s = \pm 1$), it is easy to see that thresholding the $s$-th element in $\mathcal{D}$ does not change the minimum and maximum of the output $\tilde{h}_i$.

Using these results as well as the fact that each input element across $\tilde{\boldsymbol{h}}(\mathcal{D}^{\mathrm{F}})$ can have at most two different values (because the weight of $\tilde{\boldsymbol{h}}$ is either unit or zero and $\mathcal{D}^{\mathrm{F}}$ is formatted), we have that $\tilde{\boldsymbol{h}}(\mathcal{D}^{\mathrm{F}})$ is just the dataset formatted from $\tilde{\boldsymbol{h}}(\mathcal{D})$! Finally, by induction, $(\boldsymbol{h}(\mathcal{D}))^{\mathrm{F}}$ can be interpolated by $\tilde{f}$, which yields that $(\tilde{\boldsymbol{h}}(\mathcal{D}))^{\mathrm{F}}$ can be interpolated by $\tilde{f}$ based on (*). Therefore, $\tilde{\boldsymbol{h}}(\mathcal{D}^{\mathrm{F}})$ can be interpolated by $\tilde{f}$, namely $\tilde{f} \circ \tilde{\boldsymbol{h}}$ interpolates $\mathcal{D}^{\mathrm{F}}$, which concludes the proof of Case 1.

**Case 2**: $\boldsymbol{h} : \mathbb{R}^{d_h} \to \mathbb{R}^{d_h}$ is the MaxMin layer. Here we consider the MaxMin layer with learnable bias, i.e. $\boldsymbol{h}(\boldsymbol{x}) = \mathrm{MaxMin}(\boldsymbol{x} + \boldsymbol{b})$ with parameter $\boldsymbol{b} \in \mathbb{R}^{d_h}$. Note that the bias will be absorbed into the previous linear layer so it does not change the resulting architecture, but it facilities the proof here. Assume $f \circ \boldsymbol{h}$ can interpolate dataset $\mathcal{D}$. We will construct a new MaxMin network $\tilde{f} \circ \tilde{\boldsymbol{h}}$ such that $\tilde{f}$ has the same topology as $f$, $\tilde{\boldsymbol{h}}$ is the MaxMin layer with bias $\tilde{\boldsymbol{b}}$, and $\tilde{f} \circ \tilde{\boldsymbol{h}}$ interpolates $\mathcal{D}^{\mathrm{F}}$. As in Case 1, we use the same notations of $\boldsymbol{h}(\mathcal{D})$, $\tilde{\boldsymbol{h}}(\mathcal{D})$, $(\boldsymbol{h}(\mathcal{D}))^{\mathrm{F}}$, $\boldsymbol{h}(\mathcal{D}^{\mathrm{F}})$, etc.

Given dataset $\mathcal{D}$, denote $\boldsymbol{z}^{(j)} = \boldsymbol{h}(\boldsymbol{x}^{(j)}) \, \forall j \in [n]$. It follows that $z_{2i-1}^{(j)} = \max(x_{2i-1}^{(j)} + b_{2i-1}, x_{2i}^{(j)} + b_{2i})$ and $z_{2i}^{(j)} = \min(x_{2i-1}^{(j)} + b_{2i-1}, x_{2i}^{(j)} + b_{2i})$. We can separately focus on each pair. Consider the following three sub-cases:

- Both the $(2i-1)$-th and the $2i$-th elements in $\boldsymbol{h}(\mathcal{D})$ are useful. In this sub-case, we can prove that both the $(2i-1)$-th and the $2i$-th elements in $\mathcal{D}$ are useful. This is because

$$
\begin{aligned}
2 &= u_{2i-1}(\boldsymbol{h}(\mathcal{D})) - l_{2i-1}(\boldsymbol{h}(\mathcal{D})) + u_{2i}(\boldsymbol{h}(\mathcal{D})) - l_{2i}(\boldsymbol{h}(\mathcal{D})) \\
&= \left( \max_{j \in [n]} z_{2i-1}^{(j)} + \max_{j \in [n]} z_{2i}^{(j)} \right) - \left( \min_{j \in [n]} z_{2i-1}^{(j)} + \min_{j \in [n]} z_{2i}^{(j)} \right) \\
&\leq \left( \max_{j \in [n]} x_{2i-1}^{(j)} + \max_{j \in [n]} x_{2i}^{(j)} + b_{2i-1} + + b_{2i} \right) - \left( \min_{j \in [n]} x_{2i-1}^{(j)} + \min_{j \in [n]} x_{2i}^{(j)} + b_{2i-1} + + b_{2i} \right) \\
&= u_{2i-1}(\mathcal{D}) - l_{2i-1}(\mathcal{D}) + u_{2i}(\mathcal{D}) - l_{2i}(\mathcal{D}) \leq 2.
\end{aligned}
$$

The equality holds when $u_{2i-1}(\mathcal{D}) - l_{2i-1}(\mathcal{D}) = 1$ and $u_{2i}(\mathcal{D}) - l_{2i}(\mathcal{D}) = 1$. Also, it is easy to see that thresholding the $(2i-1)$-th and the $2i$-th elements in $\mathcal{D}$ is equivalent to thresholding them in $\boldsymbol{h}(\mathcal{D})$. Formally, $l_{2i-1}(\boldsymbol{h}(\mathcal{D}^{\mathrm{F}})) = l_{2i-1}(\boldsymbol{h}(\mathcal{D}))$ and $u_{2i-1}(\boldsymbol{h}(\mathcal{D}^{\mathrm{F}})) = u_{2i-1}(\boldsymbol{h}(\mathcal{D}))$ (similar for $l_{2i}$ and $u_{2i}$). Therefore, we just set $\tilde{b}_{2i-1} = b_{2i-1}$ and $\tilde{b}_{2i} = b_{2i}$.

- Both the $(2i-1)$-th and the $2i$-th elements in $\boldsymbol{h}(\mathcal{D})$ are useless. This sub-case is trivial: these useless elements can be handled in $\tilde{f}$, because the first layer in $\tilde{f}$ is linear and the corresponding weights are zero for the useless input elements (see Case 1).

- One of the $(2i-1)$-th and the $2i$-th elements in $\boldsymbol{h}(\mathcal{D})$ is useful and the other is useless. Without loss of generality, assume the $(2i-1)$-th element in $\boldsymbol{h}(\mathcal{D})$ is useful and the $2i$-th element in $\boldsymbol{h}(\mathcal{D})$ is useless. Then we can prove that either $(l_{2i-1}(\boldsymbol{h}(\mathcal{D})) = l_{2i-1}(\mathcal{D})$ and $u_{2i-1}(\boldsymbol{h}(\mathcal{D})) = u_{2i-1}(\mathcal{D}))$ or $(l_{2i-1}(\boldsymbol{h}(\mathcal{D})) = l_{2i}(\mathcal{D})$ and $u_{2i-1}(\boldsymbol{h}(\mathcal{D})) = u_{2i}(\mathcal{D}))$. Otherwise, it is easy to see that both the $(2i-1)$-th and the $2i$-th elements in $\mathcal{D}$ are useless and thus the $(2i-1)$-th element in $\boldsymbol{h}(\mathcal{D})$ cannot be useful.

  Again without loss of generality, assume the $l_{2i-1}(\boldsymbol{h}(\mathcal{D})) = l_{2i-1}(\mathcal{D})$ and $u_{2i-1}(\boldsymbol{h}(\mathcal{D})) = u_{2i-1}(\mathcal{D})$. Then by assigning a sufficiently small bias $\tilde{b}_{2i} \to -\infty$ and setting $\tilde{b}_{2i-1} = b_{2i-1}$, one can ensure that even after thresholding/masking $\mathcal{D}$, the following relation still holds: $l_{2i-1}(\boldsymbol{h}(\mathcal{D}^{\mathrm{F}})) = l_{2i-1}(\mathcal{D}^{\mathrm{F}})$ and $u_{2i-1}(\boldsymbol{h}(\mathcal{D}^{\mathrm{F}})) = u_{2i-1}(\mathcal{D}^{\mathrm{F}})$. On the other hand, the $2i$-th element in $\boldsymbol{h}(\mathcal{D})$ is useless so it can be handled in $\tilde{f}$ by the corresponding zero weights in the first layer of $\tilde{f}$ (see Case 1).

Combining the three sub-cases, we have concluded the proof of Case 2. $\qquad\square$

Note that for a Boolean dataset $\mathcal{D}$, the formatted version is itself, i.e. $\mathcal{D}^{\mathrm{F}} = \mathcal{D}$. Thus the proof of Lemma B.22 directly leads to the following corollary:

**Corollary B.23.** *Let $\mathcal{D}$ be a Boolean dataset that can be interpolated by a MaxMin network $f$. Then there exists a network $\tilde{f}$ with the same architecture (i.e. depth and width) as $f$, such that:*

- *Denote $\mathbf{W}^{(l)} \in \mathbb{R}^{d_l \times d_{l-1}}$ as the weight matrix of the $l$-th layer of $\tilde{f}$. Then the following holds for all $i \in [d_l]$:*

$$[\mathbf{W}^{(l)}]_{i,:} \in \{s \cdot \boldsymbol{e}_r : s \in \{1, -1\}, r \in [d_{l-1}]\} \cup \{\boldsymbol{0}\}. \tag{47}$$

- *Denote $f_i^{(l)}(\boldsymbol{x})$ as the $i$-th neuron output of the $l$-th layer given input $\boldsymbol{x}$, and denote $\mathcal{S}_i^{(l)} = \{f_i^{(l)}(\boldsymbol{x}) : (\boldsymbol{x}, y) \in \mathcal{D}\}$. Then either $|\mathcal{S}_i^{(l)}| = 1$, or $|\mathcal{S}_i^{(l)}| = 2$ and $\max \mathcal{S}_i^{(l)} - \min \mathcal{S}_i^{(l)} = 1$.*

We are now ready for the proof of Theorem 3.10, which is restated below.

**Theorem B.24.** *Let $M_d$ be the minimum depth such that an $M_d$-layer MaxMin network can exactly represent any (discrete) $d$-dimensional Boolean function. Then $M_d = \Omega(d)$.*

*Proof.* Let $\tilde{f}$ be an $M_d$-layer MaxMin network that interpolates the $d$-dimensional Boolean function, satisfying the condition in Corollary B.23. Using the notations in Corollary B.23, one has

$$f_{2i-1}^{(l)}(\boldsymbol{x}) = \max\left([\mathbf{W}^{(l)}]_{2i-1,:}^{\mathrm{T}}\boldsymbol{f}^{(l-1)}(\boldsymbol{x}) + b_{2i-1}^{(l)}, [\mathbf{W}^{(l)}]_{2i,:}^{\mathrm{T}}\boldsymbol{f}^{(l-1)}(\boldsymbol{x}) + b_{2i}^{(l)}\right) \tag{48}$$

which must have one of the following forms due to the choice of weight $[\mathbf{W}^{(l)}]$:

- $f_{2i-1}^{(l)}(\boldsymbol{x}) = c$, where $c \in \mathbb{R}$ is a constant;

- $f_{2i-1}^{(l)}(\boldsymbol{x}) = \pm f_r^{(l-1)}(\boldsymbol{x}) + c$, where $r \in [d_{l-1}]$ and $c \in \mathbb{R}$ are constants;

- $f_{2i-1}^{(l)}(\boldsymbol{x}) = \max\left(s_1 \cdot f_{r_1}^{(l-1)}(\boldsymbol{x}), s_2 \cdot f_{r_2}^{(l-1)}(\boldsymbol{x})\right) + c$, where $s_1, s_2 \in \{1, -1\}$, $r_1, r_2 \in [d_{l-1}]$ and $c \in \mathbb{R}$ are constants.

In the last case, we further have $\max_{(\boldsymbol{x}, y) \in \mathcal{D}} s_1 \cdot f_{r_1}^{(l-1)}(\boldsymbol{x}) = \max_{(\boldsymbol{x}, y) \in \mathcal{D}} s_2 \cdot f_{r_2}^{(l-1)}(\boldsymbol{x})$ and $\min_{(\boldsymbol{x}, y) \in \mathcal{D}} s_1 \cdot f_{r_1}^{(l-1)}(\boldsymbol{x}) = \min_{(\boldsymbol{x}, y) \in \mathcal{D}} s_2 \cdot f_{r_2}^{(l-1)}(\boldsymbol{x})$ due to the second property in Corollary B.23. Therefore, $f_{2i-1}^{(l)}(\boldsymbol{x})$ can calculate a constant, the identity function, or the 2-ary logical OR of *literals*

(up to an additional bias). Similarly, $f_{2i}^{(l)}(\boldsymbol{x})$ can calculate a constant, the identity function, or the logical AND of *literals*.

Therefore, when considering the whole MaxMin network, it is no powerful than a 2-ary Boolean circuit. Here a 2-ary Boolean circuit refers to a directional acyclic computation graph where each internal node connects to at most two nodes by incoming edges and can only calculate the 2-ary logical AND, logical OR, and (unary) logical NOT. Define the depth of a 2-ary Boolean circuit to be the length of the longest path from the input to the output. It then follows that the depth correponds to the depth of the MaxMin netwotk, and it suffices to prove that the depth of a 2-ary Boolean circuit must be $\Omega(d)$ to represent certain Boolean functions of $d$ variables.

Note that for a 2-ary Boolean circuit that has $M$ layers and outputs a scalar, the number of nodes will not exceed $2^{M+1} - 1$, where the maximum size is achieved by a complete binary tree. However, the classic result in Boolean circuit theory (Shannon 1942) showed that for most Boolean functions of $d$ variables, a lower bound on the minimum size of 2-ary Boolean circuits is $\Omega(2^d/d)$ , which thus yields $M = \Omega(d)$ and concludes the proof. $\qquad\square$

## C   Special SortNet

### C.1   GroupSort Network

In this section, we formally prove that any GroupSort network can be exactly represented by a SortNet with the same topological structure. An $M$-layer GroupSort network with group size $G$ and hidden size $Gd$ can be defined as:

$$\begin{cases} \widetilde{\boldsymbol{z}}^{(l)} = \widetilde{\mathbf{W}}^{(l)}\widetilde{\boldsymbol{x}}^{(l-1)} + \widetilde{\boldsymbol{b}}^{(l)} & l \in [M], \\ \widetilde{x}_{iG+j}^{(l)} = \left(\{\widetilde{z}_{iG+t}^{(l)}\}_{t=1}^{G}\right)_{(j)} & l \in [M-1], i \in \{0, \cdots, d-1\}, j \in [G], \end{cases} \tag{49}$$

$$s.t. \ \|\widetilde{\mathbf{W}}^{(l)}\|_\infty \leq 1 \text{ for } l \in [M].$$

In (49), $(\cdot)_{(j)}$ denotes the $j$-th largest element in a sequence. The GroupSort network takes $\widetilde{\boldsymbol{x}}^{(0)}$ as input and outputs $\widetilde{\boldsymbol{z}}^{(M)}$. We will construct a SortNet such that the output is equal to $\widetilde{\boldsymbol{z}}^{(M)}$.

The key observation is that for any vector $\boldsymbol{z}$ in a bounded domain $\mathbb{K}$, the GroupSort activation with arbitrary group size $G$ can be represented by a sort operation plus a bias term. Concretely, we prove

$$\text{GroupSort}_G(\boldsymbol{z}) = \text{sort}(\boldsymbol{z} + \boldsymbol{b}) + \boldsymbol{c} \tag{50}$$

by assigning

$$b_{iG+j} = -iC, \quad c_{iG+j} = iC \quad \text{for } j \in [G], i \in \{0, \cdots, d-1\} \tag{51}$$

where $C$ is a sufficiently large positive constant. This is because a large $C$ dominates the relative order, so the sorting result is divided into groups of size $G$. Then $\boldsymbol{b} + \boldsymbol{c} = 0$ ensures that the introduced bias is finally eliminated.

Equipped with (50), we are now ready to construct the SortNet. We use the notations in Definition 4.1. Denote vectors $\boldsymbol{a}$ and $\boldsymbol{b}$ as

$$\begin{array}{ll} a_j = -jC & \text{for } j \in [Gd], \\ b_{iG+j} = -iC & \text{for } j \in [G], i \in [d]. \end{array}$$

The parameters of SortNet are assigned below:

$$\begin{array}{ll} \boldsymbol{w}^{(l,k)} = [\widetilde{\mathbf{W}}^{(l)}]_{k,:} \ (\text{the } k\text{th row}) & k \in [Gd] \\ \boldsymbol{b}^{(1,k)} = \boldsymbol{a} & k \in [Gd] \\ \boldsymbol{b}^{(2,k)} = \boldsymbol{b} + \widetilde{\boldsymbol{b}}^{(1)} - \widetilde{\mathbf{W}}^{(1)}\boldsymbol{a} & k \in [Gd] \\ \boldsymbol{b}^{(l,k)} = \boldsymbol{b} + \widetilde{\boldsymbol{b}}^{(l-1)} - \widetilde{\mathbf{W}}^{(l-1)}\boldsymbol{b} & k \in [Gd], 3 \leq l \leq M \\ \boldsymbol{b}^{\text{out}} = \widetilde{\boldsymbol{b}}^{(M)} - \widetilde{\mathbf{W}}^{(M)}\boldsymbol{b} \end{array}$$

The activation is simply chosen to be the identity function $\sigma(x) = x$. This leads to the following calculation which can be proved straightforwardly:

$$\boldsymbol{x}^{(1)} = \widetilde{\boldsymbol{z}}^{(1)} - \widetilde{\boldsymbol{b}}^{(1)} + \widetilde{\mathbf{W}}^{(1)}\boldsymbol{a}$$

$$\boldsymbol{x}^{(l)} = \widetilde{\boldsymbol{z}}^{(l)} - \widetilde{\boldsymbol{b}}^{(l)} + \widetilde{\mathbf{W}}^{(l)}\boldsymbol{b} \quad \text{for } 2 \leq l \leq M$$

Finally, the SortNet outputs $\boldsymbol{f}(\boldsymbol{x}) = \boldsymbol{x}^{(M)} + \boldsymbol{b}^{\text{out}} = \widetilde{\boldsymbol{z}}^{(M)}$, which matches the output of the GroupSort network. Proof finishes.

**Remark C.1.** In the above construction, the same value is assigned for all biases $\boldsymbol{b}^{(l,k)}$ of each SortNet layer. It implies that SortNet is more powerful than GroupSort, since SortNet allows diverse biases for different neurons in the same layer. This is very useful because the value of bias plays an important role in controlling the sorting behavior. For example, in a SortNet neuron (7), setting $b_i^{(l,k)}$ to a sufficiently large value will dominant $x_i^{(l-1)}$ to rank the first, which can extract the $i$-th coordinate of the input vector if $\boldsymbol{w}^{(l,k)} = \boldsymbol{e}_1$. Similarly, setting $b_i^{(l,k)}$ to a large negative value will cause $x_i^{(l-1)}$ to rank the last, and when combined with $w_i^{(l,k)} = 0$, will preclude $x_i^{(l-1)}$ in the computation of the neuron output. However, for a GroupSort layer, performing these operations will affect the output of other neurons as all neurons in the group share the same bias vector and sorting operation. The interference between neurons may lead to some neurons being wasted and not being able to express the required operations, which is undesirable. In contrast, SortNet does not face the above problems.

## C.2 $\ell_\infty$-distance Net

In this section we formally prove that any $\ell_\infty$-distance net can be exactly represented by a special SortNet with the same topological architecture. An $M$-layer $\ell_\infty$-distance net with a hidden size of $d$ can be formally defined as:

$$\widetilde{x}_k^{(l)} = \|\widetilde{\boldsymbol{x}}^{(l-1)} - \widetilde{\boldsymbol{w}}^{(l,k)}\|_\infty + \widetilde{b}_k^{(l)}, \quad l \in [M], k \in [d]. \tag{52}$$

The network takes $\widetilde{\boldsymbol{x}}^{(0)}$ as input and outputs $\widetilde{\boldsymbol{x}}^{(M)}$. We will construct a SortNet such that the output exactly matches $\widetilde{\boldsymbol{x}}^{(M)}$.

The key observation is that for any input vector $\boldsymbol{z}$ and parameter $\boldsymbol{w}$,

$$\|\boldsymbol{z} - \boldsymbol{w}\|_\infty = \max_i |z_i - w_i| = \boldsymbol{e}_1^{\mathrm{T}} \operatorname{sort}(|\boldsymbol{z} - \boldsymbol{w}|) \tag{53}$$

where $\boldsymbol{e}_1$ is the unit vector with the first element being one. Based on the above equation, we now construct a SortNet as follows. We use the notations in Definition 4.1 and set all weights $\boldsymbol{w}^{(l,k)} = \boldsymbol{e}_1$, set biases

$$\boldsymbol{b}^{(1,k)} = -\widetilde{\boldsymbol{w}}^{(1,k)},$$
$$\boldsymbol{b}^{(l,k)} = \widetilde{\boldsymbol{b}}^{(l-1)} - \widetilde{\boldsymbol{w}}^{(l,k)} \quad \text{for } 2 \le l \le M,$$
$$\boldsymbol{b}^{\text{out}} = \widetilde{\boldsymbol{b}}^{(M)},$$

and set the activation $\sigma(x) = |x|$. Then by a simple induction over layer $l$ we can prove

$$\boldsymbol{x}^{(l)} = \widetilde{\boldsymbol{x}}^{(l)} - \widetilde{\boldsymbol{b}}^{(l)}. \quad \forall \, l \in [M] \tag{54}$$

Finally, the SortNet outputs $\boldsymbol{f}(\boldsymbol{x}) = \boldsymbol{x}^{(M)} + \boldsymbol{b}^{\text{out}} = \widetilde{\boldsymbol{x}}^{(M)}$, which matches the output of the $\ell_\infty$-distance net. Proof finishes.

## D  An Efficient GPU Implementation of SortNet

In this section we describe an efficient GPU implementation of Section 4 for training and inference, which is used in this paper. Some basic tensor operations are described in the context of Pytorch framework.

### D.1  Training

Consider a fully-connected SortNet layer $\boldsymbol{f}$ with parameters $\mathbf{W}, \mathbf{B} \in \mathbb{R}^{d_{\text{out}} \times d_{\text{in}}}$, and $W_{ij} = (1 - \rho)\rho^{j-1}$. For an input batch $\mathbf{X} \in \mathbb{R}^{n \times d_{\text{in}}}$ consisting of $n$ samples, denote the output in a training iteration as $\mathbf{Z} = \boldsymbol{f}(\mathbf{X}) \in \mathbb{R}^{n \times d_{\text{out}}}$ which is stochastic and depends on the mask. The generated stochastic mask is also a matrix denoted as $\mathbf{S} \in \{0,1\}^{n \times d_{\text{in}}}$. According to the formula (8),

$$Z_{ik} = \max_{j \in [d_{\text{in}}]} S_{ij}\sigma(X_{ij} + B_{kj}) \tag{55}$$

Note that $S_{ij}$ can only be 0 or 1, therefore the multiplication between $S_{ij}$ and $\sigma(X_{ij} - B_{kj})$ is unnecessary. We can enumerate the indices such that $S_{ij} = 1$, resulting in the following calculation:

$$Z_{ik} = \max_{j, \text{ s.t. } S_{ij}=1} \sigma(X_{ij} + B_{kj}) \tag{56}$$

With hyper-parameter $\rho$, (56) will give a computational complexity of $\Theta((1-\rho)nd_{\text{in}}d_{\text{out}})$.

In fact, the above computation can be further accelerated if two adjacent SortNet layers are cascaded. In this case, the output of the first layer becomes the input of the second layer, which will also be dropped out, so the unused neurons does not need to be computed. In other words, it suffices to only calculate the output neurons which are involved in the next layer of computation. This results in a computational complexity of $\Theta((1-\rho)^2 nd_{\text{in}}d_{\text{out}})$.

Things become more complicated when it turns to a GPU implementation. Since GPU is highly parallel, it is best to share the same instructions for different samples in a batch for ease of parallelization. However, the number of ones in the mask may vary for different samples. In other words, in (56) enumerating the indices $j$ subject to $S_{ij} = 1$ cannot be efficiently executed in a parallel way.

To address the problem, we introduce a notion called sub-batch. In a sub-batch, all samples have the same mask for ease of parallelization, and different sub-batches have different masks. Typically, a sub-batch contains 32 samples (due to the GPU underlying architecture). A batch size of 512 then contains 16 sub-batches. For each sub-batch, we first generate two index sequences $s^{\text{in}} \in [d_{\text{in}}]^{d_1}$ and $s^{\text{out}} \in [d_{\text{out}}]^{d_2}$ (the input mask and output mask respectively). We then fetch the input neurons using mask $s^{\text{in}}$, which corresponds to a gather operation (e.g. `torch.gather` in Pytorch). Denote the gathered input as $\widetilde{\mathbf{X}} \in \mathbb{R}^{32 \times d_1}$. Then the output $\widetilde{\mathbf{Z}} \in \mathbb{R}^{32 \times d_2}$ of this sub-batch can be calculated by

$$\widetilde{Z}_{ik} = \max_{j \in [d_1]} \sigma(\widetilde{X}_{ij} + B_{s_k^{\text{out}}, s_j^{\text{in}}}) \quad i \in [32], k \in [d_2]. \tag{57}$$

We finally point out that the introduced sub-batch is only used to accelerate GPU implementation. If one does not care about the speed, simply using (56) for training leads to almost the same test performance.

## D.2  Inference

For inference, the sorting operations must be calculated exactly. To speed up the computation, note that the weight $w$ in Proposition 4.4 is exponentially decayed, implying that a lot of elements are close to zero. This leads to the following results:

**Proposition D.1.** *Let $w \in \mathbb{R}^d$ be a vector with $w_i = (1-\rho)\rho^{i-1}$. Then for any vector $x \in \mathbb{R}_+^d$ with non-negative elements and any integer $k > 0$,*

$$0 \leq w^{\text{T}} \text{sort}(x) - \sum_{i=1}^{k} w_i x_{(i)} \leq \rho^k \|x\|_{\infty}. \tag{58}$$

The proof of Proposition D.1 is trivial. Based on the above proposition, picking a $k$ much smaller than $d$ already suffices to control the error below the numerical precision. In this way, only a top-$k$ operation is needed. We choose $k = 10$ for all experiments. Also note that zeroing out some elements of a weight vector does not contradict the Lipschitz property of the sort neuron, and thus margin-based certification still gives a correct certified accuracy.

## E  Training Details

In this section we provide complete train details for readers to reproduce our results. Our experiments are implemented using the Pytorch framework.

### E.1  Datasets

We consider four benchmark datasets: MNIST, CIFAR-10, TinyImageNet and ImageNet ($64 \times 64$). The statistics of these datasets are listed below, including the number of classes, the dataset size, and the image size.

Table 4: Datasets used in this paper.

| Dataset | # Classes | # Images (train) | # Images (test) | Image size |
|---|---|---|---|---|
| MNIST | 10 | 60K | 10K | $28 \times 28$ |
| CIFAR-10 | 10 | 50K | 10K | $32 \times 32$ |
| TinyImageNet | 200 | 100K | 10K | $64 \times 64$ |
| ImageNet | 1000 | 1.28M | 50K | $64 \times 64$ |

For all datasets, we pre-process each training image using the same pipeline as previous works. Concretely, we use random horizontal flip (except for MNIST) and random crop as the data augmentation. Each image is padded to a larger size with `padding_mode=edge` and then randomly crop to the original size. The number of pixels padded is 1,3,5,4 for MNIST, CIFAR-10, TinyImageNet, and ImageNet, respectively. Finally, the images are normalized to have zero-mean and unit variance across the dataset. For testing, no data augmentation is performed.

## E.2  Models

This paper considers 3 types of models in total. The first is a simple fully-connected SortNet. The second is the combination of a Lipschitz SortNet backbone and a non-Lipschitz two-layer perceptron. In this way SortNet serves as a robust feature extractor and the top perceptron is used for classification. The last is a larger model similar to the second model, except we try to use a partially convolutional SortNet backbone for better performance. Details of these models are presented in Table 5. For the ImageNet dataset, we increase the number of hidden neurons of the top perceptron to 2048, since the number of classes is 1000 (larger than the original hidden size which is only 512).

Table 5: Network architectures used in this paper. In the table, SortFC($n$) denotes a fully-connected SortNet layer with $n$ output neurons, and SortConv($n$, ker=$k$) denotes a convolutional SortNet layer with $n$ output channels and a kernel size of $k$. Padding is not used. We use s=$s$ to denote strided convolutions with step $s$, and by default s=1. FC($n$) is the fully-connected linear layer with $n$ output neurons. $K$ denotes the number of classes.

| SortNet | SortNet+MLP | SortNet+MLP (2x) |
|---|---|---|
|  | SortFC(5120) | SortConv(5120,ker=60,s=4) |
| SortFC(5120) | SortFC(5120) | SortConv(5120, ker=1) |
| SortFC(5120) | SortFC(5120) | SortConv(5120, ker=1) |
| SortFC(5120) | SortFC(5120) | SortConv(5120, ker=1) |
| SortFC(5120) | SortFC(5120) | SortConv(5120, ker=1) |
| SortFC(5120) | FC(512, bias=True) | FC(512, bias=True) |
| SortFC($K$) | Tanh | Tanh |
|  | FC($K$, bias=True) | FC($K$, bias=True) |

We also list information of the model size in Table 6. Roughly speaking, SortNet and SortNet+MLP have similar computational costs, and SortNet+MLP (2x) is about 4 times larger than SortNet+MLP. We also point out that the models used in this paper have exactly the same topology and number of parameters as $\ell_\infty$-distance Net (or $\ell_\infty$-distance Net+MLP) in [73], which enables a fair comparison between our approach and theirs.

## E.3  Loss function

Let $(\boldsymbol{x}, y)$ be a training sample and $\boldsymbol{f}$ be the mapping represented by the network. For a SortNet model, since it is strictly 1-Lipschitz, we can directly apply hinge loss to increase the margin of output logits (see Proposition 3.1). To make the training more effective, we adopt the following loss function:

$$\ell(\boldsymbol{f}, \boldsymbol{x}, y) = \lambda \cdot \ell_{\mathrm{CE}}(s \cdot \boldsymbol{f}(\boldsymbol{x}), y) + \mathbb{I}\left(\arg\max_i [\boldsymbol{f}(\boldsymbol{x})]_i = y\right) \ell_{\mathrm{hinge}}(\boldsymbol{f}(\boldsymbol{x})/\theta, y) \qquad (59)$$

where $\theta$ is the hinge threshold hyper-parameter, $s$ is a learnable scalar, $\lambda$ is the mixing hyper-parameter, $\mathbb{I}(\cdot)$ is the indicator function, and $\ell_{\mathrm{CE}}$ and $\ell_{\mathrm{hinge}}$ are the cross-entropy loss and standard hinge loss,

Table 6: Statistics of model size on different datasets. In the table, "FLOPs" measures the number of floating-point operations in forward propagation, and "# Neurons" denotes the number of hidden neurons.

| | Model | CIFAR-10 | Tiny ImageNet | ImageNet $(64 \times 64)$ |
|---|---|---|---|---|
| | SortNet | 121M | – | – |
| FLOPs | SortNet+MLP | 123M | 171M | 180M |
| | SortNet+MLP (2x) | – | 651M | 684M |
| | SortNet | 121M | – | – |
| # Parameters | SortNet+MLP | 123M | 170M | 180M |
| | SortNet+MLP (2x) | – | 171M | 204M |
| | SortNet | 25.6K | – | – |
| # Neurons | SortNet+MLP | 26.1K | 26.1K | 27.6K |
| | SortNet+MLP (2x) | – | 103K | 104K |

respectively:

$$\ell_{\mathrm{CE}}(\boldsymbol{z}, y) = \log\left(\sum_i \exp(z_i)\right) - z_y, \tag{60}$$

$$\ell_{\mathrm{hinge}}(\boldsymbol{z}, y) = \max\{\max_{i \neq y} z_i - z_y + 1, 0\}. \tag{61}$$

Intuitively speaking, this loss first uses cross entropy to improve the clean accuracy, then boost the certified robustness for samples that have been correctly classified. Such a loss design is quite common and is also used in prior works [55, 74, 14]. The coefficient $\lambda$ balances the two loss terms and controls the trade-off between accuracy and robustness. We find good results can be achieved by starting with an initial value $\lambda_0$ and slowly reducing it throughout training until it approaches zero (similar to [75]).

Now consider the composite SortNet+MLP model. The whole model is non-Lipschitz, so we have to use relaxation-based methods to calculate a margin vector for the top MLP. We adopt the simplest method called *interval bound propagation* (IBP) [21], similar to [73]. After obtaining the margin vector, we use the following loss function

$$\ell(\boldsymbol{f}, \boldsymbol{x}, y) = \lambda \cdot \ell_{\mathrm{CE}}(s \cdot \boldsymbol{f}(\boldsymbol{x}), y) + \mathbb{I}\left(\arg\max_i [\boldsymbol{f}(\boldsymbol{x})]_i = y\right) \ell_{\mathrm{hinge}}(\mathrm{IBP}(\boldsymbol{f}, \boldsymbol{x}, y, \epsilon), y) \tag{62}$$

which is almost the same as (59). Here $\mathrm{IBP}(\boldsymbol{f}, \boldsymbol{x}, y, \epsilon)$ denotes the margin calculated by interval bound propagation for sample $(\boldsymbol{x}, y)$ and network $\boldsymbol{f}$ under perturbation radius $\epsilon$. Similarly, the coefficient $\lambda$ decays from an initial value $\lambda_0$ to zero throughout training, and the value $\epsilon$ follows a warmup schedule in the training process. Such a strategy is broadly applied in previous literature [21, 75, 69]. See Appendix E.5 for details of the $\lambda$ and $\epsilon$ schedule.

### E.4 Training strategy

**Parameter initialization**. All parameters (in both the SortNet and the top MLP) are initialized using the standard Gaussian distribution. The learnable scalar in the loss function is initialized to be 1. To keep the output scale of the linear layer the same as the input scale, the output of each linear neuron is further multiplied by $1/\sqrt{d}$, where $d$ is the input dimension. Such an initialization strategy is often called the NTK initialization [26].

**Training pipeline**. At each iteration, we first generate masks in each *hidden* layer and remove neurons with zero masks, resulting in a sub-network. This is similar to applying dropout in each hidden layer with a dropout rate of $\rho$ (we do not dropout the input). We then perform forward and backward propagation on the sub-network (corresponding to an $\ell_\infty$-distance net) following Section D.1 and calculate the gradients of parameters. We use two additional tricks that is adopted in [73]: batch normalization and $\ell_p$-relaxation.

**Batch normalization**. Since the output of the sort neuron (7) is always non-negative due to the absolute-value activation function, a mean-shift version of batch normalization [25] is applied after each SortNet layers. Concretely, it normalizes each hidden neuron to have zero mean for a batch of

samples in each iteration. We do not use scaling since it destroys the Lipschitz property. Unlike the original batch normalization, we do not calculate the running mean statistics in our training procedure since the model is also randomly sampled (with hidden neurons dropped out). The running mean can be calculated later after the training finishes, by conducting one pass of forward propagation on the whole training dataset for the inference model.

$\ell_p$**-relaxation**. The calculation of maximum in (8) leads to sparse gradient for input $x$, which makes optimization difficult. To alleviate the problem, we adopt $\ell_p$-relaxation proposed in [73] for training, which gives a smooth approximation of the maximum operation. Concretely, for a vector $x$ with non-negative elements, the following holds:

$$\max_i x_i = \lim_{p \to +\infty} \left( \sum_i x_i^p \right)^{1/p}. \tag{63}$$

At the initial phase, $p$ is set to a small value so that the gradients are non-sparse. As training progresses, $p$ gradually increases until reaching a large number. For the last few epochs, $p$ is set to infinity. In this paper, we exponentially increase $p$ from 8 to 1000 without tuning for all experiments, following [73].

### E.5 Details and hyper-parameters

All the experiments regarding SortNet models are run on NVIDIA Tesla V100S-PCIe GPUs. As for speed comparison to prior works, we further test time metrics for all prior works as well as our approach on the latest NVIDIA RTX-3090 GPUs (24GB memory). The CUDA version is 11.1.

**Hyper-parameters of the optimizer**. Following [73, 74], we adopt the Adam optimizer with hyper-parameters $\beta_1 = 0.9$, $\beta_2 = 0.99$ and $\epsilon = 10^{-10}$, and use a batch size of 512 for all datasets and experiments. The learning rate is initialized to be 0.02 and is decayed using a simple cosine annealing throughout the whole training process. The only exception is for ImageNet, where we use 2 GPUs in parallel due to the large dataset size. This results in a total batch size of 1024, and we change the initial learning rate to be 0.01. In all experiments, the weight decay is set to 0.02 for all but SortNet parameters (e.g., for the top MLP), and we do not use weight decay for SortNet parameters.

$\epsilon$ **Schedule**. For the SortNet+MLP model, a warmup over $\epsilon$ is required due to the use of interval bound propagation, and we adopt the same warmup schedule as in [69] (with the same hyper-parameters), which increases $\epsilon_{\text{train}}$ first-exponentially-then-linearly from 0 to $1.1\epsilon_{\text{test}}$.

**Epoch-related hyperparameters**. Such hyper-parameters are listed below, which depends on the dataset.

- **MNIST**. We train all models for 1500 epochs. The $\ell_p$ schedule starts at the 100th epoch and ends at the 1450th epoch.
- **CIFAR-10**. We train all models for 3000 epochs. The $\ell_p$ schedule starts at the 200th epoch and ends at 2950th epoch. The warmup of $\epsilon$ is in the first 500 epochs.
- **TinyImageNet**. We train all models for 1000 epochs. The $\ell_p$ schedule starts at the 100th epoch and ends at the 980th epoch. The warmup of $\epsilon$ is in the first 500 epochs.
- **ImageNet** ($64 \times 64$). We train all models for 300 epochs. The $\ell_p$ schedule starts at the 50th epoch and ends at the 290th epoch. The warmup of $\epsilon$ is in the first 200 epochs.

**Hyper-parameters $\theta$, $\rho$ and $\lambda$**. For the SortNet model, there is a hinge threshold hyper-parameter $\theta$ that has to be tuned, and we choose $\theta$ using a course grid search. Concertely, we pick $\theta = 0.6$ for $\epsilon = 0.1$ and $\theta = 0.9$ for $\epsilon = 0.3$ on MNIST, and pick $\theta = 16/255$ for $\epsilon = 2/255$ and $\theta = 48/255$ for $\epsilon = 8/255$ on CIFAR-10. As for hyper-parameter $\rho$, we find $\rho = 0.3$ typically works well on most settings, while a slight adjustment may achieve the best performance. On MNIST, we pick $\rho = 0.3$ for $\epsilon = 0.1$ and $\rho = 0.25$ for $\epsilon = 0.3$. On CIFAR-10, we pick $\rho = 0.25$ for $\epsilon = 2/255$ and $\rho = 0.15$ for $\epsilon = 8/255$ in SortNet, and pick $\rho = 0.3$ for $\epsilon = 2/255$ and $\rho = 0.4$ for $\epsilon = 8/255$ in SortNet+MLP. On Tiny-ImageNet and ImageNet $64 \times 64$, we do not tune the hyper-parameter $\rho$ and simply pick the value 0.3. Sensitivity analysis of hyper-parameter $\rho$ is shown in Section F.

Finally, as for hyper-parameter $\lambda$ in loss (59), we find choosing its value proportional to $1/\epsilon$ works well in all experiments. Such a choice is quite reasonable in balancing the cross-entropy term and the hinge term in the loss (59) because the magnitude of the gradient of the hinge term $\ell_{\text{hinge}}(f(x)/\theta, y)$ is $\mathcal{O}(1/\theta)$ which is (roughly) inversely proportional to $\epsilon$. To make the value more regular, we just pick $\lambda_0$ from the set $\{0.01, 0.02, 0.05, 0.1, 0.2, 0.5, 1.0\}$, while maintaining the property of being

inversely proportional to $\epsilon$. Throughout training, $\lambda$ is decayed from $\lambda_0$ to a vanishing small value $0.01\lambda_0$ exponentially, and is performed simultaneously with the $\ell_p$ relaxation process. The concrete values in different settings are listed below.

Table 7: The value of hyper-parameter $\lambda$ in loss (59) used in this paper.

| Setting | $\lambda_0$ | $\lambda_{\text{end}}$ |
|---|---|---|
| MNIST ($\epsilon = 0.1$) | 0.1 | 0.001 |
| MNIST ($\epsilon = 0.3$) | 0.02 | 0.0002 |
| CIFAR-10 ($\epsilon = 2/255$) | 1.0 | 0.01 |
| CIFAR-10 ($\epsilon = 8/255$) | 0.2 | 0.002 |

As for $\lambda$ in loss (62), we find the value do not depend on $\epsilon$, and in all settings we simply pick $\lambda_0 = 1.0$ and $\lambda_{\text{end}} = 0.01$ without tuning.

### E.6 Reproducing baseline methods

In this paper we also reproduce baseline methods by measuring the wall-clock time in training and certification. For all methods in Tables 1, 2 and 3, we test the computational cost on a single NVIDIA RTX-3090 GPU. We use the official Github codes and commands for each method whenever possible. In some situations, the batch size used in the original command may be too large to fit the 24GB GPU memory, so we reduce the batch size to match the memory capacity.

- **IBP** [21]. We use the official code of `CROWN-IBP` on CIFAR-10 dataset since the original code may not be reproducible according to [75]. Note that the corresponding numbers in Table 2 is also obtained from the CROWN-IBP paper. We use the official code of `auto_LiRPA` on TinyImageNet and ImageNet ($64 \times 64$) datasets. The corresponding numbers of TinyImageNet dataset in Table 3 are obtained from [69]. Due to limited GPU memory, for TinyImageNet and ImageNet ($64 \times 64$), we reduce the batch size to 64.

- **CROWN-IBP** [75]. We use the official code of `CROWN-IBP` on CIFAR-10 dataset. Due to limited GPU memory, we reduce the batch size to 256. The training speed is measured in the $\epsilon$-warmup phase, and the certification speed is measured at $\epsilon = 2/255$ where both IBP and linear relaxation is used in certification.

- **CROWN-IBP** [69]. We use the official code of `auto_LiRPA` on CIFAR-10, TinyImageNet, and ImageNet ($64 \times 64$) datasets. Due to limited GPU memory, we reduce the batch size to 256 on CIFAR-10 and 32 on TinyImageNet and ImageNet. The training speed is measured in the $\epsilon$-warmup phase, and the certification is the simple IBP for these settings.

- **IBP** [52]. We use the official code of `Fast-Certified-Robust-Training`. Due to limited GPU memory, for TinyImageNet dataset we reduce the batch size to 64.

- **CAP** [66]. We use the official code of `convex_adversarial`.

- **COLT** [3]. We use the official code of `COLT`. The training has 4 stages, and the per-epoch training time (seconds) for each stage is 51, 240, 332, 385, respectively. Therefore the average time listed in Table 2 is 252.0 seconds.

- $\ell_\infty$**-distance Net** [73, 74]. We use the official code of `L_inf-dist-net-v2`. The training speed is measured in the $\ell_p$-relaxation phase which is the slowest.

We also reproduce 100-PGD attack for the $\ell_\infty$-distance net in the original paper [73], because their paper only reported the robust accuracy under the (*weaker*) 20-step PGD attack. Furthermore, we report two additional settings: $\epsilon = 0.1$ on MNIST and $\epsilon = 2/255$ on CIFAR-10 based on their Github repo `L_inf-dist-net`, which are not presented in [73].

We also try to reproduce the result of $\ell_\infty$-distance net on TinyImageNet using the training approach proposed by [74]. We try our best to grid search over several hyper-parameters, including $\lambda_0$, $\lambda_{\text{end}}$ and $\theta$, while keeping other hyper-parameters the same as in [74]. The values are chosen from $\lambda_0 \in \{0.05, 0.1, 0.2, 0.5, 1.0\}$, $\lambda_{\text{end}} \in \{0.0005, 0.001, 0.002, 0.005, 0.01\}$, and $\theta \in \{12/255, 16/255, 20/255, 24/255\}$. We use 1000 training epochs, and the $\ell_p$-relaxation starts at the 20th epoch. Results are present in Table 3, where we can only achieve 11.04% certified accuracy, which is much lower than other methods. We hypothesis that their approach may not suit for large-scale datasets with a huge number of classes.

## F  Ablation Studies

We finally make a further investigation of the performance of SortNet models by varying the value of $\rho$ and comparing it with $\ell_\infty$-distance net (corresponding to $\rho = 0$), using the training approach proposed in Appendix E. Figure 1 presents the performance of trained models, where we plot both clean accuracy and certified accuracy w.r.t. different choices of hyper-parameters $\rho$ ranging from 0 to 0.5. It can be seen that for all four cases in Figure 1, a mid-range value consistently gives the best results. In particular, choosing $\rho = 0.3$ typically attains the peak certified accuracy, and the improvement is most significant when comparing to the the extreme case of $\rho = 0$ (often more than 5 points). Considering that the training in each figure is under the same configuration and hyper-parameters, the improvements justify that SortNet is a better Lipschitz model than $\ell_\infty$-distance net for certified $\ell_\infty$ robustness, due to the introduced full order statistics.

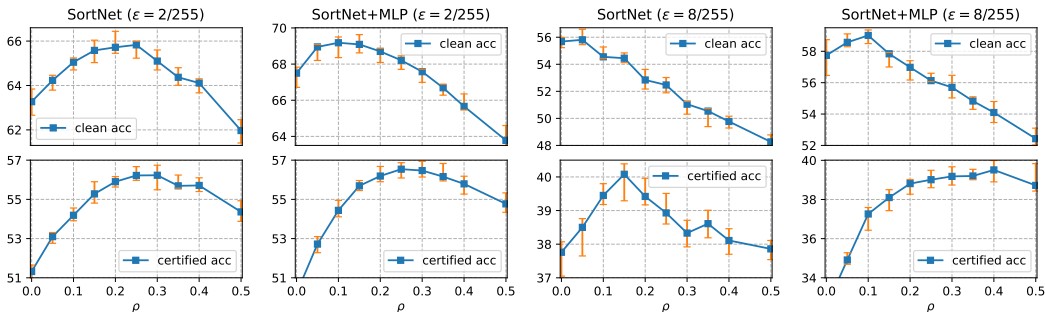

Figure 1: Performance of SortNet/SortNet+MLP trained with different hyper-paramters $\rho$ on CIFAR-10 dataset. For each $\rho$ we independently run 8 experiments and show the median value (blue square) as well as the best/worst performance (orange bar).

## G  Randomized Smoothing

Different from other approaches in Table 2, randomized smoothing is a *probabilistic* method that provides certified guarantees mainly for $\ell_2$ perturbations. To apply these methods in the $\ell_\infty$ perturbation case, a conversion of perturbation radius is performed by using norm inequalities. Specifically, to certify the robustness under $\ell_\infty$ perturbations with radius $\epsilon$, one can certify the robustness under $\ell_2$ with radius $\epsilon\sqrt{d}$ where $d$ is the input dimension. This clearly produces a lower bound estimate of the certified accuracy.

In the following table, we present the best-known results on MNIST and CIFAR-10. See [49, 5, 27] as well as [74, Appendix D] for references. In summary, randomized smoothing is competitive for the $\epsilon = 2/255$ case on CIFAR-10. However, when $\epsilon$ is relatively large, randomized smoothing cannot achieve non-trivial performance.

Table 8: Results of randomized smoothing in different settings.

| Dataset | $\epsilon$ | Clean | Certified | Reference |
|---------|-----------|-------|-----------|-----------|
| MNIST | 0.1 | $\approx$92 | $\approx$12 | [27] |
| | 0.3 | – | 10.0 | – |
| CIFAR-10 | 2/255 | 78.8 | 62.6 | [5] |
| | 8/255 | 52.3 | 25.2 | [27] |

## H  Full Results

As is pointed out above, for all SortNet models we run 8 set of experiments independently and report the median of the accuracy. In this section, we release full results of clean accuracy and certified accuracy in Tables 9 and 10. The best achieved numbers are indicated in boldface.

Table 9: Full results of clean accuracy over 8 independent runs, sorted in descending order.

| Dataset | $\epsilon$ | Model | Clean | | | | | | | | Median |
|---|---|---|---|---|---|---|---|---|---|---|---|
| MNIST | 0.1 | SortNet | **99.08** | 99.07 | 99.04 | 99.01 | 99.01 | 98.97 | 98.95 | 98.94 | 99.01 |
| | 0.3 | SortNet | **98.68** | 98.67 | 98.59 | 98.59 | 98.59 | 98.57 | 98.53 | 98.46 | 98.59 |
| CIFAR-10 | 2/255 | SortNet | **66.00** | 65.98 | 65.96 | 65.86 | 65.79 | 65.78 | 65.69 | 65.23 | 65.83 |
| | | SortNet+MLP | **67.72** | 67.72 | 67.64 | 67.59 | 67.57 | 67.11 | 66.99 | 66.99 | 67.58 |
| | 8/255 | SortNet | **54.84** | 54.77 | 54.70 | 54.59 | 54.30 | 54.30 | 54.28 | 54.12 | 54.45 |
| | | SortNet+MLP | **54.80** | 54.39 | 54.33 | 54.13 | 54.07 | 54.02 | 53.71 | 53.46 | 54.10 |
| Tiny ImageNet | 1/255 | SortNet+MLP | **24.17** | 24.07 | 24.01 | 23.97 | 23.95 | 23.85 | 23.75 | 23.52 | 23.96 |
| | | SortNet+MLP(2x) | **25.77** | 25.69 | 25.55 | 25.43 | 25.33 | 25.30 | 25.26 | 25.02 | 25.38 |
| ImageNet $64 \times 64$ | 1/255 | SortNet+MLP | **13.48** | 13.37 | 13.35 | 13.33 | 13.30 | 13.26 | 13.22 | 13.21 | 13.32 |
| | | SortNet+MLP(2x) | **14.96** | 14.87 | 14.86 | 14.84 | 14.82 | 14.79 | 14.76 | 14.71 | 14.83 |

Table 10: Full results of certified accuracy over 8 independent runs, sorted in descending order.

| Dataset | $\epsilon$ | Model | Certified | | | | | | | | Median |
|---|---|---|---|---|---|---|---|---|---|---|---|
| MNIST | 0.1 | SortNet | **98.14** | 98.13 | 98.11 | 98.07 | 98.06 | 98.05 | 98.04 | 97.93 | 98.07 |
| | 0.3 | SortNet | **93.40** | 93.40 | 93.39 | 93.39 | 93.36 | 93.25 | 93.22 | 93.19 | 93.38 |
| CIFAR-10 | 2/255 | SortNet | **56.67** | 56.65 | 56.29 | 56.23 | 56.20 | 56.07 | 56.06 | 55.97 | 56.22 |
| | | SortNet+MLP | **56.94** | 56.80 | 56.70 | 56.63 | 56.30 | 56.25 | 56.16 | 56.14 | 56.47 |
| | 8/255 | SortNet | **40.39** | 40.14 | 40.13 | 40.10 | 40.05 | 39.96 | 39.61 | 39.29 | 40.08 |
| | | SortNet+MLP | **39.99** | 39.76 | 39.70 | 39.56 | 39.45 | 39.40 | 39.25 | 38.90 | 39.51 |
| Tiny ImageNet | 1/255 | SortNet+MLP | **17.92** | 17.52 | 17.51 | 17.48 | 17.43 | 17.33 | 17.25 | 17.21 | 17.46 |
| | | SortNet+MLP(2x) | **18.18** | 17.74 | 17.68 | 17.64 | 17.64 | 17.62 | 17.59 | 17.50 | 17.64 |
| ImageNet $64 \times 64$ | 1/255 | SortNet+MLP | **9.02** | 9.00 | 8.97 | 8.93 | 8.92 | 8.89 | 8.89 | 8.89 | 8.93 |
| | | SortNet+MLP(2x) | **9.54** | 9.45 | 9.43 | 9.41 | 9.41 | 9.41 | 9.35 | 9.35 | 9.41 |

# I    Training SortNet with Shorter Epochs

In our main results, we use quite long training epochs in order to achieve the best performance. For example, the number of epochs is 3000 on CIFAR-10, which is larger than several prior works, in particular, the work of [74]. In this section, we consider reducing the training budget so that the total training time is less than [74]. A direct calculation shows that using 1800 epochs, the total training cost of SortNet will be less than [74]. So we simply change the number of epochs to 1800, *without turning any other hyper-parameters*. Results show that SortNet can achieve 56.05% certified accuracy for $\epsilon = 2/255$, which is still significantly higher than [74] (54.12% certified accuracy). For the $\epsilon = 8/255$ case, SortNet can achieve 39.81% certified accuracy, which is slightly lower than [74] (with a gap of -0.25%). However, note that we do not use several training tricks, such as the special initialization strategy or the special weight decay adopted in [74, 73]. On the other hand, we find that increasing the training budget of [74] to 3000 epochs does not help or even leads to a worse certified accuracy possibly due to overfitting.