# OpenReview forum: "Rethinking Lipschitz Neural Networks and Certified Robustness: A Boolean Function Perspective"
_NeurIPS.cc/2022/Conference — NeurIPS 2022 Accept_

### Official Review · Reviewer_6St4 · 2022-07-07

**Rating:** 8
**Confidence:** 4
**Soundness:** 4 excellent
**Presentation:** 4 excellent
**Contribution:** 4 excellent

**Summary:**

The paper studies certifiably robust Lipschitz continuous neural networks. To this end they first prove a couple of impossibility results for standard Lipschitz networks, i.e., feedforward networks of the form $x_{k+1} = \sigma(W_k x_k + b_k)$ where the activation $\sigma$ is $1$-Lipschitz and the weight matrices have unit norm. Their first results deal with the approximation of Boolean functions $g^B : \lbrace 0,1\rbrace^d\to\lbrace 0,1\rbrace$ and prove that no standard Lipschitz networks can achieve a robustness radius of more than $1/2d$, which obviously deteriorates in high dimensions. Consequently, these impossibility results are transferred to general Lipschitz functions by proving that the order statistics, the 1-Lipschitz function taking its input $x$ to the $k$-th largest component $x_{(k)}$, cannot be approximated by a standard 1-Lipschitz network.

The authors then continue the discussion by studying GroupSort and $\ell_\infty$-distance architectures, which are non-standard and are known to be universal Lipschitz function approximators. They prove that the simplified GroupSort architecture called MaxMin, which is commonly used since it's feasible to train, is NO universal Lipschitz function approximator. Furthermore, they prove that $\ell_\infty$-distance architectures can exactly represent certain Boolean functions and order statistics.

Motivated by these insights they suggest a new architecture, SortNet, which has both GroupSort and $\ell_\infty$-distance networks as special cases. In particular, it is a universal Lipschitz function approximator. Then, they suggest a practical version of of SortNet which replaces the expensive sorting operation by an unbiased estimator which just evaluates the maximum of a masked vector. In contrast to the MaxMin simplifcation, their simplified SortNet is still an universal approximator.

They conclude with numerical results which show that the proposed SortNet architecture is consistently slightly to significantly better than $\ell_\infty$-distance networks and significantly better than GroupSort.

######POST-REBUTTAL######
I increased my score to strong accept.

**Questions:**

I only have very few minor comments which could further improve the presentation:

- p.5, l.198: "whether they can approximate all Lipschitz functions" should be "whether they can approximate all 1-Lipschitz functions"
- bibliography: capitalization should be fixed, e.g., lipschitz, Mnist, Mma, etc...
- p.21, l.862: blank missing in "some k"

**Strengths And Weaknesses:**

Strengths:

The paper is very well-written and organized. All proofs are deferred to an appendix but at the same time they are concise, clear, and correct (I check most of them, not all). The paper has a clear golden thread: first, impossibility results for standard networks are proved, then theoretical explanations for the success and failure of the known GroupSort and $\ell_\infty$-distance networks are proved, finally a novel and unifying architecture is suggested and evaluated.

Weaknesses:

N/A

---

> ### Author Response · Authors · 2022-08-02
> **Response to Reviewer 6St4**
>
> We sincerely thank Reviewer 6St4 for the careful reading, positive feedback, and detailed examination of our proofs. We have fixed the typos in the updated version.

---

### Official Review · Reviewer_P71v · 2022-07-09

**Rating:** 8
**Confidence:** 4
**Soundness:** 3 good
**Presentation:** 4 excellent
**Contribution:** 3 good

**Summary:**

The authors present a generalization of some recently proposed variants of $\ell_\infty$-Lipschitz networks, i.e., networks that by design are 1-Lipschitz with respect to the $\ell_\infty$-norm, that come with guarantees of expressivity. This type of networks are useful as they automatically come with certificates of robustness without requiring expensive additional procedures, however some previous proposed designs suffered from low expressivity or difficulties during training.

The authors also present a stronger versions of a negative results on approximation of Lipschitz functions by standard Lipschitz neural networks (those with bounded weight matrices and element-wise 1-Lipschitz activations), as well as new negative results on the expressivity of MaxMin networks.

The proposed architecture is expensive to evaluate and hence, the authors propose to replace the layers with a stochastic approximation that drastically reduces the computation. The performance of the proposed approach is compared with state-of-the-art baselines showing promising but inconclusive results. Remarkably, the proposed architecture can be trained in a reasonable amount of time on the TinyImagenet and Imagenet datasets, which previously proposed methods found to be challenging.

*** after rebuttal ***
Authors have addressed most of my concerns. I am inclined to increase my score. Will expand later on why.

**Questions:**

1. can you provide confidence intervals for the numbers in the experimental section?
2. could you better back up the claim that the improvements over $\ell_\infty$-distance nets are significant?
3. Could you clarify if there is a theoretical advantage of SortNet over $\ell_\infty$-distance net?
4. Could you clarify in line 34 what you mean by *far from satisfactory*? Is there a number that would be considered *satisfactory*?

**Limitations:**

One limitation that I think is not mentioned, is that by introducing an stochastic estimation *between layers*, the gradient at a realization of the stochastic variables might not constitute an stochastic gradient of the full loss, and hence the optimization algorithms falls out of the SGD paradigm. Nevertheless in practice it looks like the performance obtained by this training procedure (similar to dropout) is enough to justify its heuristic nature.

**Strengths And Weaknesses:**

**Originality**: The paper builds upon previous work that introduced the GroupSort activation and another work that introduced $\ell_\infty$-distance networks, but generalizes both approaches to a more expressive architecture. The expressivity claim is backed by formal approximation results that show, in particular, that the more computationally efficient version of the GroupSort-activation network, the MaxMin network still has limited expressivity. To my knowledge, such results are new. Another new result is that the $\ell_\infty$-distance network can approximate any discrete boolean function, a result that was not part of the original paper introducing such architecture. In contrast proposition 3.1 is well-known (or a slightly less general version) but is provided mostly for completeness.

**Quality** + **Clarity**: The work backs up all claims with carefully presented proofs. Indeed the readability of the paper is vastly above the average subsmission and overall, the statements, proofs and high level ideas of the paper are easy to grasp, even though the proofs have a high level of technical difficulty. This is a major strength of this work. Nevertheless, we also find what in my opinion is the main weakness of this paper. Sadly, **the accuracy/robustness/certified-robustness numbers in the experimental evaluation are provided without a confidence interval**. Because some of these numbers appear to improve over the baselines by small amounts, it is not clear if such improvements are significant enough to warrant presenting them in **bold** and claiming a real improvement.

Perhaps the authors have had time to do multiple runs between the time of submission and rebuttal period, and could clarify if the results do not appear to be better only by chance. One issue is that such experiments are costly to evaluate, but confidence intervals, at least for the smaller scale datasets like MNIST, CIFAR10, multiple runs should be standard and I think it is reasonable to expect them. On the other hand, the time numbers are clearly lower and one would not expect the randomness to drastically change them. Another good thing is that the biggest increase in performance appears to be in the Imagenet Dataset, which is usually considered more significant. The inclusion of confidence intervals at least for MNIST+CIFAR10 would definitely improve my score.

**Significance**: The certified robustness problem is a long-standing problem and understanding the limits of what can be achieved is a significant problem. Even though $\ell_\infty$-robustness can be seen as a simplified model of real threats to computer vision systems, it can be seen as a way to devise more robust models in practice. Having a general architecture that subsumes previous promising Lipchitz networks and showing it is expressive enough to represent all boolean functions while being able to train it in a reasonable time seems significant enough. However, it was not so clear to me what is the precise theoretical advantage of SortNet (this work) vs $\ell_\infty$-distance network. I think not theoretical advantage was presented but rather it was shown through experiments that SortNet is somehow easier to train.

**miscellaneous:**
1. missing references [A] (for $\ell_2$-metric) and [B] (for $\ell_\infty$-metric) which are methods to compute the Lipschitz constant of neural networks, that can be used to regularize the loss and obtain certified robustness.
2. Proof between lines 729-730 in the appendix: "With loss of generality" -> "Without loss of generality" (I think this is what you meant)

**References**:
[A] Efficient and accurate estimation of lipschitz constants for deep neural networks.
Mahyar Fazlyab,  Alexander Robey, Hamed Hassani, Manfred Morari, George J. Pappas. NeurIPS 2019

[B] Lipschitz constant estimation of Neural Networks via sparse polynomial optimization
Fabian Latorre, Paul Rolland, Volkan Cevher. ICLR 2020

---

> ### Author Response · Authors · 2022-08-02
> **Response to Reviewer P71v (Part 2/2)**
>
> **The improvements over $\ell_\infty$-distance nets**. In this paper, we performed thorough experiments with 6 different settings:
>
> - (a) MNIST $\epsilon=0.1$
> - (b) MNIST $\epsilon=0.3$
> - (c) CIFAR-10 $\epsilon=2/255$
> - (d) CIFAR-10 $\epsilon=8/255$
> - (e) TinyImageNet $\epsilon=1/255$
> - (f) ImageNet $\epsilon=1/255$
>
> As can be seen, out of these 6 settings on 4 datasets, SortNet consistently outperforms $\ell_\infty$-distance net across 5 settings. The certified accuracy gap of the settings (c),(e),(f) are all prominent (i.e. $>1.8\%$). For MNIST, since the performance of $\ell_\infty$-distance is already above 90% (even approaching 98%), we believe the current gap is also significant. Finally, for setting (d), SortNet at least matches the performance of $\ell_\infty$-distance net while being faster to train. Considering that the problem of certified $\ell_\infty$ robustness is very challenging, we think overall the improvement over $\ell_\infty$-distance nets is significant. Nevertheless, the theoretical contribution in this paper is more important, and we believe we do not overemphasize the performance comparison with $\ell_\infty$-distance network in this paper.
>
> **Theoretical advantage of SortNet vs $\ell_\infty$-distance network**. Thanks for the good question. One advantage of the general SortNet is that it can precisely represent piecewise linear 1-Lipschitz functions, while $\ell_\infty$-distance nets cannot. Indeed, a straightforward calculation shows that the gradient norm of $\ell_\infty$-distance nets with respect to the input must be exactly one, and the gradient is sparse and has exactly one non-zero element. Therefore, when fitting piecewise linear 1-Lipschitz functions using $\ell_\infty$-distance nets, the learned function will have a ''jagged'' shape. As a result, the approximation error cannot be zero and depends on the network size. On the other hand, SortNet with a finite size can perfectly fit such functions.
>
> **Clarification of line 34**. Sorry for the misleading. The previous work of Anil et al. achieved 79% certified accuracy under $\epsilon=0.1$ and 2% accuracy under $\epsilon=0.3$ on MNIST according to their experiments, which we think can be largely improved. We have changed the words ''far from'' to the word ''not''. We believe the modified statement would be appropriate.
>
> **Miscellaneous**. We have discussed these references with our work in the updated version. We also thank the reviewer for pointing out the typo and we have corrected it.
>
> We hope our response can clarify your concerns. We are happy to go into more detail regarding any of them and we look forward to your reply.

---

> ### Author Response · Authors · 2022-08-02
> **Response to Reviewer P71v (Part 1/2)**
>
> We sincerely thank Reviewer P71v for the careful reading, valuable suggestions, and positive feedback. Below we would like to give detailed responses to each of your comments.
>
> **Providing confidence intervals**. Thanks for the suggestion. We have followed your advice and run multiple experiments on MNIST and CIFAR-10 during these days. Due to the time limit, we have not completed the ImageNet experiments yet. The current results are shown as follows, where we present the results of five independent runs and report the 95% confidence interval. These results are run on NVIDIA RTX 3090 GPUs.
>
> MNIST ($\epsilon=0.1$):
>
> |           | Exp1  | Exp2  | Exp3  | Exp4  | Exp5  | Confidence interval |
> | ----      | ----  | ----  | ----  | ----  | ----  | ------------------- |
> | Clean     | 99.13 | 99.03 | 99.08 | 99.02 | 99.00 | 99.05 ± 0.04        |
> | Certified | 98.22 | 98.05 | 98.12 | 98.26 | 98.14 | 98.16 ± 0.07        |
>
> MNIST ($\epsilon=0.3$):
>
> |           | Exp1  | Exp2  | Exp3  | Exp4  | Exp5  | Confidence interval |
> | ----      | ----  | ----  | ----  | ----  | ----  | ------------------- |
> | Clean     | 98.58 | 98.56 | 98.54 | 98.54 | 98.46 | 98.54 ± 0.04        |
> | Certified | 93.60 | 93.43 | 93.45 | 93.56 | 93.40 | 93.49 ± 0.08        |
>
> CIFAR-10 (SortNet $\epsilon=2/255$):
>
> |           | Exp1  | Exp2  | Exp3  | Exp4  | Exp5  | Confidence interval |
> | ----      | ----  | ----  | ----  | ----  | ----  | ------------------- |
> | Clean     | 65.86 | 65.96 | 65.69 | 65.98 | 65.79 | 65.86 ± 0.11        |
> | Certified | 56.65 | 56.67 | 56.20 | 56.23 | 56.29 | 56.41 ± 0.20        |
>
> CIFAR-10 (SortNet $\epsilon=8/255$):
>
> |           | Exp1  | Exp2  | Exp3  | Exp4  | Exp5  | Confidence interval |
> | ----      | ----  | ----  | ----  | ----  | ----  | ------------------- |
> | Clean     | 54.84 | 54.30 | 54.59 | 54.28 | 54.70 | 54.54 ± 0.22        |
> | Certified | 40.39 | 40.13 | 40.14 | 39.96 | 40.05 | 40.13 ± 0.14        |
>
> CIFAR-10 (SortNet+MLP $\epsilon=2/255$):
>
> |           | Exp1  | Exp2  | Exp3  | Exp4  | Exp5  | Confidence interval |
> | ----      | ----  | ----  | ----  | ----  | ----  | ------------------- |
> | Clean     | 67.64 | 67.72 | 67.57 | 67.59 | 67.72 | 67.65 ± 0.06        |
> | Certified | 56.80 | 56.94 | 56.70 | 56.63 | 56.30 | 56.67 ± 0.21        |
>
> CIFAR-10 (SortNet+MLP $\epsilon=8/255$):
>
> |           | Exp1  | Exp2  | Exp3  | Exp4  | Exp5  | Confidence interval |
> | ----      | ----  | ----  | ----  | ----  | ----  | ------------------- |
> | Clean     | 54.80 | 54.13 | 54.33 | 54.02 | 54.39 | 54.33 ± 0.26        |
> | Certified | 39.56 | 39.99 | 39.76 | 39.45 | 39.70 | 39.70 ± 0.18       |
>
> ImageNet (SortNet+MLP $\epsilon=1/255$):
>
> |           | Exp1  | Exp2  | Exp3  |
> | ----      | ----  | ----  | ----  |
> | Clean     | 13.21 | 13.48 | 13.37 |
> | Certified | 9.00  | 9.02  | 8.97  |
>
> It can be seen that in most settings, we still significantly outperform other baselines of this paper. The only exception is the CIFAR-10 ($\epsilon=8/255$) setting, where SortNet still achieves the best but the gap is within the confidence interval. For the ImageNet dataset, we find the variance of different runs to be very small possibly because the test set has 50000 images (10x more than MNIST and CIFAR-10). We will update these results in our paper once the ImageNet results are ready.

---

### Official Review · Reviewer_G1MR · 2022-07-10

**Rating:** 8
**Confidence:** 4
**Soundness:** 4 excellent
**Presentation:** 4 excellent
**Contribution:** 4 excellent

**Summary:**

This paper presents novel theoretical results and a novel architecture for designing Lipschitz constrained neural networks (with respect to the infinity norm). The theoretical results provide new insights on two prior approaches and give a strong justification for the method introduced by the authors. The authors also provide a novel stochastic approximation to train their model efficiently. If I understand correctly, this comes at with a cost to expressiveness but is seemingly insignificant. Overall, I found this paper to be clearly written and with significant theoretical and empirical results.

The theoretical results are thorough and exciting. Unlike some prior work (e.g. Anil et al.) the authors provide (lower) bounds on approximation quality in addition to exact completeness results. Theorems 3.5 and 3.6 show that standard Lipschitz neural networks fail to achieve robustness as input dimensionality increases for simple 1-Lipschitz functions. These results are not surprising but are more concrete than prior results that I am aware of. Theorem 3.8 then gives a finite-dimensional completeness result for the distance nets that were introduced in prior work (though universal approximation results have already been shown, e.g. Zhang et al. 2021). Theorem 3.9 then shows that the previously introduced MaxMin networks require depth that grows with input dimension to approximate the order statistics accurately (confirming a negative result that had been explored empirically by Anil et al. 2019 and theoretically by Huster et al. 2019).

Empirically, the introduced SortNet architecture shows excellent performance (relative to the class of models it is compared against). SortNet typically achieves clean accuracy that is comparable to the clean accuracy of other certified networks but achieves improved certified robustness across the board. Importantly, it is also faster to train and compute certified bounds than other methods.

**Questions:**

"Based on our knowledge of Boolean circuit theory" --- as somebody without this knowledge, I'm keen to know a little more about this. Can you provide some brief intuition (in response, and in the paper) for why MaxMin nets require O(d) scaling while the distance nets appreciate constant depth.

The SortNet architecture introduces the weights as network parameters. To make the network training efficient, the weights are fixed to a geometric series. This contains the distance nets as a special case, but once the weights are fixed the networks are now, assuming an identity activation, less expressive than GroupSort networks that are able to learn the weights alongside using the order statistics. Is learning the weights entirely unimportant? Is it beneficial to fix them?


### Minor comments

In line 227 you refer to the absolute value function as an example of a GNP element-wise activation function. This is just a small comment that absolute value and MaxMin are equivalent in terms of expressivity (at least in both the 2-norm and infinity-norm settings). See Appendix A.3 in Anil et al. (it is shown explicitly for 2-norm).

L198-199: The negative answer is only shown for so-called Standard 1-Lipschitz neural networks.

L170: "Lipschitz" -> "1-Lipschitz"

L274: "MinMax networks" should be "MaxMin" networks as elsewhere in the paper

**Limitations:**

The authors do discuss limitations but only very briefly. So far as I can tell, they only discuss:

- Not proving a tight upper bound on MaxMin networks for general Boolean functions
- Not treating rho as a learnable parameter

I feel that there are more details to be discussed here, some of which I raise in review. A broader discussion of the limitations of this research relative to related work would be appreciated. All of these may not be necessary, but here are some areas that could be discussed:
- Architecture limited to fully-connected networks
- Results/architecture only applicable under l_p norm constraints [though I believe some of the same ideas could be transferred more generally]
- Limitations of norm-bounded threat models for robustness in practice
- Limitations of margin-based robustness guarantees (in terms of training and certification)
- Fixed weights when using stochastic approximation (related to fixed rho, but more general)

**Strengths And Weaknesses:**

## Strengths

The theoretical results are well presented and complete, providing upper and lower bounds on approximation error and highlighting order statistics as an important class of functions. There is more to do in this space. For example, the authors show that order statistics are hard to learn for MaxMin networks but why is this important for classification tasks? But I consider the paper to address all reasonable areas of interest in sufficient depth.

I found the SortNet architecture compelling and the stochastic approximation of the linear function of the order statistics is very neat. The dropout connection is also interesting (and perhaps nested dropout is worth a mention).

The empirical results are thorough and compare across the appropriate metrics for the tasks considered. As always, including error bars would strengthen the empirical results further.

## Weaknesses

The authors address only the infinity norm and make no claims on how these results may generalize to other l_p norms beyond section 3.1.

The authors results are only (obviously) applicable to fully-connected networks. This is a limitation which is shared by other work in the area, but not exclusively (Li et al. 2019 study 1-Lipschitz convolutions with respect to the 2-norm). It is unclear whether incorporating convolutional layers would improve performance on tasks which typically demand them (ImageNet).

Generally, it seems that the larger community has disengaged a little with norm-bounded threat models for adversarial robustness. This _is_ still an active area and I personally believe that certified robustness guarantees in this model remain an important topic of study. However, I reduce my score a little, in part, because I do not think the empirical results presented here would be enough to make waves in the deep learning community more broadly.

---

> ### Author Response · Authors · 2022-08-02
> **Response to Reviewer G1MR (Part 3/3)**
>
> **Other $\ell_p$ norms**. Indeed, it is a good question whether our results can be generalized to other $\ell_p$-norms. In the updated version, we have discussed the $\ell_p$ case in detail in lines 458-462. We believe the main impossibility results should approximately hold when $p$ is large, and we will rigorously write it down in future work. However, it definitely does not apply in the standard $\ell_2$-norm: in this case MaxMin is equivalent to the absolute value function in terms of expressive power, as pointed out by Anil et al. [1] (also by Reviewer G1MR), and empirical results suggest that these $\ell_2$ Lipschitz networks are expressive [2] (although it is still a fantastic open problem to formally prove that they are universal approximators).
>
> Therefore, our results reflect an interesting ``phase transition'' in the expressive power of standard Lipschitz networks when $p$ is switched from 2 to a large number. Coincidentally, a similar limitation is also proved when using randomized smoothing, which suffers from the curse of dimensionality when $p>2$ [3]. This raises an interesting question of why the effect of $p$ is very similar for both methods and how things change as $p$ increases. We will investigate these aspects in future work.
>
> **Regarding convolutional architectures**. The proposed network can be applied to the convolutional architecture by treating the image pixels in each convolutional window as the input vector and using shared weights and biases. We have tried the convolutional architecture in ImageNet-like experiments (see Table 5 in Appendix E.2) using a simple architectural design, and the results can be improved (SortNet 2x in Table 3). For future work, we are interested in designing better convolutional architectures with higher performance.
>
> **Minor Comments**. Thanks for pointing out these typos. We have corrected them in the updated version.
>
> The MaxMin activation and absolute value activation are proved to be equivalent for the $\ell_2$-norm in Anil et al. [1]. But for the $\ell_\infty$-norm, Anil et al. did not show the equivalence between these two activations. In this paper, we prove that they are actually *not* equivalent, and the MaxMin activation is *strictly* more powerful than the absolute value in the $\ell_\infty$-norm case.
>
> **Limitations**. Thanks for the suggestion. We have followed your advice and discussed the limitations in detail in the updated version. We believe most of them in your list are present, e.g., general $\ell_p$-norm perturbations, margin-based certification, architectures beyond fully-connected networks, and using learnable weights. We also discussed the potential broader impact of this paper.
>
> [1] Sorting out Lipschitz function approximation. ICML 2019.
>
> [2] Improved deterministic l2 robustness on CIFAR-10 and CIFAR-100. ICLR 2022.
>
> [3] Randomized Smoothing of All Shapes and Sizes. ICML 2020.

---

> > ### Comment · Reviewer_G1MR · 2022-08-08
> > **Response to rebuttal**
> >
> > Thank you for the detailed discussion of the points that I raised.
> >
> > The new theoretical result is exciting and helps to complete the previously presented theoretical work. The additional results, that provide the error bars, are also a great addition.
> >
> > The discussion here on other $\ell_p$ norms is interesting. I think this discussion would make a valuable addition to the paper. If not in the main text, then in the supplementary material.
> >
> > I will maintain my score.

---

> ### Author Response · Authors · 2022-08-02
> **Response to Reviewer G1MR (Part 2/3)**
>
> **Including error bars would strengthen the empirical results further**. Thanks for the suggestion. We have run multiple experiments on MNIST and CIFAR-10 during these days. Due to the time limit, we have not completed the ImageNet experiments yet. The current results are shown as follows:
>
> MNIST ($\epsilon=0.1$):
>
> |           | Exp1  | Exp2  | Exp3  | Exp4  | Exp5  | Confidence interval |
> | ----      | ----  | ----  | ----  | ----  | ----  | ------------------- |
> | Clean     | 99.13 | 99.03 | 99.08 | 99.02 | 99.00 | 99.05 ± 0.04        |
> | Certified | 98.22 | 98.05 | 98.12 | 98.26 | 98.14 | 98.16 ± 0.07        |
>
> MNIST ($\epsilon=0.3$):
>
> |           | Exp1  | Exp2  | Exp3  | Exp4  | Exp5  | Confidence interval |
> | ----      | ----  | ----  | ----  | ----  | ----  | ------------------- |
> | Clean     | 98.58 | 98.56 | 98.54 | 98.54 | 98.46 | 98.54 ± 0.04        |
> | Certified | 93.60 | 93.43 | 93.45 | 93.56 | 93.40 | 93.49 ± 0.08        |
>
> CIFAR-10 (SortNet $\epsilon=2/255$):
>
> |           | Exp1  | Exp2  | Exp3  | Exp4  | Exp5  | Confidence interval |
> | ----      | ----  | ----  | ----  | ----  | ----  | ------------------- |
> | Clean     | 65.86 | 65.96 | 65.69 | 65.98 | 65.79 | 65.86 ± 0.11        |
> | Certified | 56.65 | 56.67 | 56.20 | 56.23 | 56.29 | 56.41 ± 0.20        |
>
> CIFAR-10 (SortNet $\epsilon=8/255$):
>
> |           | Exp1  | Exp2  | Exp3  | Exp4  | Exp5  | Confidence interval |
> | ----      | ----  | ----  | ----  | ----  | ----  | ------------------- |
> | Clean     | 54.84 | 54.30 | 54.59 | 54.28 | 54.70 | 54.54 ± 0.22        |
> | Certified | 40.39 | 40.13 | 40.14 | 39.96 | 40.05 | 40.13 ± 0.14        |
>
> CIFAR-10 (SortNet+MLP $\epsilon=2/255$):
>
> |           | Exp1  | Exp2  | Exp3  | Exp4  | Exp5  | Confidence interval |
> | ----      | ----  | ----  | ----  | ----  | ----  | ------------------- |
> | Clean     | 67.64 | 67.72 | 67.57 | 67.59 | 67.72 | 67.65 ± 0.06        |
> | Certified | 56.80 | 56.94 | 56.70 | 56.63 | 56.30 | 56.67 ± 0.21        |
>
> CIFAR-10 (SortNet+MLP $\epsilon=8/255$):
>
> |           | Exp1  | Exp2  | Exp3  | Exp4  | Exp5  | Confidence interval |
> | ----      | ----  | ----  | ----  | ----  | ----  | ------------------- |
> | Clean     | 54.80 | 54.13 | 54.33 | 54.02 | 54.39 | 54.33 ± 0.26        |
> | Certified | 39.56 | 39.99 | 39.76 | 39.45 | 39.70 | 39.70 ± 0.18       |
>
> ImageNet (SortNet+MLP $\epsilon=1/255$):
>
> |           | Exp1  | Exp2  | Exp3  |
> | ----      | ----  | ----  | ----  |
> | Clean     | 13.21 | 13.48 | 13.37 |
> | Certified | 9.00  | 9.02  | 8.97  |
>
> We will update these results in our paper once the ImageNet results are ready.

---

> ### Author Response · Authors · 2022-08-02
> **Response to Reviewer G1MR (Part 1/3)**
>
> We sincerely thank Reviewer G1MR for the insightful comments, valuable suggestions, and positive feedback. Below we would like to give detailed responses to each of your comments.
>
> **Why do MaxMin nets require $O(d)$ scaling**? Thanks for the question. We are excited to show you that we have successfully proved this conjecture during these days (see Appendix B.7 in the updated version). The proof is quite non-trivial and brings new insights into how MaxMin networks express Boolean functions, which we believe is novel and interesting. We give a brief proof sketch below, which can also be found in Section 3.4 in the updated version.
>
> The key insight is that for any Boolean function, if it can be represented by some MaxMin network $f$, it can also be represented by a *special* MaxMin network with the same topology as $f$, such that all the weight vectors $w$ are sparse with at most one non-zero element, either 1 or -1 (Lemma B.22 and Corollary B.23). We prove this lemma by induction on the depth of the network. The induction step is divided into two parts and five sub-cases, for each of which we give an explicit construction of the network parameters.
>
> Corollary B.23 implies that weight vectors only perform the *neuron selection* operation and *have no use* in representing Boolean functions. Therefore, MaxMin networks reduce to *2-ary Boolean circuits*, i.e., directed acyclic graphs whose internal nodes are logical gates including NOT and the 2-ary AND/OR. Note that for a 2-ary Boolean circuit with $M$ layers and a scalar output, the number of nodes will not exceed $2^{M+1}-1$ (achieved by a complete binary tree). However, the classic result in Boolean circuit theory (Shannon 1942) showed that for most Boolean functions of $d$ variables, a lower bound on the minimum size of 2-ary Boolean circuits is $\Omega(2^d/d)$, which thus yields $M=\Omega(d)$ and concludes the proof.
>
> We also remark that as a corollary, $M$-layer MaxMin networks are not universal approximators for the $d$-dimensional 1-Lipschitz function class if $M=o(d)$.
>
> **Regarding fixing the weights in SortNet**. Thanks for the good question. The above proof may give insights into this question and further justify the design of SortNet. In particular, we prove that when using MaxMin networks to express Boolean functions, the learned weights will inherently be sparse, and the role of biases is much more important than the weights. We suspect such a result may transfer to general SortNet architectures with absolute value activation. In practice, while the learned function is clearly not Boolean-valued, we believe the intuition still makes sense: the learned weight may still have a certain degree of sparsity, and the biases are more important than the weights. This possibly justifies that fixing the weights to geometric series is reasonable. Nevertheless, it is still likely that better results can be achieved using trainable weights. We will study how to design efficient training strategies for the general setting in future work.

---

### Official Review · Reviewer_wXX6 · 2022-07-11

**Rating:** 5
**Confidence:** 3
**Soundness:** 3 good
**Presentation:** 2 fair
**Contribution:** 2 fair

**Summary:**

1. Problem/Motivation
- The paper looks into the relationship between certified robustness and Lipschitz continuity for the $l_\infty$ case.
- It has been observed that Lipschitz networks don't work for $l_\infty$ as well as they work for $l_2$ setting. However, recent work from Zhang et al. created some particular 1-Lipschitz network which has good robustness.
- The authors try to understand this phenomenon.

2. Methodology/Technical Finding

A) Methodology
- Authors study this question by using discrete Boolean functions and prove some impossibility results (negative results) for this setting.
- This relates to the difficulty in training certifiably robust Lipschitz network for the $l_\infty$ perturbations setting.

B) Explaining performance of recent Lipschitz networks (Zhang et al.)
- Authors use the above machinery to examine how some recent networks for this setting work well thus justifying their empirical performance.

C) Designing even better Lipschitz networks
- From the above insights, authors propose new networks.

3. Experiments
- Authors show the empirical performance of the proposed SortNet, where the methods perform almost at par (maybe slightly better) than recent methods.

########## POST REBUTTAL ##############
After going through other reviews and the authors rebuttal, I have decided to keep my score as it is (borderline accept) and not increase the rating. My main concern remains the same, that we need to move beyond theoretically convenient settings, to scale to networks used in practise.
The authors shared 2 threads of work, which I am aware of. Here are my concerns with them:
- Randomised smoothing: The test time cost of randomised smoothing is very high as it needs to run for many perturbations. Thus even if they work on large datasets, the methods are not practical.
- Certified robustness using Lipschitz networks: Are we able to convince the practitioners to use Lipschitz networks?

**Questions:**

These are suggestions regarding writing. I hope it is useful for authors.
1. Better sectioning
- Look at the summary I wrote and how I divided into subgroups. If you do so throughout the paper, it would be great. Start from the introduction where you can have these as different paragraphs with paragraph names as bold.
- Then match them with same names (either as paragraphs or subsections) in the method section.

2. In the method section, before bringing a theorem, you should try to motivate the theorem. More space should be spent of motivation and linking theorems to the whole them or theme of the subgroup (see point 1).

**Limitations:**

The work addresses Certified Robustness, which is on the reliability line of work, thus positive societal impact.

**Strengths And Weaknesses:**

I have given a thorough summary in the above box.

Strengths
- The technical part of the paper is solid. Authors have used Boolean functions to explain the workings/robustness of Lipschitz networks.

Weaknesses:
1. Writing: The method section can be improved. It is a bit dense and hard to follow. I have provided suggestions for improvement in the Questions box.
2. Problem Relevance: I am not sure about this whole line of work. Certified robustness has been there for a while now. It is not scaling up and there is less chance of scaling up. So, I am not sure if this is so relevant for the community.

---

> ### Author Response · Authors · 2022-08-02
> **Response to Reviewer wXX6**
>
> We thank Reviewer wXX6 for the comments and suggestions. We have followed your advice and revised our submission accordingly. In particular, we can elaborate further on the motivation of each theorem given the extra one page. We have also modified the introduction part by splitting it into different paragraphs with paragraph names as bold as you suggested.
>
> Regarding the problem relevance, there have been a large number of recent advances in certified robustness, which successfully enable network training scaling up to **ImageNet**-like dataset. For example, using randomized smoothing [1] one can efficiently train a deep neural network (such as ResNet50) on Imagenet with good robustness guarantees under $\ell_2$ perturbation, and a recent work even achieved 71% top-1 accuracy under $\epsilon= 0.5$ [2]. There are also many papers focusing on certified robustness using Lipschitz networks this year, e.g. [3] (ICLR22 spotlight), [4] (ICLR22), [5] (ICML22). We believe this direction is promising and there is a high opportunity that we can learn robust models with certified guarantees for both $\ell_2$/$\ell_\infty$ perturbations.
>
> We hope our response as well as the new version of this paper can clarify your concerns. We are happy to go into more detail regarding any of them and we look forward to your reply.
>
> [1] Certified Adversarial Robustness via Randomized Smoothing. ICML 2019.
>
> [2] (Certified!!) adversarial robustness for free! arxiv preprint 2206.10550.
>
> [3] Improved deterministic l2 robustness on cifar-10 and cifar-100. ICLR 2022.
>
> [4] Boosting the Certified Robustness of L-infinity Distance Nets. ICLR 2022.
>
> [5] A dynamical system perspective for Lipschitz neural networks. ICML 2022.

---

> > ### Author Response · Authors · 2022-08-07
> > **Looking forward to feedback**
> >
> > Dear reviewer wXX6:
> >
> > We believe we have incorporated most of your suggestions in the updated version. We would be grateful if you can confirm whether our response has addressed your concerns and please let us know if any questions remain. Thank you for your consideration!

---

### Author Response · Authors · 2022-08-02
**General Response: New Results & Paper Updates**

We sincerely thank all the reviewers and the area chair for their efforts in reviewing our paper. We would like to take this opportunity to highlight that we have recently proved an important theoretical result conjectured in the initial submission, which we believe can further strengthen this work.

**New results**. We successfully proved the conjecture raised in Section 3.4, showing that MaxMin networks require a depth of $\Omega(d)$ to represent Boolean functions of $d$ variables. The proof is more challenging than other proofs in this paper and leverages the tool of Boolean circuit theory in computer science. Please see Section 3.4 in the updated version for a precise description as well as a proof sketch. Significance and further implications of this result are shown below:

1. Such a lower bound is much stronger than the result of representing order statistics which requires a depth of $\Omega(\log_2 d)$ (Theorem 3.9).

2. As a corollary, universal approximation is impossible if the depth of MaxMin networks is $o(d)$ (Corollary 3.11).

3. Our proof discovers that the weights in a MaxMin network will inherently become sparse when learning Boolean functions, and MaxMin networks fit Boolean functions purely by using the biases and the activation. This further justifies the design of SortNet.

4. We discovered an interesting relationship between MaxMin/$\ell_\infty$-distance networks and Boolean circuits. It links the certified robustness area to the field of theoretical computer science (TCS), which may inspire future works to solve more deep learning problems using various tools in TCS. Therefore, we believe this paper may have a broader impact beyond the certified robustness community.

**We solved an open problem raised in a concurrent work**. A concurrent work arXiv:2204.06233 [1] investigated the expressive power of general 1-Lipschitz networks with piecewise linear activation (which they called the spline networks). They proved that using 1-Lipschitz piecewise linear activation with 3 linear regions, the corresponding network achieves the maximum expressive power compared with other Lipschitz activations and can approximate any one-dimensional 1-Lipschitz function. They pose the high dimension setting as an important open problem (in page 14). However, Theorem 3.6 addresses the open problem with a negative answer, stating that such networks are not expressive even for the two-dimensional setting.

**Paper updates**. Below we highlight the major updates of the revised submission.  All major changes in the main paper have been marked $\textcolor{red}{\text{red}}$.

- Section 3.3, lines 211-217: add justification of order statistics as an important class of functions for measuring the expressive power of neural networks.

- Section 3.3, lines 239-246: add discussions of the concurrent work [1] and show that we addressed an open problem in their paper.

- Section 3.4, lines 288-310: add the new result (Theorem 3.10), which gives an $\Omega(d)$ lower bound on the depth of MaxMin networks in representing Boolean functions.

- Section 6, lines 458-470: add detailed discussions on the limitation of this paper.

- Lines 471-473: discuss the broader impact of this paper.

- Appendix B.7: give a proof of Theorem 3.10.

[1] Approximation of Lipschitz functions using deep spline neural networks. arXiv preprint 2204.06233.

---

### Meta-Review · Area_Chair_5C4D · 2022-08-23

**Recommendation:** Accept
**Confidence:** Certain

**Metareview:**

The paper presents novel theoretical results and a novel architecture for designing Lipschitz constrained neural networks (with respect to the infinity norm). The authors have addressed all the concerns from the reviewers properly. All the reviewers agreed that the paper contains significant contributions and should be accepted at NeurIPS 2022.

**Award:**

No

---

### Decision · Program_Chairs · 2022-09-14

Accept